# On the (un) interpretability of Ensembles: A Computational Analysis

## Abstract

Despite the widespread adoption of ensemble models, it is widely acknowledged within the ML community that they offer limited *interpretability*. For instance, while a single decision tree is considered interpretable, ensembles of decision trees (e.g., boosted-trees) are usually regarded as black-boxes. Although this reduced interpretability is widely acknowledged, the topic has received only limited attention from a theoretical and mathematical viewpoint. In this work, we provide an elaborate analysis of the interpretability of ensemble models through the lens of *computational complexity* theory. In a nutshell, we explore different forms of explanations and analyze whether obtaining explanations for ensembles is strictly computationally less tractable than for their constituent base models. We show that this is indeed the case for ensembles that consist of interpretable models, such as decision trees or linear models; but this is not the case for ensembles consisting of more complex models, such as neural networks. Next, we perform a fine-grained analysis using parameterized complexity to measure the impact of different problem parameters on an ensemble's interpretability. Our findings reveal that even if we shrink the *size* of all base models in an ensemble substantially, the ensemble as a whole remains intractable to interpret. However, an analysis of the *number* of base models yields a surprising dynamic — while ensembles consisting of a limited number of decision trees can be interpreted efficiently, ensembles that consist of a small (even *constant*) number of linear models are computationally intractable to interpret.

## 1 Introduction

Ensemble learning is a widely acclaimed technique in ML that leverages the strengths of multiple models instead of relying on a single one. This approach has been proven to enhance predictive accuracy, mitigate variance, and handle imbalanced or noisy datasets effectively (Dong et al. (2020); Sagi & Rokach (2018)).

However, a significant challenge with ensemble models is their perceived lack of interpretability (Guidotti et al. (2018); Sagi & Rokach (2021); Hara & Hayashi (2018); Bénard et al. (2021); Kook et al. (2022)). The reason behind this is straightforward — utilizing several models simultaneously makes the decision-making process inherently more complex, and thus more challenging to understand. For instance, while it is feasible to trace the decision-making path in a single decision tree, this level of straightforward traceability is not achievable in tree ensembles (Sagi & Rokach (2021); Hara & Hayashi (2018); Bénard et al. (2021)). Despite the general acknowledgment of this issue in the ML community, there has been only limited exploration of the interpretability of ensemble models, from a theoretical and rigorous viewpoint.

In this work, we aim to establish a sound theoretical basis for evaluating the interpretability of ensemble models. Recent research provides a basis for a rigorous study of interpretability by incorporating the principles of *computational complexity* theory (Barceló et al. (2020); Wäldchen et al. (2021); Arenas et al. (2022)). Specifically, Barceló et al. (2020) suggest that by exploring the computational complexity required to produce various types of explanations on different ML models, we can improve our grasp of interpretability. There, a model is deemed interpretable if it is possible to efficiently generate an explanation for its decisions. Conversely, if generating an explanation is computationally difficult, the model is considered uninterpretable.

We perform such an analysis for ensemble interpretability, by examining the computational complexity of deriving various types of explanations for ensembles, in comparison to their individual base models. We focus on *formal* notions of explanations that are grounded in logical and mathematical guarantees. Such explanations are often mentioned within a subfield of interest known as *formal explainable AI* (Marques-Silva & Ignatiev (2022)). Providing explanations with mathematical guarantees is vital in systems where safety is paramount. Moreover, such guarantees enable a thorough evaluation of the computational complexity involved in deriving explanations. Within this scope, we focus on three widely recognized formal types of explanations:

1. **Sufficient Reason Feature Selection,** where, we analyze the complexity of selecting the $k$ most important features based on the common *sufficiency* criterion. We also explore choosing subsets of *minimal cardinality*, and a *probabilistic* version of these explanations.

2. **Contrastive Explanations,** where we study the complexity involved in identifying input features that represent the smallest change required for *altering* a given prediction.

3. **Shapley Value Feature Attributions,** where we analyze the complexity of providing an *exact* computation of the highly incorporated *shapley value* attribution index.

We analyze the computation of these explanations across a broad spectrum of ensembles, including any that employ either *voting* or *weighted-voting* inference methods (such as XGBoost, Adaboost, Gradient Boosting, Random Forest, etc; a complete formalization can be found in Appendix A). Moreover, we focus on exploring three common types of *base models* that are integral to these ensembles: (i) decision trees, (ii) linear models, and (iii) neural networks. Our reason for choosing these base-models is that they are both widely used and also cover a broad spectrum of interpretability. Consequently, we are able to analyze the interpretability of diverse kinds of ensembles.

Our findings present a range of computational complexity results for generating various explanations across different types of ensembles. These results span strict computational complexity classes within the polynomial hierarchy (e.g., PTIME, NP, $\Sigma_2^P$), the counting hierarchy (e.g., #P), and parameterized complexity classes (Downey & Fellows, 2012) (e.g., FPT, W[1]). These results highlight how specific parameters, such as the number of base models and their sizes, influence the complexity of interpreting ensembles.

## 1.1 MAIN CONTRIBUTIONS

1. **Ensembles are computationally hard to interpret.** We provide a range of intractability results (NP, $\Sigma_2^P$, #P-Hardness, etc.) for generating multiple types of explanation, for diverse *arbitrary* ensembles, emphasizing foundational limitations of computing explanations for these models.

2. **Ensembles are *less* computationally interpretable.** Aiming to show that the ensemble aggregation itself is responsible for this intractability, we demonstrate *strict complexity separations* between the complexity classes corresponding to providing explanations for an ensemble's base models (e.g., linear models, decision trees), which can often be obtained in polynomial or linear time, and their ensembles which are intractable to explain. However, for expressive base models like neural networks, we prove that no such gaps exist.

3. **Shrinking base-model sizes does not make an ensemble more computationally interpretable.** Having shown that interpreting arbitrary ensembles is intractable, we adopt a *parameterized complexity* (Downey & Fellows, 2012) perspective to investigate whether *simplifying an ensemble* can render it efficiently interpretable. Our first result reveals that under all of our analyzed settings, ensembles with constant-size base models remain intractable (NP, $\Sigma_2^P$, #P-Hard, etc.). This confirms that reducing the sizes of base-models does not improve an ensemble's computational interpretability.

4. **Shrinking the number of trees *does* make an ensemble more computationally interpretable.** A more optimistic result emerges in a different simplified setting, showing that reducing the number of decision trees in an ensemble (e.g., Random Forests, XGBoost) enables tractable (poly-time) explanation computation for many explanation types. While our findings indicate that this complexity is not solely governed by the number $k$ of models (this is demonstrated by proving W[1], coW[1]-Hardness, etc.), we prove that under certain relaxations, such as limiting the number of leaves in each tree, the complexity of interpreting

them is solely determined by the number $k$ of decision trees (i.e., they are fixed-parameter tractable). These results pave the way for practical and efficient algorithmic implementations in this context.

5. **An ensemble with only 2 linear models is already computationally uninterpretable.** However, for another type of base-model widely regarded as highly interpretable — linear models — we demonstrate that an ensemble comprising just two linear models already becomes intractable to interpret. This demonstrates how rapidly the integration of linear models can cause a significant loss of interpretability, and give rise to negative complexity results even in a highly simplified scenario involving just two base models.

We believe that these results provide a rigorous, mathematically grounded, and nuanced understanding of ensemble interpretability while also highlighting the potential of utilizing complexity theory in developing a formal understanding of interpretability.

Due to space constraints, we include only a brief outline of the proofs of our various claims within the paper, and relegate the complete proofs to the appendix.

## 2 PRELIMINARIES

**Complexity Classes.** This paper assumes that readers have a basic understanding of standard complexity classes, including polynomial time (PTIME) and nondeterministic polynomial time (NP and coNP). We also discuss the common class within the second order of the polynomial hierarchy, $\Sigma_2^P$. This class contains problems that can be solved within NP when provided access to an oracle capable of solving coNP problems in constant time. It clearly holds that NP, coNP $\subseteq \Sigma_2^P$; and it is also widely believed that NP, coNP $\subsetneq \Sigma_2^P$ (Arora & Barak (2009)). The paper additionally discusses the complexity class #P which represents the total count of accepting paths in a polynomial-time nondeterministic Turing machine. It is widely believed that $\Sigma_2^P \subsetneq$ #P (Arora & Barak (2009)).

**Domain.** We consider a set of $n$ input features $\{1, \ldots, n\}$ with the assignments: $\mathbf{x} := (\mathbf{x}_1, \ldots, \mathbf{x}_n)$. We denote the entire feature space by $\mathbb{F} := \{0, 1\}^n$, and focus on different forms of explanations for interpreting the classifier $f : \mathbb{F} \to \{0, 1\}$. We study *local* forms of explanations, meaning those that interpret a prediction over a specific input instance $\mathbf{x}$. Following common practice, we focus on Boolean input and output values, which enable a cleaner presentation (Arenas et al. (2021a); Wäldchen et al. (2021); Barceló et al. (2020)). However, many of our findings are also applicable to cases involving real values (see Appendix D for additional information).

**Ensembles and Base Models.** We regard an ensemble as a classification function, comprised of $k$ different base models (each being a classification function in itself). As we focus on post-hoc explanations, our interest lies in the inference of these models. This paper focuses on two key types of ensemble inference: *majority voting* and *weighted voting*. In the majority voting setting, we assume that the prediction is obtained via a majority vote between the $k$ participating base models. In the weighted-voting setting, we assume that each base-model is associated with a weight and that the ensemble prediction is based on an aggregation of all individual predictions. All of our complexity results apply for *both* families of ensembles. Moreover, we consider three different families of *base model* types: (i) Free Binary Decision diagrams (FBDDs), which are a generalization of decision trees; (ii) Perceptrons (for our analysis of linear models); and (iii) Multi-Layer Perceptrons (MLPs) with ReLU activation units. A comprehensive and detailed formalization of all base-model types and ensembles is relegated to Appendix A.

In essence, this formalization covers a broad spectrum of popular ensemble techniques, including random forests, boosted trees (e.g., XGBoost), along with other types of ensembles that incorporate decision trees, neural networks, or various linear models (for example, logistic regression, SVM classifiers, etc.). Although our primary emphasis is on classification models, several of our findings are also applicable to regression scenarios (see Appendix D for further details), thus making them relevant for different regression approaches, (e.g., linear regression models, etc.). While we mainly discuss *homogeneous* ensembles (composed of identical model types), many of our results carry on to *heterogeneous* ensembles (comprising various model types) as well. Further details are provided in Appendix D.

## 3 EXPLAINABILITY QUERIES

While the concept of model interpretability can vary based on perspective, we focus on several widely recognized forms of explanations from the literature. In line with previous research (Barceló et al. (2020); Arenas et al. (2021b); Bassan et al. (2024)), we conceptualize each type of explanation as an *explainability query*. An explainability query takes both $f$ and $\mathbf{x}$ as inputs and aims to address specific inquiries while providing some type of interpretation for the prediction $f(\mathbf{x})$. Typically, explainability queries provide answers to decision problems or to counting problems.

**Sufficient reason Feature Selection.** We consider the common *sufficiency* criterion for feature selection, which is based on common explainability methods (Ribeiro et al. (2018); Carter et al. (2019); Ignatiev et al. (2019b)). A *sufficient reason* is a subset of input features, $S \subseteq [n]$, such that when we fix the features of $S$ to their corresponding values in $\mathbf{x} \in \mathbb{F}$, then the prediction always remains $f(\mathbf{x})$, regardless of any different assignment to the features in the subset $\overline{S}$. We use the notation of $(\mathbf{x}_S; \mathbf{z}_{\bar{S}})$ to denote an assignment where the values $\mathbf{x}$ are assigned to $S$ and the values of $\mathbf{z}$ are assigned to $\overline{S}$. We can hence formally define $S$ to be a sufficient reason with respect to $\langle f, \mathbf{x} \rangle$ iff it holds that for all $\mathbf{z} \in \mathbb{F}$: $f(\mathbf{x}_S; \mathbf{z}_{\bar{S}}) = f(\mathbf{x})$.

A typical assumption that is made in the literature suggests that smaller sufficient reasons (that is, those with a lesser cardinality of $|S|$) are more useful than larger ones (Ribeiro et al. (2018); Carter et al. (2019); Ignatiev et al. (2019b)). This leads to a particular interest in obtaining *cardinally minimal sufficient reasons*, also referred to as *minimum sufficient reasons*, and consequently to our first explainability query:

---

**MSR (Minimum Sufficient Reason)**:
**Input**: Model $f$, input $\mathbf{x}$, and $d \in \mathbb{N}$
**Output**: *Yes* if there exists some $S \subseteq [n]$ such that $S$ is a sufficient reason with respect to $\langle f, \mathbf{x} \rangle$ and $|S| \leq d$, and *No* otherwise.

---

To provide a comprehensive understanding of the complexity results of sufficient reasons, we study two additional common explainability queries (Barceló et al. (2020); Bassan et al. (2024)) which represent refinements of the MSR query: (i) The *Check-Sufficient-Reason* (*CSR*) query, which given a subset $S$ validates if it is a sufficient reason; (ii) The *Count-Completions* (*CC*) query which represents a generalized version of the CSR query, where given a subset of features, we return the relative portion of assignments that maintain a prediction. This form of explanation relates to the probability of maintaining a classification. Due to space constraints, we relegate the full formalization of the CSR and CC queries to Appendix C.

**Contrastive Explanations.** An alternative approach to interpreting models involves examining subsets of features which, when modified, could lead to a change in the model's classification (Dhurandhar et al. (2018); Guidotti (2022)). We consider a subset $S \subseteq [n]$ as *contrastive* if changing its values has the potential to alter the original classification $f(\mathbf{x})$; or, more formally, if there exists some $\mathbf{z} \in \mathbb{F}$ for which $f(\mathbf{x}_{\bar{S}}; \mathbf{z}_S) \neq f(\mathbf{x})$. Similarly to sufficient reasons, smaller contrastive reasons are generally assumed to be more meaningful. These represent the minimum change required for changing the original prediction. Hence, it is also natural to focus on *cardinally-minimal contrastive reasons*.

---

**MCR (Minimum Change Required)**:
**Input**: Model $f$, input $\mathbf{x}$, and $d \in \mathbb{N}$.
**Output**: *Yes*, if there exists some contrastive reason $S$ such that $|S| \leq d$ for $f(\mathbf{x})$, and *No* otherwise.

---

**Shapley Values.** In the additive feature attribution setting, each feature $i \in [n]$ is assigned an importance weight $\phi_i$. A common method for allocating weights is by using the *Shapley value* attribution index (Lundberg & Lee (2017)), defined as follows:

$$\phi_i(f, \mathbf{x}) := \sum_{S \subseteq [n] \setminus \{i\}} \frac{|S|!(n - |S| - 1)!}{n!} (v(S \cup \{i\}) - v(S)) \tag{1}$$

where $v(S)$ is the *value function*, and we use the common *conditional expectation* value function $v(S) := \mathbb{E}_{\mathbf{z} \sim \mathcal{D}_p}[f(\mathbf{z})|\mathbf{z}_S = \mathbf{x}_S]$ (Sundararajan & Najmi (2020); Lundberg & Lee (2017)).

In our complexity analysis, we assume feature independence, which follows common practice in computational complexity frameworks (Arenas et al. (2023); Van den Broeck et al. (2022)), as well as in practical methods for computing Shapley values, such as the KernelSHAP approach in the SHAP library (Lundberg & Lee (2017)). For a complete formalization, refer to Appendix C.

---

**SHAP (Shapley Additive Explanation)**:
**Input**: Model $f$, input $\mathbf{x}$, and $i \in [n]$
**Output**: The shapley value $\phi_i(f, \mathbf{x})$.

---

## 4    ENSEMBLE MODELS ARE LESS INTERPRETABLE

We seek to compare the complexity of solving an explainability query $Q$ for an ensemble, compared to solving it for the ensemble's constituent base models. We are particularly interested in cases where there exists *strict computational complexity gaps* between the two settings (e.g., solving $Q$ for a single base model can be performed in polynomial time, whereas solving it for the ensemble is NP-Complete). To identify these gaps, we use the notion of *c-interpretability* (computational interpretability) as defined in Barceló et al. (2020).:

**Definition 1.** *Let $\mathcal{C}_1$ and $\mathcal{C}_2$ be two classes of models and let $Q$ be an explainability query for which $Q(\mathcal{C}_1)$ is in complexity class $\mathcal{K}_1$ and $Q(\mathcal{C}_2)$ is in complexity class $\mathcal{K}_2$. We say that $\mathcal{C}_1$ is strictly more c-interpretable than $\mathcal{C}_2$ with respect to $Q$ iff $Q(\mathcal{C}_2)$ is hard for the complexity class $\mathcal{K}_2$ and $\mathcal{K}_1 \subsetneq \mathcal{K}_2$.*

### 4.1    COMPLEXITY GAP IN SIMPLE BASE MODELS

We begin with examining two types of base-models known for their simplicity and interpretability — decision trees and linear models. Our results affirm the existence of a complexity gap when an ensemble consists of these base-models, as depicted in Table 1. Our initial complexity results are presented in the following proposition, with the proof provided in Appendix (**?**):

Table 1: The "complexity gap" when interpreting a single model and an ensemble of models in the case of decision trees and linear models. Cells highlighted in blue represent novel results, presented here; whereas the rest were already known previously.

|  | Decision Trees | | Linear Models | |
| --- | --- | --- | --- | --- |
|  | Base-Model | Ensemble | Base-Model | Ensemble |
| **Check Sufficient Reason (CSR)** | PTIME | coNP-C | PTIME | coNP-C |
| **Minimum Contrastive Reason (MCR)** | PTIME | NP-C | PTIME | NP-C |
| **Minimum Sufficient Reason (MSR)** | NP-C | $\Sigma_2^P$-C | PTIME | $\Sigma_2^P$-C |
| **Count Completions (CC)** | PTIME | #P-C | #P-C | #P-C |
| **Shapley Additive Explanations (SHAP)** | PTIME | #P-H | #P-C | #P-H |

**Proposition 1.** *Ensembles of decision trees and ensembles of linear models are (i) coNP-Complete with respect to CSR, (ii) NP-Complete with respect to MCR, (iii) $\Sigma_2^P$-Complete with respect to MSR, (iv) #P-Complete with respect to CC, and (v) #P-Hard with respect to SHAP.*

Based on these findings, we leverage our previously established notation to deduce an interpretability separation between ensembles of decision trees/linear models and their corresponding base models:

**Theorem 1.** *Decision trees are strictly more c-interpretable than ensembles of decision trees with respect to CSR, MSR, MCR, CC, and SHAP. The same result holds for linear models (and ensembles of linear models) with respect to CSR, MSR, and MCR.*

In the previous theorem, there is no complexity gap for the CC and SHAP queries in the context of linear models. Nevertheless, it is still possible to demonstrate a complexity separation if we assume

that the weights and biases are given in unary form, a concept often termed as *pseudo-polynomial time*. This is demonstrated in the following proposition, with its proof provided in Appendix F.

**Proposition 2.** *While CC and SHAP can be solved in pseudo-polynomial time for linear models, ensembles of linear models remain #P-Hard even if the weights and biases are given in unary. Therefore, assuming that the weights and biases are given in unary, linear models are strictly more c-interpretable than ensembles of linear models with respect to CC and SHAP.*

## 4.2   NO COMPLEXITY GAP IN COMPLEX BASE MODELS

We revealed a complexity gap between interpreting simple models (decision trees, linear models) and ensembles comprised thereof. This raises the question of whether a similar complexity gap is observed with more complex models, such as neural networks. Our findings indicate that this disparity does not apply to neural networks. In fact, we establish a much stronger claim, confirming this not only for the explainability queries we examined, but in fact for *any* explainability query whose complexity class remains consistent under polynomial reductions. This includes all $\mathcal{K}$-Complete explainability queries, where $\mathcal{K}$ belongs to the polynomial hierarchy (classes such as PTIME, NP, $\Sigma_2^P$, etc.), or their associated counting classes (#P, etc.).

**Proposition 3.** *There is no explainability query Q for which the class of MLPs is strictly more c-interpretable than the class of ensemble-MLPs.*

The proof appears in Appendix G. The reasoning behind this outcome is straightforward — given that an ensemble of MLPs can be reduced to a single MLP in polynomial time, it follows that there do not exist two distinct complexity classes (that are closed under polynomial reductions) that differ in complexity when it comes to interpreting a single model versus an ensemble of such models. To extend this concept to models beyond MLPs, we define this characteristic and label it as: *closed under ensemble construction*.

**Definition 2.** *We say that a class of models $\mathcal{C}$ is* closed under ensemble construction *if given an ensemble $f$ containing models from $\mathcal{C}$, we can construct in polynomial time a model $g \in \mathcal{C}$ for which $\forall \boldsymbol{x} \in \mathbb{F}, f(\boldsymbol{x}) = g(\boldsymbol{x})$.*

Clearly, the aforementioned property correlates also to the expressiveness of the corresponding base-model, and does not hold for linear models and decision trees (assuming that P≠NP).

## 5   IMPACT OF BASE-MODEL COUNT AND SIZE ON ENSEMBLE INTERPRETABILITY

Up until now, our analysis did not consider any problem-specific paramaters. For example, the MCR query is polynomial-time solvable for a single decision tree, but becomes NP-Complete for an ensemble of $k$ models. This raises questions about how different parameters, such as the size of the participating base-models and the number of base models, contribute to this effect. We explore these aspects by using *parameterized complexity* (Downey & Fellows (2012)), an important branch of computational complexity theory, to study how different parameters affect the entire complexity of interpreting ensembles. We begin by outlining some fundamental concepts in parameterized complexity, which are crucial for this study. In this field, we deal with two-dimensional instances denoted as $\langle \mathcal{X}, k \rangle$ where $\mathcal{X}$ represents the original encoding, and $k$ is a specified parameter. We briefly describe the three main scenarios in parameterized complexity.

**1. Problems solvable in $|\mathcal{X}|^{O(1)} \cdot g(k)$ time.** This is the best-case scenario concerning the parameter $k$, and includes problems that are *fixed parameter tractable* (*FPT*) concerning $k$. This implies that the problem can be solved in $|\mathcal{X}|^{O(1)} \cdot g(k)$ time for some computable function $g$, or intuitively — that the complexity can be *primarily controlled by the parameter $k$*. For instance, the Vertex-Cover problem is FPT when $k$ is the vertex cover size. This means that if the size of the vertex cover is limited, the problem can be solved efficiently, even for arbitrarily large graphs.

**2. Problems solvable in $O(|\mathcal{X}|^{g(k)})$ time.** This class encompasses problems within the complexity class XP, solvable in $O(|\mathcal{X}|^{g(k)})$ time. When $k$ is fixed, these problems can be solved in polynomial time. However, unlike in FPT, if $|\mathcal{X}|$ becomes very large, these problems may remain challenging even for relatively small values of $k$. For a fine-grained analysis of complexity classes within XP,

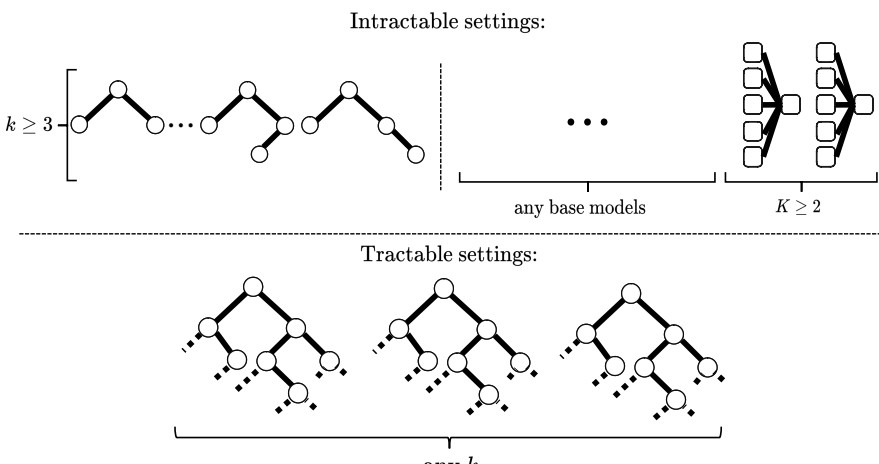

Figure 1: Illustration of some key aspects of our parameterized complexity results: For many scenarios, even highly simplified ensembles with constant-size base models ($k \geq 3$) remain intractable to interpret. However, simplifying an ensemble by reducing the number of decision trees within it (for models like Random Forest and XGBoost) does make the computation of various explanations tractable. Finally, we demonstrate that in numerous cases, ensembles consisting of just $k \geq 2$ linear models already become intractable to interpret, underscoring the challenges of interpreting ensembles with linear models, even in significantly simplified settings.

researchers have explored the W-hierarchy (Downey & Fellows (2012)), all of which is subsumed within XP. This includes $W[t]$ for every $t \geq 1$. For instance, the *Clique* problem is $W[1]$-Complete. Here, $t$ indicates the depth of a Boolean circuit used for the problem's reduction (detailed formalization can be found in Appendix B), and W[P] involves arbitrary depth. A fundamental assumption in parameterized complexity is that FPT $\subsetneq$ W[1] $\subsetneq$ W[2] $\subsetneq$ ...W[t] $\subsetneq$ W[P] $\subsetneq$ XP (Downey & Fellows (2012)). Hence, while all problems in XP are solvable in polynomial time when $k$ is constant, they lack FPT algorithms, indicating inefficiency at solving even when they are W[1]-Hard if $|\mathcal{X}|$ is arbitrarily large.

**3. Problems that are NP-Hard when $k$ is constant.** At the edge of the parameterized spectrum lies the complexity class *para-NP*, signifying the highest sensitivity level of the variable $k$. A problem is para-NP-Hard concerning $k$ if it is NP-hard even when $k$ is constant. As long as P$\neq$NP, then XP $\subsetneq$ para-NP. An illustrative example is the Graph-Coloring problem, where the parameter $k$ represents the number of colors. This problem is NP-Hard for any $k \geq 3$. Intuitively, this means that, unlike FPT and XP, even if the number of colors is substantially small, the problem is intractable.

**Extensions of parametrized complexity.** Finally, we briefly mention extensions of paramterized complexity classes to either counting problems (Flum & Grohe (2004)) or to higher orders of the polynomial hierarchy (de Haan (2019)) (classes such as $\Sigma_2^P$). Specifically, we mention the $\#W$-hierarchy which is an extension of the W-hierarchy to counting problems. There, too it is widely believed that FPT $\subsetneq$ #W[1] $\subsetneq$ #W[2] $\subsetneq$ ...#W[t] $\subsetneq$ #W[P] $\subsetneq$ XP. Moreover, we mention the class XNP, which is an extension of XP to the second order of the polynomial hierarchy. It is widely believed that XNP $\subsetneq$ para-$\Sigma_2^P$ (de Haan & Szeider (2017)). Finally, the concept of para-NP extends to other classes (de Haan (2019)). A problem is para-$\mathcal{K}$-Hard if it is $\mathcal{K}$-Hard even when $k$ is constant. This includes para-coNP, para-$\Sigma_2^P$, para-#P etc.

## 5.1 IMPACT OF BASE-MODEL SIZES ON ENSEMBLE INTERPRETABILITY

We start by studying how the sizes of an ensemble's base models influence its interpretability. In this setting, we take the size of the largest base model in an ensemble as our parameter $k$. We prove that with this parameterization, *all* of the aforementioned explainability queries become intractable already for a *constant* value of $k$.

**Proposition 4.** *An ensemble consisting of either linear models, decision trees or neural networks, parameterized by the maximal base-model size is (i) para-coNP Complete with respect to CSR, (ii) para-NP-Complete with respect to MCR, (iii) para-$\Sigma_2^P$-Complete with respect to MSR, (iv) para-#P-Complete with respect to CC, and (v) para-#P-Hard with respect to SHAP.*

The proof of Proposition 4 appears in Appendix H and provides a *negative* outcome regarding the interpretability of ensembles. The proposition implies that the uninterpretability of ensembles is not a result of the sizes of the participating base models, but rather of the aggregation process itself, such as the majority vote in voting ensembles. This implies that even if we reduce the size of our corresponding models to a constant size, the ensemble still remains intractable to interpret, and hence uninterpretable from a complexity perspective. This result applies to all of our base-model types, ensembles, and explanation forms.

## 5.2 IMPACT OF NUMBER OF BASE-MODELS ON ENSEMBLE INTERPRETABILITY

A more intricate dynamic emerges when we take the *number of base models* that participate in the ensemble as our parameter $k$. Table 2 shows our parameterized complexity results under this setting. The results for MLPs are straightforward, since they are closed under ensemble construction. Decision trees and linear models however reveal an interesting trend — while linear models lose their tractability with a constant number of models in the ensemble, tree ensembles tend to remain XP-tractable, meaning they can be solved in polynomial time when the number of base models is fixed. The following subsections explore these findings in more detail.

Table 2: Parameterized complexity classes for explainability queries of ensemble models, parametrized by the number of models participating in the ensemble, $k$.

|  | **Decision Trees** | **Linear Models** | **Neural Networks** |
| --- | :---: | :---: | :---: |
| **Check Sufficient Reason** | coW[1]-C | para-coNP-C (**k=2**) | para-coNP-C (k=1) |
| **Minimum Contrastive Reason** | W[1]-H, in W[P] | para-NP-C (**k=2**) | para-NP-C (k=1) |
| **Minimum Sufficient Reason** | para-NP-H (k=1), in XNP | para-$\Sigma_2^P$-C (**k=5**) | para-$\Sigma_2^P$-C (k=1) |
| **Count Completions** | #W[1]-C | para-#P-C (k=1) | para-#P-C (k=1) |
| **Shapley Additive Explanations** | #W[1]-H, in XP | para-#P-C (k=1) | para-#P-C (k=1) |

## 5.3 THE NUMBER OF DECISION TREES IN AN ENSEMBLE

**1. Problems solvable in $O(|\mathcal{X}|^k)$ time.** We begin by outlining the complexity results for decision tree ensembles, demonstrating that four of the five queries analyzed (CSR, MCR, CC, and SHAP) fall within XP, with some also belonging to lower complexity classes in the W-hierarchy.

**Proposition 5.** *For ensembles of $k$-decision trees, (i) the CSR query is coW[1]-Complete; (ii) the MCR query is W[1]-Hard and in W[P]; (iii) the CC query is #W[1]-Complete; and (iv) the SHAP query is #W[1]-Hard and in XP.*

Since all these queries fall within XP, they can be solved in $O(|\mathcal{X}|^{g(k)})$ time. However, given that all previously mentioned queries are W[1]-hard or coW[1]-hard, it is generally accepted that there are no FPT algorithms capable of solving them (Downey & Fellows (2012)). This leads us to the understanding that if the number of base models $k$ is constant, there exist *polynomial algorithms* that solve the CSR, MCR, CC, and SHAP queries for $k$ ensembles of decision trees. However, assuming that FPT $\subsetneq$ W[1], there are no algorithms that run in $|\mathcal{X}|^{O(1)} \cdot g(k)$ time that solve these queries.

Although these queries cannot be solved in FPT time under general conditions, if we assume that the number of leaf nodes $m$ in each tree is bounded by a constant (even if the total size $|f_i|$ of each tree is arbitrarily large), it is possible to develop FPT algorithms to solve these queries. This follows directly from the fact that all the aforementioned algorithms have a runtime of $O(m^k)$. This leads to the following conclusion: if the maximal number of leaves in each tree $m$ is constant, there exist FPT algorithms that solve the CSR, MCR, CC, and SHAP queries for $k$-ensembles of decision trees (even if the size $|f_i|$ of each base-model, and hence the size of the ensemble $|f|$, is arbitrarily large).

**2. Problems solvable in $O(|\mathcal{X}|^k)$ non-deterministic time.** The only explainability query that is not in XP for tree ensembles is MSR. This makes sense since this problem is NP-Hard only for a *single* decision tree (Barceló et al. (2020)), and hence it is para-NP-Hard for an ensemble of $k$ trees. However, we can show a similar behavior, by proving its membership in XNP, the variant of XP for the second order of the polynomial hierarchy (Flum & Grohe (2004)). This is illustrated in the following proposition, with its proof provided in Appendix J.

**Proposition 6.** *The MSR query for a $k$-ensemble of decision trees is para-NP-Hard and in XNP.*

Although the MSR query is not in XP, it is possible to show that a *relaxed* version of this query is. In this version, we obtain a *subset-minimal* (or "locally minimal") rather than a cardinally minimal sufficient reason. In other words, obtaining a subset $S \subseteq [n]$ which is a sufficient reason, such that for any $i$, $S \setminus \{i\}$ is not a sufficient reason (refer to the proof in Appendix K):

5.4    THE NUMBER OF LINEAR MODELS IN AN ENSEMBLE

In contrast to tree ensembles, where explanations can be computed in polynomial time when $k$ is constant, ensembles of linear models become hard to interpret with only a *constant* number of base-models. We demonstrate that this property holds across all the complexity classes we analyzed, as stated in the following proposition, with the proof provided in (**?**):

**Proposition 7.** *$k$-ensembles of linear models are (i) para-coNP Complete with respect to CSR; (ii) para-NP-Complete with respect to MCR; (iii) para-$\Sigma_2^P$-Complete with respect to MSR; (iv) para-$\#P$-Complete with respect to CC, and (v) para-$\#P$-Hard with respect to SHAP.*

**Tree Ensembles vs. Linear Model Ensembles.** The previous subsections showed that decision tree and linear model ensembles exhibit very distinct behaviors when parameterized by the number of base models $k$, giving rise to the following corollary:

**Theorem 2.** *$k$-ensemble-decision trees are strictly less c-interpretable than $k$-ensemble-linear models with respect to CSR, MSR, MCR, CC, and SHAP.*

From a practical viewpoint, we can interpret a decision tree ensemble in polynomial time if we reduce the number of trees. On the other hand, interpreting ensembles of linear models proves to be intractable, even if we reduce the number of base-models up to a constant. We highlight that the hardness results apply also to heterogeneous ensembles. In other words, any ensemble that contains some mixture of models, and contains a constant number of linear models, is intractable to interpret. We further emphasize this polynomial vs. non-polynomial distinction with the following corollary:

**Theorem 3.** *If the number $k$ of base models is constant, there exist polynomial algorithms for solving the CSR, MCR, CC, and SHAP queries for $k$-ensembles of decision trees. However, assuming that $P \neq NP$, there are no polynomial algorithms for solving the CSR, MCR, CC, and SHAP queries for ensembles of linear models even when $k$ is constant.*

Although the MSR query is NP-hard to solve for both a constant number of trees and a constant number of linear models, it is still possible to demonstrate a distinction between the two. While NP-complete problems are known to be challenging for optimization, they can sometimes be addressed using various optimization tools such as Boolean satisfiability (SAT) or satisfiability modulo theory (SMT) solvers (Moskewicz et al. (2001); Barrett & Tinelli (2018)). In contrast, problems that are hard for the intractable class $\Sigma_2^P$ require a worst-case *exponential* number of calls to an NP oracle. Therefore, the difference between classifying a problem as NP or $\Sigma_2^P$-hard becomes crucial in this context. In our context, assuming the number of base models $k$ is constant, then solving the MSR query for ensembles of decision trees is in NP. However, for ensembles of linear models, this problem is $\Sigma_2^P$-Hard, which demonstrates an additional complexity gap.

Finally, we observe that the algorithms designed to solve CSR, MCR, CC, and SHAP for decision tree ensembles operate within $O(m^k)$ time (implying $O(|\mathcal{X}|^k)$ time), where $m$ represents the maximum number of leaves in any single tree of the ensemble. Thus, by limiting $m$ to a constant, these queries can be efficiently solved in FPT time for any $k$, even if the size of the tree is arbitrarily large. However, when the size of a linear model is arbitrarily large, these tasks become NP-hard (or coNP-hard) to solve, even if $k$ is constant. This particularly implies that when assuming that the size of each linear model $|f_i|$ in an ensemble $f$ is arbitrarily large, the CSR, MCR, CC, and SHAP queries are NP (or

coNP)-Hard, even when $k$ is constant. However, this is not the case for enembles of decision trees, which can be interpreted in FPT time even for an unbounded $k$ and for arbitrarily large $|f_i|$ models.

## 6 RELATED WORK

Our work relates to the field of formal XAI, which focuses on explanations with mathematical guarantees (Marques-Silva et al. (2020)). Previous studies have discussed the computational complexity associated with obtaining such explanations for various ML models (Barceló et al. (2020); Wäldchen et al. (2021); Arenas et al. (2022)). In this paper, our primary focus was on understanding the complexity of *ensemble* models and their impact on model interpretability. Some preliminary work has yielded results on the complexity of generating certain types of interpretations for specific ensembles (e.g., random forest classifiers) (Izza et al. (2021); Audemard et al. (2022; 2023)). Other studies have investigated the use of parameterized complexity in the context of model interpretability. This approach has successfully highlighted distinctions between shallow and deep neural networks (Barceló et al. (2020)) and enabled parameterization based on the size of explanations (Ordyniak et al. (2023)). In more recent work, Ordyniak et al. (2024) analyzes the complexity of various logic-based reasoning queries (primarily local/global abductive and contrastive reasoning queries), parameterized by different problem parameters, including those related to ensembles. A comprehensive review of all prior computational complexity results relevant to this study is provided in the appendix.

Another line of work focuses on obtaining explanations with formal guarantees on tree ensembles by encoding them as propositional logic formulas and then solving these queries with Boolean satisfiability (SAT) (Izza & Marques-Silva), Maximum satisfiability (MaxSAT) (Ignatiev et al. (2022)), Mixed integer linear programming (MILP) (Parmentier & Vidal (2021); Chen et al. (2019)) or satisfiability modulo (SMT) (Ignatiev et al. (2019c)) solvers.

In our work, we used certain terms that have sometimes been discussed in the literature under different names. For instance, "sufficient reasons" have also been referred to as abductive explanations (Ignatiev et al. (2019a)). Moreover, subset minimal sufficient reasons are related (though not exactly equivalent to) prime implicants in the case of Boolean formulas (Darwiche & Marquis (2002)). Finally, a term akin to the CC query can be found in the literature as the $\delta$-relevant set (Wäldchen et al. (2021); Izza et al. (2021)). This term pertains to determining whether the completion count surpasses a predefined threshold $\delta$.

## 7 LIMITATIONS AND FUTURE WORK

Similarly to other studies on the computational complexity of obtaining explanations, our work is limited to specific explanation forms and base-model types. Nevertheless, we believe our work still offers a comprehensive overview of a variety of widely used explanation formats and settings, paving the way for future exploration of the complexity involved in obtaining additional forms. Furthermore, while most of our findings can be extended from classification to regression settings, a few specific results require further investigation to fully address the regression case. These considerations, along with other potential extensions and open problems related to our work, are discussed in Appendix D.

## 8 CONCLUSION

We present a theoretical framework based on complexity theory to assess the interpretability of ensemble models. Our work provides mathematical evidence for the folklore belief: "*ensembles are not interpretable*". However, we prove that a strict difference between base models and ensembles exists only in simple base models, such as linear models and decision trees, but not in complex base models, such as neural networks. We then propose a parameterized complexity view of these results, and derive some unexpected conclusions. Intuitively, we show that reducing the size of the base models within an ensemble cannot make it interpretable. However, limiting the number of base models can. This is the case for tree ensembles, which are efficiently interpretable with a reduced number of trees, but not for linear model ensembles, which are hard to interpret with even a constant number of base-models. We believe that these results provide novel insights into ensemble interpretability, and highlight the importance of considering computational complexity aspects for enriching our understanding of ML interpretability.

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

# Appendix

The appendix contains formalizations and proofs that were mentioned throughout the paper:

**Appendix A** contains formalizations of base-model types and ensembles.
**Appendix B** contains background on parameterized complexity.
**Appendix C** contains additional formalizations concerning explainability queries.
**Appendix D** contains information regarding possible expansions of our framework to (i) discrete and continuous input/output domains, (ii) regression tasks, and (iii) heterogeneous ensembles.
**Appendix E** contains the proof of Proposition 1.
**Appendix F** contains the proof of Proposition 2.
**Appendix G** contains the proof of Proposition 3.
**Appendix H** contains the proof of Proposition 4.
**Appendix I** contains the proof of Proposition 5.
**Appendix J** contains the proof of Proposition 6.
**Appendix K** contains the proof of Proposition 7.
**Appendix L** contains the proof of Proposition 8.

## A ENSEMBLES AND BASE-MODEL TYPES

In this appendix, we describe the various ensembles that are incorporated in our work and the corresponding base-model types that consist in these models.

### A.1 ENSEMBLE FORMALIZATION

Numerous well-known ensemble techniques exist; however, our research is geared towards *post-hoc* interpretation, thus we emphasize the *inference* phase rather than the training of these ensembles. Our analysis is focused on ensemble families that utilize either *majority voting* or *weighted-voting* methods during inference. This includes *bagging* ensembles such as random forests, which implement majority-voting-based inference, and *boosting* ensembles like XGBoost, Gradient Boosting, and Adaboost, which employ weighted voting for inference. Moreover, we will examine how all of our results hold to either *hard-voting* or *soft-voting* inference as well as the potential to apply our findings to alternative inference techniques such as *weighted averaging*, or *meta-model decision* inference. Such inference techniques are commonly found in other ensemble types like *stacking* ensembles, or those utilized in regression tasks.

**Majority Voting Inference.**

In *majority voting* inference, the condition $f(\mathbf{x}) = 1$ is satisfied if and only if there are at least $\lceil \frac{k}{2} \rceil$ base-models within an ensemble $f$ where $f_i(\mathbf{x}) = 1$. Put simply, the decision for $f(\mathbf{x})$ is determined by the majority consensus of the models involved in $f$. Formally, for any $\mathbf{x} \in \mathbb{F}$ we define $f$ as follows:

$$f(\mathbf{x}) := \begin{cases} 1 & if \ |\{ \ i \ | \ f_i(\mathbf{x}) = 1 \ \}| \ \geq \ \lceil \frac{k}{2} \rceil \\ 0 & otherwise \end{cases} \tag{2}$$

**Weighted Voting Inference.**

For *weighted voting* inference, we consider a weight $\phi_i \in \mathbb{Q}$ that describes the importance of each model participating in the ensemble. Considering this is a binary classification, we can define the prediction as being determined by the sign of the total aggregation of all weights. Formally, for any $\mathbf{x} \in \mathbb{F}$ we define $f$ as follows:

$$f(\mathbf{x}) := step(\sum_{1 \leq i \leq n} \phi_i \cdot f_i(\mathbf{x})) \tag{3}$$

where $step(\mathbf{z}) = 1 \iff \mathbf{z} \geq 0$.

**Weighted voting is harder than majority voting**: When examining the explainability query $Q$, our analysis aims to evaluate the complexity classes of two families of ensembles: majority voting ensembles, denoted as $\mathcal{C}_\mathcal{M}$, and weighted voting ensembles, denoted as $\mathcal{C}_\mathcal{W}$. It is straightforward to demonstrate that the following relationship is true:

**Lemma 1.** *Let $\mathcal{C}$ denote a class of models, $\mathcal{C}_\mathcal{M}$ the class of majority voting ensembles of models from $\mathcal{C}$, and $\mathcal{C}_\mathcal{W}$, the class of weighted voting ensembles of models from $\mathcal{C}$. Then for any explainability query $Q$ it holds that $Q(\mathcal{C}_\mathcal{M}) \leq_P Q(\mathcal{C}_\mathcal{W})$.*

*Proof.* The lemma holds by a simple reduction that starts with a majority voting ensemble $f$ and constructs a weighted voting ensemble $f'$ where each weight is assigned an equal attribution. In other words: $\phi'_i = \frac{1}{n}$ for all $i \in [n]$.

$\square$

The previous statement holds technical significance as it simplifies the process of establishing complexity class completeness results for both $\mathcal{C}_\mathcal{M}$ and $\mathcal{C}_\mathcal{W}$:

**Observation 1.** *For proving that the complexity of solving both $Q(\mathcal{C}_\mathcal{M})$ and $Q(\mathcal{C}_\mathcal{W})$ are complete for some complexity class $\mathcal{K}$ (closed under polynomial reductions), it suffices to prove membership for $Q(\mathcal{C}_\mathcal{W})$ and hardness for $Q(\mathcal{C}_\mathcal{M})$.*

From this point forward, whenever we mention the computational complexity of an ensemble of models in our text, it applies to both majority voting ensembles and weighted voting ensembles, as we prove the completeness of complexity classes for both families. We highlight this differentiation in our proofs, which are applicable to both types of ensembles.

**Extension to *Soft* Voting.** In contrast to *hard* voting that is common in the binary classification setting, within probabilistic classification soft voting can also be implemented. In this case, each model $f_i$ in the ensemble outputs some given probability value i.e, $f_i : \mathbb{F} \to [0, 1]$. Then, in the case of *majority soft voting*, the inference is computed by:

$$f(\mathbf{x}) := step(\sum_{1 \leq i \leq n} \frac{f_i(\mathbf{x})}{n}) \tag{4}$$

whereas in *weighted soft voting* the inference is computed by incorporating equation 3 for each $f_i : \mathbb{F} \to [0, 1]$.

We start by defining a property that will be used in the next lemma. We say that a class $\mathcal{C}$ of models $f : \mathbb{F} \to [0, 1]$ is *scalar multiplicative* if given some constant $\lambda \in \mathbb{R}$ and for all $f \in \mathcal{C}$ we can construct, in polynomial time a model $f' \in \mathcal{C}$ for which $f' = \lambda f$.

**Lemma 2.** *Let $\mathcal{C}$ denote a class of models, $\mathcal{C}_\mathcal{W}$ the class of weighted (hard) voting ensembles of models from $\mathcal{C}$, $\mathcal{C}_{\mathcal{SW}}$, the class of (soft) weighted voting ensembles of models from $\mathcal{C}$, and $\mathcal{C}_{\mathcal{SM}}$, the class of (soft) majority voting ensembles of models from $\mathcal{C}$. Then for any explainability query $Q$ it holds that $Q(\mathcal{C}_\mathcal{W}) =_P Q(\mathcal{C}_{\mathcal{SW}}) =_P Q(\mathcal{C}_{\mathcal{SM}})$. The only restriction is for the condition $Q(\mathcal{C}_\mathcal{W}) \geq_P Q(\mathcal{C}_{\mathcal{SM}})$ and it is that $\mathcal{C}_\mathcal{W}$ is scalar multiplicative.*

*Proof.* Given a soft majority voting ensemble $f$, we can construct a weighted hard (voting) ensemble $f'$ for which each weight $\phi'_i(\mathbf{x}) := \frac{f_i(\mathbf{x})}{n}$. For the other direction, given a weighted hard (voting) ensemble $f$, and assuming that $\mathcal{C}_\mathcal{W}$ is scalar multiplicative, we can construct a soft majority voting ensemble $f'$. We do this by, for every $i \in [n]$, constructing $f'_i(\mathbf{x}) = \phi_i \cdot f_i(\mathbf{x})$ (since $\mathcal{C}_\mathcal{W}$ being scalar multiplicative). Overall, from these two reductions, we get that $Q(\mathcal{C}_\mathcal{W}) =_P Q(\mathcal{C}_{\mathcal{SM}})$.

For the second part of the claim — we start with a weighted hard voting ensemble $f$ and construct a weighted soft voting ensemble $f'$ by assigning each weight $\phi'_i(\mathbf{x}) := \frac{\phi_i(\mathbf{x})}{n}$. For the other direction, given a weighted soft voting ensemble $f$ we construct a weighted hard voting ensemble $f'$ by setting $\phi'_i(\mathbf{x}) := \phi_i(\mathbf{x}) \cdot f_i(\mathbf{x})$. Overall, this implies that $Q(\mathcal{C}_\mathcal{W}) =_P Q(\mathcal{C}_{\mathcal{SW}})$.

$\square$

Lemma 2 establishes that our proofs (including those for membership and hardness) apply to *soft-voting* ensembles, both for majority and weighted voting scenarios. This is because our proofs are

conducted within the framework of *weighted* (hard voting ensembles), and all the complexity classes we consider are closed under polynomial reductions.

**(Weighted) Averaging.** In the *regression* setting, another common inference technique involves a *weighted averaging* of all base-model predictions. Formally, given a set of $k$ regression base-models $f_i : \mathbb{F} \to \mathbb{R}$, then we define $f$ by incorporating equation 4 over each $f_i$. In essence, this is the same formalization as that of majority soft voting, and hence Lemma 2 holds for this family of inference models as well. However, when shifting our focus to regression, we must also consider different formalizations of some of the query forms discussed in our paper, such as the definition of a sufficient reason. We discuss these specific adjustments for the regression setting under Appendix D.

**Meta learner decision.** Another common ensemble inference method often used in *stacking* ensembles involves employing a *meta-model* to aggregate the outputs of the $k$ base models. In our particular scenario, this involves a model $g : \{0,1\}^k \to \{0,1\}$, which is trained to classify the outputs from each base model $f_i$ within a specified domain. It is important to note that if $g$ can function as a majority voting system among the $k$ models — a capability all analyzed model types possess, including MLPs, Perceptrons, FBDDs— then all the *hardness* findings discussed in this paper automatically apply to this setup as well. For instance, a *stacking* ensemble comprising a constant number of linear base-models remains intractable to interpret, as demonstrated in Proposition 7. However, the examination of *membership* results that were presented in this paper will vary depending on the type of model used for the meta-model $g$.

## A.2 BASE-MODEL TYPES

In this subsection, we formalize the three base-model types that were analyzed throughout the paper: (i) FBDDs, (ii) Perceptrons, and (iii) MLPs .

**Free Binary Decision Diagram (FBDD).** (Lee (1959)) A *binary decision diagram* (*BDD*) is an acyclic-directed graph and serves as a graphical model for a Boolean function $f : \mathbb{F} \to \{0,1\}$. This graph embodies the given Boolean function in the following manner: (i) Each internal node $v$ is associated with a unique binary input feature from the set $\{1, \ldots, n\}$; (ii) Every internal node $v$ has precisely two outgoing edges, corresponding to the values $\{0,1\}$ which are assigned to $v$; (iii) In the BDD, each variable is encountered no more than once on any given path $\alpha$; (iv) Each leaf node is labeled either *True* or *False*.

Thus, assigning a value to the inputs $\mathbf{x} \in \mathbb{F}$ uniquely determines a specific path $\alpha$ from the root to a leaf in the BDD. The function $f(\mathbf{x})$ is assigned a 1 if the terminal node leaf is labeled *True*, and 0 if it is labeled *False*. The size of the BDD, denoted as $|f|$, is measured by the total number of edges in its graph. Our study focuses on the widely used *Free Binary Decision Diagrams* (*FBDDs*), which permit different varying orderings of the input variables $\{1, \ldots, n\}$ across any two distinct paths, $\alpha$ and $\alpha'$. This ensures that no two nodes along any single path $\alpha$ share the same label. Essentially, a *decision tree* is a type of FBDD with a fundamental graph structure that is a tree.

**Multi-Layer Perceptron (MLP).** (Gardner & Dorling (1998); Ramchoun et al. (2016)) An MLP, denoted by $f$, consists of $t-1$ *hidden layers* ($g^j$ where $j$ ranges from 1 to $t-1$) and a single output layer ($g^t$). The layers are defined recursively — each layer $g^{(j)}$ is computed by applying the activation function $\sigma^{(j)}$ to the linear combination of the outputs from the previous layer $g^{(j-1)}$, the corresponding weight matrix $W^{(j)}$, and the bias vector $b^{(j)}$. This is represented as $g^{(j)} := \sigma^{(j)}(g^{(j-1)}W^{(j)} + b^{(j)})$ for each $j$ in $\{1, \ldots, t\}$. The model includes $t$ weight matrices $(W^{(1)}, \ldots, W^{(t)})$, $t$ bias vectors $(b^{(1)}, \ldots, b^{(t)})$, and $t$ activation functions $(\sigma^{(1)}, \ldots, \sigma^{(t)})$.

In the described MLP, the function $f$ is defined to output $f := g^{(t)}$. The initial input layer $g^{(0)}$ is denoted by $\mathbf{x} \in \{0,1\}^n$, which serves as the model's input. The dimensions of the biases and weight matrices are specified by the sequence of positive integers $\{d_0, \ldots, d_t\}$. We specifically focus on weights and biases that are rational numbers, represented as $W^{(j)} \in \mathbb{Q}^{d_{j-1} \times d_j}$ and $b^{(j)} \in \mathbb{Q}^{d_j}$, which are parameters that are optimized during training. Given that the model is a binary classifier for indices $\{1, \ldots, n\}$, it follows that $d_0 = n$ and $d_t = 1$. The primary activation function $\sigma^{(i)}$ that we focus on is the *ReLU* activation function, defined as $reLU(x) = \max(0, x)$, except for the output layer, where a *sigmoid* function is typically used for the classification. Since our focus is only on the

*post-hoc* interpretation of the corresponding model, we will equivalently assume the existence of a step function for the final layer activation, where we denote $step(\mathbf{z}) = 1 \iff \mathbf{z} \geq 0$.

**Perceptron.** (Ralston et al. (2003)) A perceptron represents a single-layer MLP or in other words $t = 1$. It is defined by the function $f(\mathbf{x}) = step((\mathbf{w} \cdot \mathbf{x}) + b)$ with $b$ belonging to the set of rational numbers, and $\mathbf{w}$ being a matrix in $\mathbb{Q}^{n \times d_1}$. Consequently, for the perceptron function $f$ it holds without loss of generality that $f(\mathbf{x}) = 1$ if and only if $(\mathbf{w} \cdot \mathbf{x}) + b \geq 0$.

# B  PARAMETERIZED COMPLEXITY BACKGROUND

In parameterized complexity, we deal with *parameterized problems* $L \subseteq \Sigma^* \times \mathbb{N}$ where $\Sigma$ is some finite alphabet. The elements of the paramaterized problems are hence two-dimensional instances denoted as $\langle \mathcal{X}, k \rangle$ where $\mathcal{X}$ represents the original encoding and $k$ is the *parameter*.

## B.1  PARAMETERIZED REDUCTIONS

**FPT Reductions.** The parameterized complexity classes that we will discuss here are closed under a specific kind of reductions, known as fixed-parameter tractable (FPT) reductions. A given mapping $\phi : \Sigma^* \times \mathbb{N} \to \Sigma^* \times \mathbb{N}$ between instances from a parameterized problem $P_1$ to another parameterized problem $P_2$ is an *FPT reduction* iff it holds that: (i) $(\mathcal{X}, k)$ is in $P_1$ if and only if $\phi(\mathcal{X}, k)$ is in $P_2$; (ii) there exists a computable function $g$ for which $k' \leq g(k)$ when $k'$ is the parameter of $\phi(\mathcal{X}, k)$; and (iii) $\phi(\mathcal{X}, k)$ can be computed in $|\mathcal{X}|^{O(1)} \cdot g'(k)$ time for some computable function $g'$.

**FPT Parsimonious Reductions.** For the counting version of FPT reductions (Flum & Grohe (2004)), given two paramterized counting problems $F : \Sigma^* \times \mathbb{N} \to \mathbb{N}$, and $G : \Sigma^* \times \mathbb{N} \to \mathbb{N}$ we define an FPT parsimonious reduction from $F$ to $G$ as an algorithm that computes for any instance $\langle \mathcal{X}, k \rangle$ of $F$ an instance $\langle \mathcal{Y}, \ell \rangle$ of $G$ in time $g_1(k) \cdot |\mathcal{X}|^c$ such that $\ell \leq g_2(k)$ and $F(\mathcal{X}, k) = G(\mathcal{Y}, \ell)$, for some computable functions $g_1, g_2 : \mathbb{N} \to \mathbb{N}$, and a constant $c \in \mathbb{N}$.

## B.2  PARAMETERIZED COMPLEXITY CLASSES.

We now will formalize the parameterized complexity classes that are relevant for this work.

**FPT.** A problem is *fixed parameter tractable* (*FPT*) concerning $k$ iff there exists a $|\mathcal{X}|^{O(1)} \cdot g(k)$ time algorithm solving the problem for some computable function $g$.

**XP and the W-Hierarchy.** The class XP describes all problems that can be solved in $O(|\mathcal{X}|^{g(k)})$ time for some computable function $g$. This class additionally encompasses the $W$-hirerchy (Downey & Fellows (2012)), which can be described using boolean circuits. We recall that a boolean circuit $C$ is represented as a rooted directed acyclic graph. Nodes with no incoming edges are referred to as input gates, and the singular node without any outgoing edges is the output gate. The internal nodes of the circuit are designated as OR, AND, or NOT gates. NOT gates are characterized by having exactly one incoming edge. AND and OR gates can have up to two incoming edges, termed small gates, or more than two, termed large gates. The *depth* of a circuit is measured by the longest path of edges from any input node to the output node. The *weft* of a circuit refers to the largest number of large gates on any path from an input node to the output node. An assignment for $C$ maps the input gates to binary values {0,1}. The *hamming weight* of an assignment reflects the count of input gates assigned the value 1. This assignment determines the output at each gate based on its specific function. A circuit is satisfied by an assignment if it results in the output gate producing a value of 1.

We can now characterize the $W$-Hierarchy through reductions to the general *Weighted Circuit Satisfiability* problem (*WCS*), which is defined as follows:

---

**Weighted Circuit Satisfiability (WCS[$C_{t,d}$]):**
**Input**: A boolean Circuit $C$ with weft at most $t$ and depth at most $d$, and an integer $k$,
**Parameter**: $k$
**Output**: *Yes*, if $C$ has a satisfying assignment of Hamming weight exactly $k$

---

We then say that a problem $Q$ belongs to $W[t]$ if there is an FPT reduction from $Q$ to WCS$[C_{t,d}]$, for some fixed constant $d$. If there exists an FPT reduction from $Q$ to WCS$[C_{t,d}]$, where the constructed circuit $C$ is allowed to have an arbitrary weft $t$, then we say that $Q$ belongs to $W[P]$. It is widely believed that the following relation holds (Downey & Fellows (2012)):

$$\text{FPT} \subsetneq W[1] \subsetneq W[2] \subsetneq \ldots \subsetneq W[t] \subsetneq W[P] \subsetneq \text{XP} \tag{5}$$

**XNP.** Our paper will also briefly discuss the *XNP* complexity class (de Haan & Szeider (2017)), which is a generalization of the XP class to the second order of the polynomial hierarchy. XNP describes the set of problems that can be solved by a non-deterministic algorithm in $O(|\mathcal{X}|^{g(k)})$ time for some computable function $g$. It is widely believed that XNP $\subsetneq$ para-$\Sigma_2^P$.

**The #$W$-Hirerchy.** The $W$-Hierarchy can be extended to its equivalent counting based hiearchy, which is termed the #$W$-hierarchy (Flum & Grohe (2004)). Similarly to the $W$-herarchy, where we use the problem WCS$[C_{t,d}]$, we now denote #WCS$[C_{t,d}]$ as the problem of counting the number of assignments of Hamming weight $k$ for a Boolean circuit $C$ with depth $d$ and weft $t$ (that is not a fixed constant, but may depend on the size of $C$). Similarly to the $W$-hierarchy, we define #$W[t]$ as any problem $Q$ for which there exists an FPT parsimonious reduction from $Q$ to #WCS$[C_{t,d}]$ for some constant $t$. If $t$ is arbitrary, then $Q$ is in #$W[P]$. Similarly to the $W$-hierarchy, it here too is believed that:

$$\text{FPT} \subsetneq \#W[1] \subsetneq \#W[2] \subsetneq \ldots \subsetneq \#W[t] \subsetneq \#W[P] \subsetneq \text{XP} \tag{6}$$

**Para-NP.** A problem is in *para-NP* concerning some paramater $k$ if there exists a *non-deterministic* algorithm which solves the problem in $O(|\mathcal{X}|^{O(1)} \cdot g(k))$ time for some computable function $g$. A problem is *para-NP-Hard* if and only if the non-parameterized problem is NP-Hard when $k$ is set to some constant.

**Beyond Para-NP.** Other relevant classes are the extensions of the para-NP class to other classes, such as *para-coNP*, *para-$\Sigma_2^P$*, etc. (Flum & Grohe (2003)). Let $\mathcal{K}$ be a classical complexity class. Then para-$\mathcal{K}$ is the class of all parameterized complexity problems $P$, with $P \subseteq \Sigma^* \times \mathbb{N}$, for which there exists an alphabet $\Pi$, a computable function $g : \mathbb{N} \to \Pi^*$, and a problem $Q \subseteq \Sigma^* \times \Pi^*$ such that $\mathcal{X} \in \mathcal{K}$ and for all instances $(\mathcal{X}, k) \in \Sigma^* \times \mathbb{N}$ of $P$ we have:

$$(\mathcal{X}, k) \in P \iff (\mathcal{X}, g(k)) \in Q \tag{7}$$

Intuitively, the class para-$\mathcal{K}$ contains all problems that are in $\mathcal{K}$ after a *pre-computation* involving the parameter. Put differently, a problem is in para-$\mathcal{K}$ if it can be solved by two algorithms $\mathbb{P}$ and $\mathbb{A}$, where $\mathbb{P}$ is arbitrary and $\mathbb{A}$ has resources that are constrained by $\mathcal{K}$. The pre-computation that is performed by $\mathbb{P}$ involves only the paramater, which then transforms $k$ into a string $g(k)$. Then, the second algorithm $\mathbb{A}$ solves the problems given $g(k)$ from the pre-computation and the original input $x$, with resources that are constrained by the complexity class $\mathcal{K}$. We define a problem as *para-$\mathcal{K}$-Hard* if there exists an FPT-reduction from any problem in para-$\mathcal{K}$ to it. Alternatively, a problem is para-$\mathcal{K}$-Hard if it is $\mathcal{K}$-Hard even when $k$ is constant. (Flum & Grohe (2003); de Haan (2016)). It is widely believed that the following relations hold (Downey & Fellows (2012); Flum & Grohe (2004)):

$$\text{XP} \subsetneq \text{para-NP}, \quad \text{XNP} \subsetneq \text{para-}\Sigma_2^P \tag{8}$$

## C  Additional Query Formalizations

Here, we define the two remaining queries referenced throughout the paper: the *Check-Sufficient-Reason* (*CSR*) and *Count Completions* (*CC*) queries, which have also been analyzed in previous works (Barceló et al. (2020); Bassan et al. (2024)). We will then provide further details on the computational complexity analysis of computing Shapley values, as discussed in the paper.

The *CSR* query answers whether a specific subset $S$ is a sufficient reason. More formally:

---

**Check Sufficient Reason (CSR)**:
**Input**: A model $f$, an instance $\mathbf{x}$, and a subset $S$.
**Output**: *Yes*, if $S$ is a sufficient reason of $\langle f, \mathbf{x} \rangle$, and *No* otherwise.

---

For the *CC* query we consider a *relaxed* version of the *CSR* query which instead of validating whether a specific subset is sufficient or not, asks for the relative *portion* of assignments maintaining a given prediction, given that the other features are independently and uniformly distributed. We start by defining the *completion count* of a given subset:

$$c(S, f, \mathbf{x}) := \frac{|\{\mathbf{z} \in \{0,1\}^{|\overline{S}|}; f(\mathbf{x}_S; \mathbf{z}_{\overline{S}}) = f(\mathbf{x})\}|}{|\{\mathbf{z} \in \{0,1\}^{|\overline{S}|}|} \tag{9}$$

Now, the *CC* query is defined as follows:

---

**CC (Count Completions)**:
**Input**: Model $f$, input $\mathbf{x}$, and subset of features $S$.
**Output**: The completion count $c(S, f, \mathbf{x})$.

---

We provide here a more expanded formalization of the *shapley-value* that is incorporated in the paper. The *shapley value* attribution is:

$$\phi_i(f, \mathbf{x}) := \sum_{S \subseteq [n] \setminus \{i\}} \frac{|S|!(n - |S| - 1)!}{n!} (v(S \cup \{i\}) - v(S)) \tag{10}$$

where $v(S)$ is the *value function*, and we use the common *conditional expectation* value function $v(S) := \mathbb{E}_{\mathbf{z} \sim \mathcal{D}_p}[f(\mathbf{z})|\mathbf{z}_S = \mathbf{x}_S]$ (Sundararajan & Najmi (2020); Lundberg & Lee (2017)). We follow common conventions in frameworks that assessed the computational complexity of computing *exact* calculations of Shapley values (Arenas et al. (2023); Van den Broeck et al. (2022)), as well as practical frameworks that compute Shapley values, such as the kernelSHAP method in the SHAP libary (Lundberg & Lee (2017)), and assume that each input feature is independent of all other features, or in other words, every feature $i \in [n]$ is assigned some probability value $[0,1]$, i.e., $p : [n] \to [0,1]$. These are called *product distributions* in the work of Arenas et al. (2023) or *fully factorized* in the work of Van den Broeck et al. (2022). We then formally define our distributions $\mathcal{D}_p(\mathbf{x})$ as:

$$\mathcal{D}_p(\mathbf{x}) := \Big( \prod_{i \in [n]; \mathbf{x}_i = 1} p(i) \Big) \cdot \Big( \prod_{i \in [n], \mathbf{x}_i = 0} (1 - p(i)) \Big) \tag{11}$$

Clearly the uniform distribution is a special case of $\mathcal{D}_p$, obtained by setting $p(i) := \frac{1}{2}$ for every $i \in [n]$.

## D  FRAMEWORK EXTENSIONS

**Input and Output Domains.** To make our proofs cleaner and easier to understand, we followed common conventions (Barceló et al. (2020); Arenas et al. (2022); Wäldchen et al. (2021); Arenas et al. (2021a); Bassan et al. (2024)) and presented them using boolean input and output values. It should be emphasized that our analysis is not confined to binary input feature domains but is also applicable to features with $k$ possible *discrete* values, where $k$ is any integer. Furthermore, we can modify our method to include inputs incorporating *continuous* input domains. We will next provide a short overview of the various contexts in which this extension is applicable.

Regarding MLP explainability queries, earlier research indicates that the complexity of a *satisfiability* query on an MLP extends to scenarios involving continuous inputs. Specifically, the work of (Katz et al. (2017)) and (Sälzer & Lange (2021)) proves that verifying an arbitrary satisfiability query on an MLP with ReLU activations, over a continuous input domain, remains NP-complete. The *CSR* query

mentioned in this work, when $S := \emptyset$ is akin to *negating* a satisfiability query, and this implies that the *CSR* query in MLPs remains coNP complete for the continuous case as well. We recall that the complexity of the *MSR* query for MLPs is $\Sigma_2^P$-Complete (Barceló et al. (2020)). This complexity arises from the use of a coNP oracle, which determines whether a subset of features is sufficient, essentially addressing the *CSR* query. Given that *CSR* can also be adapted to handle continuous outputs, the logic applied to *CSR* can similarly be applied to demonstrate that the *MSR* query can be extended to continuous domains.

For Perceptrons, the completeness proofs remain valid in a continuous domain for the explainability queries. The continuity of inputs does not alter the membership proofs, for the same reasons that hold for MLPs. For hardness proofs, notice that all reductions that were derived from the subset sum (*SSP*) problem (or generalized subset sum (*GSSP*) problem), can be adjusted to substitute any call to $\max\{z_i, z_j\}$ in our original proof with $\max([z_i, z_j])$.

Finally, the proofs that apply to tree classifiers (or ensemble tree classifiers) for queries are equally valid for continuous inputs. This extension to continuous inputs was demonstrated by previous works (see for instance, (Huang et al. (2021))). This logic implies also to the complexity of ensembles consisting of the aggregation of a few decision trees.

**Regression and Probabilistic Classification.** Another avenue for extending our framework could involve redefining the explanation forms we have proposed to be more flexible, allowing them to be applied to different contexts such as probabilistic classification or regression.

Potential relaxations of our definitions might include integrating *probabilistic* concepts of sufficiency (Wäldchen et al. (2021); Izza et al. (2021); Arenas et al. (2022); Wang et al. (2021)), applying them within bounded $\epsilon$-ball domains (Wu et al. (2024); Izza et al. (2024); La Malfa et al. (2021)), or focusing on meaningful distributions (Yu et al. (2022); Gorji & Rubin (2022)). Additionally, our definitions could be expanded beyond binary classification to address regression or probabilistic classification scenarios. For instance, in the context of a regression model $f : \mathbb{F} \to \mathbb{R}$, a *sufficient reason* might be defined as a subset $S \subseteq \{1, \ldots, n\}$ of input features such that:

$$\forall \mathbf{z} \in \mathbb{F} \quad ||f(\mathbf{x}_S; \mathbf{z}_{\bar{S}}) - f(\mathbf{x})||_p \leq \delta \tag{12}$$

for some $0 \leq \delta \leq 1$ and an appropriate $\ell_p$-norm. Other notations explored in our work, such as contrastive explanations, can also be adapted within this framework. It is important to note, however, that some complexity results related to Shapley values may differ when transitioning from classification to regression. For example, while computing Shapley values for linear regression models is computationally feasible, the same task may become intractable when applied to classification (Van den Broeck et al. (2022)).

**Homogeneous and Heterogeneous Ensembles.** The complexity analysis primarily focuses on homogeneous ensembles (ensembles containing base models of the same type). However, most of our results extend to heterogeneous ensembles as well. Clearly, all hardness results for homogeneous ensembles also apply to heterogeneous ensembles and will always align with the "hardest" complexity class among the associated models. For example, consider an ensemble composed of both linear models and decision trees. Suppose that for some explainability query $Q$, interpreting an ensemble of linear models is $\mathcal{K}_1$-Complete for a complexity class $\mathcal{K}_1$, and interpreting an ensemble of decision trees is $\mathcal{K}_2$-Complete for a complexity class $\mathcal{K}_2$. If we assume, without loss of generality, that $\mathcal{K}_1 \subsetneq \mathcal{K}_2$, then a heterogeneous ensemble consisting of both linear models and decision trees will be $\mathcal{K}_2$-Hard. This holds for both our non-parameterized results (Section E) and parameterized results (Section 5). For instance, according to Proposition 1, for an ensemble comprising FBDDs and Perceptrons, computing the MSR query would be $\Sigma_2^P$-Hard. Furthermore, based on Proposition 5 and Proposition 7, obtaining the MCR query for a heterogeneous ensemble containing both Perceptrons and FBDDs is para-NP-Hard (though when the ensemble consists only of FBDDs, this query is in coW[1]).

# E    PROOF OF PROPOSITION 1

**Proposition 1.** *Ensembles of FBDDs and ensembles of Perceptrons are (i) coNP-Complete with respect to CSR, (ii) NP-Complete with respect to MCR, (iii) $\Sigma_2^P$-Complete with respect to MSR, (iv) #P-Complete with respect to CC, and (v) #P-Hard with respect to SHAP.*

*Proof Sketch.* We build upon a proof from (Izza & Marques-Silva) that reduces DNF formulas to random forest models to demonstrate that computing prime implicants for random forests is $D^P$-Complete. We expand this by showing that DNF formulas can be transformed into ensembles of FBDDs or Perceptrons in polynomial time, fitting our broader category of *poly-subset constructable functions* which efficiently represent any disjunction of literals. We confirm that FBDDs and Perceptrons belong to this category and proceed to obtain the complexity of various queries through reductions: the CSR query from the TAUT problem, the MCR query from the Vertex-Cover problem, and the MSR query from the Shortest-Implicant-Core problem (Umans (2001)). We note that while (Audemard et al. (2022)) asserts to have established the hardness for the MSR query through a reduction from minimal unsatisfiable sets to DNFs, there is a noted technical gap in this proof (details in Appendix 5), also observed in other similar proofs (Huang et al. (2021)). To the best of our knowledge, we are the first to address this issue effectively with a non-trivial approach, enabling us to confirm $\Sigma_2^P$-Hardness for *DNFs* and related ensembles of poly-subset-constructable functions.

*Full Proof.* We will, in fact, prove this claim for a broader class of models, which we define as *poly-subset-constructible* models (and for the *MCR* query we will require an additional constraint). We will then demonstrate that both FBDDs and Perceptrons fall into this category. Intuitively, poly-subset-constructable models are those for which, given a partial assignment over a subset $\mathbf{x}_S$, we can polynomially construct a function that returns $1$ if and only if the features in $S$ are assigned the values in $\mathbf{x}$. More formally:

**Definition 1.** *We say that a class of models $\mathcal{C}$ is **poly-subset constructable** iff given an assignment $\boldsymbol{x} \in \mathbb{F}$, and a subset $S \subseteq \{1, \ldots, n\}$, it is possible to construct a model $f \in \mathcal{C}$, in polynomial time, for which for all $\boldsymbol{y} \in \mathbb{F}$ it holds that:*

$$f(\mathbf{y}) = \begin{cases} 1 & if \ \ \mathbf{y}_S = \mathbf{x}_S \\ 0 & otherwise \end{cases} \tag{13}$$

We will start by proving that each one of the models analyzed within our framework are poly-subset constructable:

**Lemma 3.** *Perceptrons, FBDDs, and MLPs are all poly-subset constructable.*

*Proof.* We will begin with FBDDs. Given an assignment $\mathbf{x}$ and a subset $S$, we can simply construct an FBDD with a single accepting path $\alpha$ that corresponds to the partial assignment of the features in $S$ with their respective values from $\mathbf{x}$. It clearly follows that $\forall \mathbf{z} \in \mathbb{F}, \ f(\mathbf{x}_S; \mathbf{z}_{\bar{S}}) = 1$. Additionally, for any $\mathbf{y} \in \mathbb{F}$ that does not match the assignments of $\mathbf{x}$ on $S$, we have that $f(\mathbf{y}) = 0$.

For Perceptrons, given some input $\mathbf{x} \in \mathbb{F}$ and a subset $S$, we construct a perceptron with $n$ input features. The single hidden layer $h^1$ is constructed as follows:

$$h_i^1 := \begin{cases} 1 & if \ \ \mathbf{x}_i = 1 \wedge i \in S \\ -1 & if \ \ \mathbf{x}_i = 0 \wedge i \in S \\ 0 & if \ \ i \notin S \end{cases} \tag{14}$$

We additionally define the single bias term as follows:

$$b^1 := -[\sum_{1 \leq i \leq n} (h_i^1 \cdot \mathbf{x}_i)] + \frac{1}{2} \tag{15}$$

It clearly satisfies that:

$$\sum_{1 \leq i \leq n} (h_i^1 \cdot \mathbf{x}_i) = |\{i \in S \wedge \mathbf{x}_i = 1\}| \tag{16}$$

Moreover, it holds that:

$$\sum_{i \in S \wedge \mathbf{x}_i = 1} \mathbf{z}_i \cdot h_i^1 + \sum_{i \in \bar{S} \wedge \mathbf{x}_i = 0} \mathbf{z}_i \cdot h_i^1 = \sum_{i \in S \wedge \mathbf{x}_i = 1} \mathbf{z}_i - \sum_{i \in \bar{S} \wedge \mathbf{x}_i = 0} \mathbf{z}_i \tag{17}$$

Now assume some assignment $(\mathbf{x}_S; \mathbf{z}_{\bar{S}})$ for any $\mathbf{z} \in \mathbb{F}$. It satisfies that:

$$f(\mathbf{x}_S; \mathbf{z}_{\bar{S}}) = step(\sum_{i \in S \wedge \mathbf{x}_i = 1} (\mathbf{x}_S; \mathbf{z}_{\bar{S}})_i - \sum_{i \in \bar{S} \wedge \mathbf{x}_i = 0} (\mathbf{x}_S; \mathbf{z}_{\bar{S}})_i - [\sum_{1 \le i \le n} (h_i^1 \cdot \mathbf{x}_i)] + \frac{1}{2}) =$$
$$step(|\{i \in S \wedge \mathbf{x}_i = 1\}| - \sum_{i \in S \wedge x_i = 1} \mathbf{z}_i \cdot h_i^1 + \frac{1}{2}) = step(\frac{1}{2}) = 1 \tag{18}$$

Now assume some assignment $\mathbf{y} \in \mathbb{F}$ for which $S$ does *not* match the values over $\mathbf{x}$. It holds that:

$$f(\mathbf{y}) = step(\sum_{i \in S \wedge \mathbf{x}_i = 1} \mathbf{y}_i - \sum_{i \in \bar{S} \wedge \mathbf{x}_i = 0} \mathbf{y}_i - [\sum_{1 \le i \le n} (h_i^1 \cdot \mathbf{x}_i)] + \frac{1}{2}) =$$
$$step(|\{i \in S \wedge \mathbf{x}_i = 1 \wedge \mathbf{y}_i = 1\}| - |\{i \in S \wedge \mathbf{x}_i = 0 \wedge \mathbf{y}_i = 1\}| - \sum_{1 \le i \le n} (h_i^1 \cdot \mathbf{x}_i) + \frac{1}{2}) = 0 \tag{19}$$

Hence concluding the construction.

□

The construction for Perceptrons clearly holds for MLPs as well.

**Lemma 4.** *Let $\phi$ be a DNF $\phi := t_1 \vee \ldots \vee t_n$ formula, and let $\mathcal{C}$ be a poly-subset constructable class of functions. Then, it is possible to construct, in polynomial time, a hard voting ensemble $f$ consisting of $2n - 1$ base-models $f_i \in \mathcal{C}$.*

*Proof.* We follow a similar reduction to the one proposed in previous work, which demonstrated that obtaining a prime implicant on a random forest is $D^P$-Complete (Izza & Marques-Silva). The key distinction here is that we need to show that our result holds for any poly-subset constructable class.

Let $\phi := t_1 \vee \ldots \vee t_n$ be a DNF formula. First, we construct $n$ models $\langle f_1, \ldots, f_n \rangle$, where each model $f_i$ corresponds to its respective clause $t_i$. This construction can be completed in polynomial time, given that we assume $\mathcal{C}$ is poly-subset constructable. Each clause $t_i$ corresponds to a partial assignment of values based on the literals that appear in $t_i$. For example, if $t_i := \mathbf{x}_i \wedge \overline{\mathbf{x}_j} \wedge \mathbf{x}_k$, then $t_i$ represents a partial assignment of the features $S := \{i, j, k\}$ with the corresponding assignment of 101.

We can now leverage the fact that $\mathcal{C}$ is poly-subset constructable to build a model $f_i$ corresponding to each $t_i$ such that for all $i \in \{1, \ldots, n\}$, $f_i(\mathbf{x}) = t_i(\mathbf{x})$. To complete the ensemble $f$, we add $n - 1$ models that always return True, denoted as $f_1^t, \ldots, f_{n-1}^t$. Our final ensemble is $f := \langle f_1, \ldots, f_n, f_1^t, \ldots, f_{n-1}^t \rangle$. If $\phi$ is true, then at least one clause $t_i$ is satisfied, meaning that at least $n > \frac{2n-1}{2}$ models in $f$ return True, and therefore $f$ is True. If $\phi$ is false, all models $f_1, \ldots, f_n$ return False, which means that at least $n > \frac{2n-1}{2}$ models return False, and thus $f$ is false. This concludes the construction.

□

**Lemma 5.** *Let $\mathcal{C}$ be some poly-subset constructable class. Then the MSR query for a $k$ ensemble of models from $\mathcal{C}$ is $\Sigma_2^P$-Complete.*

*Proof.* Membership is evident since we can guess an assignment of features $S$ of size $k$, and then use a coNP oracle to verify that $S$ is indeed sufficient. In other words: $\forall \mathbf{z} \in \mathbb{F}$, $f(\mathbf{x}_S; \mathbf{z}_{\bar{S}}) = f(\mathbf{x})$. Next, we will prove $\Sigma_2^P$-Hardness. We begin by briefly discussing a proof proposed by Audemard et al. (2022), highlighting a technical gap in this proof, which also appears in a different proof from Huang et al. (2021). Finally, we will present an alternative proof that resolves this technical issue.

**A techincal gap in previous reductions.** We begin by referring to a previous reduction proposed by Audemard et al. (2022) (Proposition 5), which demonstrates that the *MSR* query for random-forest classifiers is $\Sigma_2^P$-Hard. This is essentially the same objective as our proof, though in our case, we extend the result to a more general class of classifiers (those that are poly-subset constructible) rather than just decision trees. The proof relies on a reduction from the problem of finding *minimal*

*unsatisfiable sets* (*MUS*s) of size $k$, a problem frequently discussed in this context (Ignatiev et al. (2019a)). We will first point out an *existing gap* in this proof, which appears to be similar to gaps in other proofs suggested in previous works (Huang et al. (2021), Proposition 7). Afterward, we will present how this gap can be addressed, and demonstrate that the aforementioned problem is indeed $\Sigma_2^P$-Hard. To the best of our knowledge, we are the first to propose a solution to resolve this gap.

The approach behind the reductions in Audemard et al. (2022); Huang et al. (2021) begins with a given CNF $\phi := c_1 \wedge \ldots \wedge c_m$ and an integer $k$. For each clause, the reductions define $t_i := c_i \vee \overline{s_i}$, introducing a new variable $s_i$ (referred to in Huang et al. (2021) as the *selector* variable, and denoted in Audemard et al. (2022) by $y_i$). Then, they define $\phi' := t_1 \wedge t_2 \wedge \ldots \wedge t_m$, which is still a valid CNF. Negating $\phi'$ produces a DNF equivalent to $\neg\phi'$. The work in Huang et al. (2021) halts the reduction at this point, as their focus is on proving hardness for DNF classifiers. However, Audemard et al. (2022) proceeds further by reducing the DNF classifier to an ensemble of decision trees (via a procedure similar to the one provided in Lemma 4, which reduces a DNF to a more general ensemble of poly-subset constructable functions). Next, both reductions assume $\mathbf{x}$ to be a vector containing only "1"s. The core argument in both reductions is that any selection of $k$ clauses $\{c_1', \ldots, c_k'\}$ from $c_1, \ldots, c_m$ corresponds to selecting $k$ features $\{s_1', \ldots, s_k'\}$ from $s_1, \ldots, s_m$. Consequently, they claim that a minimal unsatisfiable set in $\phi$ of size $k$ corresponds to a sufficient reason of size $k$ involving the variables $s_1, \ldots, s_m$ in $f$.

However, we identify a technical gap in these reductions. In the DNF proof from Huang et al. (2021) and the decision tree ensemble proof from Audemard et al. (2022), the features that can be included in a sufficient reason may include not only the selector variables $s_1, s_2, \ldots, s_m$ (or equivalently $y_1, y_2, \ldots, y_m$ in Audemard et al. (2022)), but also the original variables $x_1, x_2, \ldots, x_n$ from the CNF $\phi$. Consequently, a sufficient subset may be chosen that includes both selector and original variables, which does not guarantee equivalence between the two problems.

**An alternative approach that avoids the technical gap.** Instead of reducing minimal unsatisfiable sets to DNFs and then transforming them further, we will take a different approach. We will reduce from the $\Sigma_2^P$-Complete *Shortest Implicant Core* problem (Umans (2001)) for DNFs, a problem that is related but not entirely equivalent to finding cardinally minimal unsatisfiable sets.

It is already known that computing the *MSR* query over MLPs is $\Sigma_2^P$-Hard (Barceló et al. (2020)), and this complexity result can be derived from the *Shortest-Implicant-Core* problem, not through DNFs, but rather through *boolean circuits*. Specifically for MLPs, this reduction leverages the fact that any boolean circuit can be reduced to an MLP, making the reduction more adaptable. For ensembles, however, a more complicated approach is needed (which will enable its incorporation to DNFs). We begin by introducing the Shortest Implicant Core problem, first defining an implicant, followed by the formal definition of the Shortest-Implicant-Core problem:

**Definition 2.** *Consider $\phi$ as a boolean formula over the literals $\{x_1, \ldots, x_n\}$. An implicant $C := x_1' x_2', \ldots x_l'$ of $\phi$ is defined as a partial assignment over $\phi$'s literals (for all $1 \leq i \leq l$, $x_i'$ is equal to either $x_j$ or $\overline{x_j}$ and each $x_i'$ contains an instance of a different literal), ensuring that any completion of this assignment results in $\phi$ evaluating to true.*

For example for the DNF furmula: $\phi := x_1 \overline{x_2} x_3 \vee x_1 \overline{x_2 x_3}$, we have that $x_1 \overline{x_2}$ is an implicant of $\phi$, since any completion of $x_1 \overline{x_2}$ evaluates $\phi$ to True. We now define the *Shortest-Implicant-Core* problem as follows:

---

**Shortest Implicant Core**:
**Input**: A DNF Formula $\phi := t_1 \vee \ldots \vee t_n$, and an integer $k$.
**Output**: *Yes*, if there exists an implicant $C \subseteq t_n$ of $\phi$ of size $k$, and *No* otherwise.

---

It is firstly important to mention that the *Shortest-Implicant-Core* problem is generally more suitable than the *Shortest-Implicant* problem for a reduction from the *MSR* query since the *core* $t_n$ is important for representing the specific *local* assignment $\mathbf{x} \in \mathbb{F}$ for which the sufficient reason is provided. Additionally, since we are working with DNFs and not general boolean circuits, it is known that the Shortest-Implicant problem for DNFs is GC-Complete (Umans (2001)), which is a class that is strictly less complex than $\Sigma_2^P$.

We will start by briefly presenting the proof presented by Barceló et al. (2020) for reducing the Shortest Implicant Core problem to the problem of solving the *MSR* query for *MLPs*. We will then discuss the desired modifications needed in order to produce this problem for ensembles containing poly-subset constructable functions instead.

For example, let us assume we have the following DNF formula:

$$\phi := x_1\overline{x_5} \vee \overline{x_2 x_6} \vee x_3 x_6 \vee \overline{x_1 x_2} x_4 \vee x_1 x_3 x_5 \tag{20}$$

A straightforward approach to reduce Shortest Implicant Core to MSR is to construct an MLP $f$ that is equivalent to $\phi$, which is feasible since any Boolean circuit can be reduced to an MLP (Barceló et al. (2020)). For the input $\mathbf{x}$, we can create an input where the features corresponding to $t_n$ (in this case, features 1, 3, and 5 are in $C$) are set to 1, while the remaining features can be assigned other values.

There is an issue with this construction. The problem is that the sufficient reason in the constructed MLP may include features that are not part of $t_n$. Specifically, if we set $k := 2$, we observe that no implicant of size 2 exists for $\phi$, yet setting features 3 and 6 to 1 determines the prediction of $f(\mathbf{x})$, indicating that a sufficient reason of size 2 *does* exist. To address this issue, Barceló et al. (2020) suggests constructing a different formula as follows:

$$\phi' := \bigwedge_{i=1}^{3} \left( x_1\overline{x_5} \vee \overline{x_2^i x_6^i} \vee x_3 x_6^i \vee \overline{x_1 x_2^i} x_4^i \vee x_1 x_3 x_5 \right) \tag{21}$$

We can then once again transform this formula into an equivalent MLP $f$, which is possible since, as mentioned before, any Boolean circuit can be transformed into an MLP. More generally, this construction is formalized as follows. Let $X_c$ denote the set of variables that are not mentioned in $t_n$. Then, $\phi'$ will be defined as:

$$\phi^{(i)} := \phi[x_j \to x_j^i, \text{ for all } x_j \in X_c]$$
$$\phi' := \bigwedge_{i=1}^{k+1} \phi^{(i)} \tag{22}$$

However, in our case, the following construction does not apply directly. This is because, as proven in Lemma 4, any DNF can be transformed into an equivalent ensemble of poly-subset constructable functions. However, this does not necessarily imply that we can reduce any Boolean circuit to such models, and since $\phi'$ is no longer a DNF, we cannot apply the same reduction as in (Barceló et al. (2020)).

Instead, we will address this issue differently. Starting with the DNF $\phi$, we will create a new DNF, $\phi'$, which includes only the literals from $t_n$. We will subsequently demonstrate that any implicant $C \subseteq t_n$ for $\phi$ also serves as an implicant for $\phi'$. To facilitate this construction within polynomial time, we rely on the fact that any term $t_i$ in $\phi$, aside from $t_n$, has at most a constant size. We will thus formalize our problem as follows:

---

**Shortest Implicant Core (Constant DNF)**:
**Input**: An integer $k$, a constant integer $d = O(1)$, and a DNF Formula $\phi := t_1 \vee \ldots \vee t_n$, where for any $1 \leq i \leq n-1$, $t_i$ contains less than $d$ literals.
**Output**: *Yes*, if there exists an implicant $C \subseteq t_n$ of $\phi$ of size $k$, and *No* otherwise.

---

**Claim 1.** *Shortest Implicant Core (Constant DNF) is $\Sigma_2^P$-Hard.*

*Proof.* We note that this proof directly follows from the hardness proof of the $\Sigma_2^P$-hardness of the Shortest-Implicant problem (Umans (2001)). The hardness is established using a reduction from $QSAT_2$, where each term in $\phi$ is constructed from a 3-DNF formula conjoined to another constructed 3-DNF. This setup results in a DNF where each term in $\phi$ is of constant size. Consequently, it

follows that the Shortest Implicant Core, where each term in $\{t_1, \ldots, t_{n-1}\}$ is of constant size, is also $\Sigma_2^P$-hard to decide.

$\square$

We will first demonstrate how, from a DNF $\phi := t_1 \vee \ldots \vee t_n$ where the size of any $\{t_1, \ldots t_{n-1}\}$ is bounded by some $d$, we can construct another DNF $\phi'$ in polynomial time and size, where $\phi'$ exclusively includes literals from $t_n$. This approach allows us to later use $t_n$ directly as our desired $\mathbf{x} \in \mathbb{F}$ in the reduction. We will also demonstrate that any implicant $C \subseteq t_n$ for $\phi$ equally serves as an implicant $C \subseteq t_n$ for $\phi'$.

**The construction.** The idea is that if a term $t_i$ in $\phi$ includes literals not present in $t_n$, we can "eliminate" these non-$t_n$ elements by iterating over all possible assignments that cover the literals not included in $t_n$ (and this number is polynomial, as each term in $\phi$ is of constant size). Formally, let there be a DNF $\phi := t_1 \vee t_2 \vee \ldots t_n$. For each $t_i$ where $1 \le i \le n - 1$, let us denote $r_i$ as the subset of literals from $t_i$ which do not participate in $t_n$. First, we define $\phi'$ to be the disjunction of any $t_i$ for which $1 \le i \le n - 1$ and $r_i$ is empty (in other words the "part" of the DNF that only contains assignments of literals that appear in $t_n$).

Now, for each $t_i$, if $r_i$ is not empty, we consider every subset $S \subseteq \{t_1, \ldots, t_{n-1}\}$ where the size of $S$ is $j \le 2^{|r_i|}$. We use $\{x_1^i, x_2^i, \ldots, x_l^i\}$ to represent the literals participating in $r_i$. By iterating through $S$, we can verify whether any potential assignment for $\{x_1^i, x_2^i, \ldots, x_l^i\}$ is encompassed by $S \cup t_i$. Consider, for example, $r_i := x_1 \overline{x_7}$. The literals involved are $\{x_1, x_7\}$. Assume $S := [x_1 x_7 x_8], [\overline{x_1 x_8} x_9], [\overline{x_1} x_7]$. Here, every potential assignment to $x_1 x_7$ (which includes $[x_1 x_7], [\overline{x_1} x_7], [x_1, \overline{x_7}], [\overline{x_1} \overline{x_7}]$) is encompassed by $S \cup t_i$. Another example of a subset $S$ such that $S \cup t_i$ covers all assignments is: $S := \{[\overline{x_1} x_5], [x_1 x_7 x_5]\}$. However, the subset $S := [x_1 x_7], [\overline{x_1} x_7], [x_1 \overline{x_7}]$ fails to enable $S \cup t_i$ to cover all assignments since it misses the assignment $\overline{x_1 x_7}$. It is worth noting that validating whether $S$ covers all literal assignments can be done in linear time with respect to $|S|$, as each $t_i \in S$ accounts for covered assignments, allowing us to save each covered assignment and, after iterating through the entire $S$, check whether all assignments were covered.

For each $t_i$, we initatie a set $\mathbb{S}_i$. For each $S$ that we iterate on concerning $t_i$, if we have that $S \cup t_i$ covers all assignments of literals of $r_i$ we will add $\bigwedge_{t_l \in S} \overline{r_l}$ to $\mathbb{S}_i$. Now, we are in a position to define our refined DNF formula:

$$\phi'' := \phi' \vee [ \bigvee_{1 \le i \le n-1} [ \bigvee_{t_l \in \mathbb{S}_i} t_l ] ] \vee t_n \tag{23}$$

We first note that the construction of $\phi''$ is polynomial. For each $t_i$ we iterate over all subsets of size $2^{|r_i|}$ or less. Overall the number of subsets we iterate on is bounded by:

$$\sum_{1 \le i \le n-1} ( \sum_{1 \le j \le 2^{|r_i|}} \binom{n}{j} ) \le \sum_{1 \le i \le n-1} ( \sum_{1 \le j \le 2^{|r_i|}} n^j ) \le \sum_{1 \le i \le n-1} ( 2^{|r_i|} \cdot n^{2^{|r_i|}} ) \le$$
$$\sum_{1 \le i \le n-1} ( 2^d \cdot n^{2^d} ) \le n \cdot ( 2^d \cdot n^{2^d} ) \tag{24}$$

and since $d$ is constant, the derived term is polynomial in $n$. We have that both the runtime of constructing $\phi''$ as well as the size of $\phi''$ are polynomial in $n$. We now will prove the following lemma regarding the construction:

**Claim 2.** *Any $C \subseteq t_n$ is an implicant for $\phi$ if and only if it is an implicant for $\phi''$.*

*Proof.* If $C \subseteq t_n$ is an implicant, this indicates the existence of a subset $S \subseteq \{t_1, \ldots, t_n\}$ such that the partial assignment $C$ ensures any completion of $C$ within $\bigvee_{t_l \in S} t_l$ evaluates to true (thereby guaranteeing that any completion of $C$ within $\phi$ also evaluates to true). Initially, we assume that each $t_l \in S$ incorporates literals from $t_n$. This leads to the identification of this subset as a subset of terms in $\phi'$, and consequently in $\phi''$. Thus, we identify a subset $S' \subseteq \{t'_1, \ldots, t'_m\}$, where $\{t'_1, \ldots, t'_m\}$ represents the terms of $\phi''$, satisfying $\bigvee_{t_l \in S} t_l = \bigvee_{t_l \in S'} t_l$. Therefore, any extension of $C$ in $\bigvee_{t_l \in S} t_l$

that holds true, ensures that any extension of $C$ in $\bigvee_{t_l \in S'} t_l$ is true (and by extension, any completion of $C$ in $\phi''$). This leads us to conclude that $C$ is also an implicant of $\phi''$.

Revisiting our earlier notation, let $r_i$ represent the set of literals from $t_i$ that are absent in $t_n$. For any $t_l \in S$ where $r_l$ is non-empty, it is required that all assignments to the literals of $r_l$ are covered by $S$ (failing which, there will be an assignment making $\bigvee_{t_l \in S} t_l$ untrue). Essentially, $S$ consists of each $t_i$ where $r_i$ is non-empty, including terms that cover all assignments for $r_i$, and potentially terms where $r_i$ is empty (which would then be included in $\phi'$). We observe that any assignment covering all assignments for $r_i$ can be at most of size $2^{|r_i|}$. This assignment will be generated by our approach and incorporated within $\bigvee_{t_i \in S} t_i$ and subsequently in $\phi''$. Consequently, we identify a subset $S' \subseteq \{t'_1, \ldots, t'_m\}$, where $\{t'_1, \ldots, t'_m\}$ are the terms of $\phi''$, such that $\bigvee_{t_l \in S} t_l = \bigvee_{t_l \in S'} t_l$. Thus, any completion of $C$ over $\bigvee_{t_l \in S} t_l$ that proves true, equally confirms that any completion of $C$ over $\bigvee_{t_l \in S'} t_l$ is true (and thus any completion of $C$ over $\phi''$ is true). Ultimately, this demonstrates that $C$ is also a prime implicant of $\phi''$.

For the second direction, let us assume that $C$ is an implicant for $\phi''$. Consequently, there is a subset $S \subseteq \{t'_1, \ldots, t'_m\}$ of terms from $\phi''$, denoted by $\{t'_1, \ldots, t'_m\}$, where any completion of $C$ over $\bigvee_{t_l \in S'} t_l$ invariably results in true (thus guaranteeing any completion of $C$ over $\phi''$ is true). If $S$ is entirely within $\phi'$, it follows that $S$ is also part of $\phi$, and therefore, there exists a corresponding set $S' \subseteq \{t_1, \ldots, t_n\}$ in which any completion of $C$ over $\bigvee_{t_l \in S} t_l$ is true, as is any completion of $C$ over $\phi$. Therefore, $C$ is also an implicant of $\phi$.

Now, if $S$ includes terms that are not from $\phi'$, these must be assignments in the form of $\bigwedge_{t_l \in S''} \overline{r_l}$, where $S''$ is a subset of $\{t_1, \ldots, t_n\}$ and covers all potential assignments over the literals of $r_i$ for some $t_i \in \{t_1, \ldots t_n\}$. Consequently, we can select the terms from each $S''$, as well as those from $\phi'$, to form a corresponding subset $S' \subseteq \{t_1, \ldots, t_n\}$. It is important to note that all terms in $S'$ that lack any literals from $t_n$ are part of $\phi'$ and thus appear equivalently in both $S$ and $S'$. Additionally, for any term $t_l \subseteq S'$ where $r_l \neq \emptyset$, all assignments over the literals in $r_l$ are covered by both $S$ and $S'$. Therefore, we conclude that $\bigvee_{t_l \in S} t_l = \bigvee_{t_l \in S'} t_l$; thus, if any completion of $C$ confirms that $\bigvee_{t_l \in S} t_l$ is true, it also confirms that $\bigvee_{t_l \in S'} t_l$ is true (and thus $\phi$). Ultimately, this shows that $C$ is also a prime implicant of $\phi$, completing the proof of the claim.

$\square$

This claim proves that a prime implicant $C \subseteq t_n$ of size $k$ exists for $\phi$ if and only if a prime implicant $C \subseteq t_n$ of size $k$ exists for $\phi''$, thus concluding that the Shortest Implicant Core problem for constant DNF is $\Sigma_2^P$-Hard.

$\square$

**Concluding the reduction.** We now finalize our proof for reducing the Shortest-Implicant-Core (Constant DNF) to the MSR for an ensemble of poly-subset constructable functions. Given a tuple $\langle \phi, C, k, d \rangle$, we can construct $\phi'$ from $\phi$ in polynomial time, a task feasible particularly because $d$ is constant. Utilizing Lemma 4, we convert $\phi'$ into an equivalent ensemble $f$ of poly-subset constructable functions. Notably, the literals in $\phi'$ are exclusively those from $t_n$. Therefore, we use $t_n$ to construct our input vector, where any positive assignment in $C$ is represented as a "1" in $\mathbf{x}$ and any negative assignment as a "0". For example, if $t_n := \mathbf{x}_1 \overline{\mathbf{x}_2} \mathbf{x}_3$, we would set $\mathbf{x} := (101)$.

Now, since the input $\mathbf{x}$ includes only features present in $f$, we can assert that an implicant in $t_n$ of size $k$ for $\phi'$ exists if and only if a sufficient reason of size $k$ exists for $\langle f, \mathbf{x} \rangle$. Given that any implicant $C$ in $t_n$ for $\phi'$ also serves as an implicant $C$ in $\phi$, it follows that an implicant of size $k$ for $\phi$ exists if and only if there is a sufficient reason of size $k$ for $\langle f, \mathbf{x} \rangle$, thereby completing our reduction.

$\square$

In contrast to the *MSR* query which was non-trivial, for the *CSR*, *MCR*, *CC*, and *SHAP* queries we can incorporate similar reductions to other works which show hardness for *DNFs*, since we know that an ensemble which is encompassed by models from a class that is poly-subset constructable, can be reduced to DNFs (Lemma 4).

**Lemma 6.** *Let $\mathcal{C}$ be some poly-subset constructable class of functions. Then the CSR query for a $k$ ensemble of models from $\mathcal{C}$ is coNP-Complete.*

*Proof.* Membership in coNP is straightforward since we can guess an assignment $\mathbf{z} \in \mathbb{F}$ and verify whether $f(\mathbf{x}_S; \mathbf{z}_{\bar{S}}) \neq f(\mathbf{x})$ to determine if $S$ is *not* a sufficient reason. For the hardness result, we can apply the same proof provided by Barceló et al. (2020), which involves a reduction from the tautology (*TAUT*) problem for DNFs. In their work, DNFs are reduced to equivalent MLPs to establish coNP-hardness for MLPs. Similarly, in our case, we can utilize the exact same reduction since, in Lemma 4, we demonstrated that any DNF can be reduced to an ensemble of models from $\mathcal{C}$, where $\mathcal{C}$ is poly-subset constructable. This leads to the conclusion that the problem is coNP-complete.

$\square$

In the specific case of *MCR*, we will establish the complexity of the problem for a set of models $\mathcal{C}$ that remains poly-subset constructable, but also adheres to an additional property, which we define as being "closed under symmetric construction". This property is formalized as follows:

**Definition 3.** *Let $\mathcal{C}$ be a class of functions. Then $\mathcal{C}$ is closed under symmetric construction iff for any $f \in \mathcal{C}$, then $\neg f$ can be constructed in polynomial time.*

We will prove that ensembles of FBDDs, Perceptrons, and MLPs all satisfy the aforementioned property:

**Lemma 7.** *The class of ensembles of FBDDs, the class of ensembles of MLPs, and the class of ensembles of Perceptrons are all closed under symmetric construction. In other words, suppose we are given $f$ — an ensemble of $k$ models $f_1, \ldots, f_k$ that are all either Perceptrons, FBDDs, or MLPs; then, $f' := \neg f$ can be constructed in polynomial time.*

*Proof.* We first note that negating individual FBDDs, Perceptrons, and MLPs can be done in polynomial time, as demonstrated by Amir et al. (2024). Applying this transformation to any FBDD $f_i$ within the ensemble results in negating the entire model $f$. The only asymmetric element in this negation arises in majority voting ensembles—where a "1" classification occurs iff $\lceil \frac{k}{2} \rceil$ of the models are classified as "1"—and the negation effectively changes the ensemble to classify "1" iff $\lfloor \frac{k}{2} \rfloor$ models are classified as "1". This asymmetry can be addressed by constructing an additional model, when needed, that always classifies "1". This can be done using Perceptrons, FBDDs, or MLPs. Similarly, in weighted voting ensembles, asymmetry may arise due to the distinction between strict and non-strict inequalities. In other words, while $f$ classifies as "1" iff $\sum_{1 \leq i \leq k} \phi_i \geq 0$, in the negated model $f'$, it will classify as "1" iff $\sum_{1 \leq i \leq k} \phi_i > 0$. This can again be resolved by adding an extra model $f_i$ that always classifies "1" with a very small weight, thereby addressing this issue. Given that any weight in $f_i$, $\phi_i$, is a rational number $\frac{p_i}{q_i}$, the weight of the newly constructed model can be set to $q_1 \cdot q_2 \ldots \cdot q_k$. This additional model will correct the asymmetry in cases where the weighted sum of all models is exactly zero, allowing the ensemble to be negated.

$\square$

We are now prepared to prove the following claim, which will establish that solving *MCR* for ensembles of FBDDs, Perceptrons, and MLPs is NP-complete. This result follows directly from Lemma 7 and Lemma 3.

**Lemma 8.** *Let $\mathcal{C}$ be some poly-subset constructable class of functions, which is also closed under symmetric construction. Then the MCR query for a $k$ ensemble of models from $\mathcal{C}$ is NP-Complete.*

*Proof.* Membership holds because we can guess a subset $S$ and check whether $|S| \leq d$ and $f(\mathbf{x}_S; \mathbf{z}_{\bar{S}}) \neq f(\mathbf{x})$, which confirms the existence of a contrastive reason of size $d$.

For hardness, we can use a similar proof to Barceló et al. (2020), which demonstrates that the MCR problem is NP-Hard for MLPs. They reduce the problem from the *Vertex-Cover* problem, which is NP-Complete. To achieve this, given a graph $G := \langle V, E \rangle$, they construct an equivalent CNF formula:

$$\bigwedge_{(u,v) \in E} (x_u \vee x_v) \tag{25}$$

and encode the CNF formula as an equivalent MLP. In Lemma 4, we proved that any DNF can be reduced to an ensemble with poly-subset constructable base-model functions. However, to extend the proof of Barceló et al. (2020) to apply to our ensembles, we must show that any *CNF* can also be

transformed into an ensemble constructed from these models. This is possible because the family of models $\mathcal{C}$ is not only poly-subset constructable but also closed under symmetric construction. We begin by proving the following relation:

**Claim 3.** *Let $\phi$ be a CNF $\phi := t_1 \wedge \ldots \wedge t_n$ formula, and let $\mathcal{C}$ be a poly-subset constructable class of functions. Then, it is possible to construct, in polynomial time, a hard voting ensemble $f$ consisting of $2n - 1$ base-models $f_i \in \mathcal{C}$ where $\mathcal{C}$ is both a poly-subset constructable class, and is closed under symmetric construction.*

*Proof.* We begin by negating $\phi$ in polynomial time and derive a DNF formula equivalent to $\neg\phi$. From Lemma 4, we know that any DNF can be transformed in polynomial time into an ensemble of models $f \in \mathcal{C}$ (since $\mathcal{C}$ is poly-subset constructable). Given that $\mathcal{C}$ is also closed under symmetric construction, we can negate $f$ and construct $\neg f$ in polynomial time. Thus, we obtain an ensemble $f' := \neg f$ equivalent to $\phi$, completing our proof.

$\square$

We can now conclude our proof, as we know that ensembles of FBDDs, Perceptrons, and MLPs are all poly-subset constructable and closed under symmetric construction. Therefore, any CNF can be reduced to an equivalent ensemble composed of models from this class, allowing us to leverage the vertex-cover problem to establish NP-Hardness.

$\square$

**Lemma 9.** *Let $\mathcal{C}$ be some poly-subset constructable class of functions. Then the CC query for a $k$ ensemble of models from $\mathcal{C}$ is $\#P$-Complete.*

*Proof.* Membership is straightforward by definition. For the hardness result, we can reduce from the Model Counting problem for DNFs, which is known to be $\#P$-Hard Valiant (1979). Given a DNF $\phi$, we can construct an equivalent ensemble of poly-subset constructable functions using Lemma 4 and set $S := \emptyset$ for the *CC* query. Thus, solving *CC* is equivalent to model counting, and the reduction holds.

$\square$

**Lemma 10.** *Let $\mathcal{C}$ be some poly-subset constructable class of functions. Then the SHAP query for a $k$ ensemble of models from $\mathcal{C}$ is $\#P$-Hard.*

*Proof.* We follow the proof of Arenas et al. (2021c), which demonstrated a connection between computing *SHAP* under fully factorized distributions and the model counting problem. The following relation is established:

$$\#f := f(\mathbf{x}) - 2^n \cdot \sum_{i \in \{1, \ldots, n\}} \phi_i(f, \mathbf{x}) \tag{26}$$

where $\#f$ represents the number of positive assignments of $f$ and $\phi$ denotes the Shapley value. This establishes that computing *SHAP* is at least as hard as the model counting problem. Since, as shown in Lemma 4, any DNF can be constructed into an ensemble of poly-subset constructable functions, and model counting for DNFs is known to be $\#P$-Hard Valiant (1979), this concludes the proof, demonstrating $\#P$-Hardness for computing *SHAP*.

$\square$

# F    PROOF OF PROPOSITION 2

**Proposition 2.** *While the CC and SHAP queries can be solved in pseudo-polynomial time for Perceptrons, ensemble-Perceptrons remain $\#P$-Hard even if the weights and biases are given in unary.*

*Proof sketch.* The findings for ensembles are derived from those in Proposition 1. We base the pseudo-polynomial algorithm for CC on the work of (Barceló et al. (2020)). Additionally, we achieve similar outcomes for SHAP using a non-trivial dynamic programming algorithm that solves the SHAP query for Perceptrons in pseudo-polynomial time.

*Full Proof.* For the *CC* query, we refer to the results from Barceló et al. (2020), which demonstrated that when the weights and biases are provided in unary, the *CC* query can be solved in polynomial time for perceptrons. However, according to the findings in Proposition 1, when the weights and biases are not given in unary, the *CC* query is $\#P$-Complete for *ensembles* of perceptrons.

For the *SHAP* query, we use a proof that was proposed by the work of Arenas et al. (2023) which showed a connection between the computation of shapley values for models (under some mild conditions) and the following portion:

$$H_{\mathcal{D}_p}(f, \mathbf{x}, S, k) := \sum_{S \subseteq [n], |S|=k} \mathbb{E}_{\mathbf{z} \sim \mathcal{D}_p}[f(\mathbf{z})|\mathbf{z}_S = \mathbf{x}_S] \tag{27}$$

Specifically, this relationship applies to models that are *closed under conditioning*, defined as follows:

**Definition 4.** *A model $f : \mathbb{F} \to \{0, 1\}$ is closed under conditioning if given some assignment $\boldsymbol{x} \in \mathbb{F}$ and some subset $S \in [n]$, we can construct in polynomial time a function $f' : \mathbb{F} \to \{0, 1\}$ for which it holds that for all $\boldsymbol{y} \in \mathbb{F}$: $f'(\boldsymbol{y}) = f(\boldsymbol{x}_S; \boldsymbol{y}_{\bar{S}})$.*

The connection between models that are closed under conditioning and the computation of $H_{\mathcal{D}_p}$ is as follows (Arenas et al. (2023)):

**Lemma 11.** *Let there be a model $f$ which is closed under conditioning. Then solving the* SHAP *query for $f$ and any $i \in [n]$ can be reduced, in polynomial time, to the problem of computing $H_{\mathcal{D}_p}$*

To apply this lemma to our scenario, we begin by proving the following claim:

**Claim 4.** *Any perceptron $f := \langle \boldsymbol{w}, b \rangle$ is closed under conditioning.*

*Proof.* Given a perceptron $f := \langle \mathbf{w}, b \rangle$, an assignment $\mathbf{x} \in \mathbb{F}$, and a subset $S \in [n]$, we can construct a second perceptron $f' := \langle \mathbf{w}', b' \rangle$, where $\mathbf{w}' := \mathbf{w} \cdot \mathbf{1}_{\bar{S}}$. In other words, we "zero-out" all the feature values in $\mathbf{w}$ corresponding to the set $S$ and leave the features in $\overline{S}$ unchanged. Additionally, we define $b' := b + \sum_{i \in S, \mathbf{x}_i = 1} \mathbf{w}_i$. It now holds that for any $\mathbf{y} \in \mathbb{F}$:

$$
\begin{aligned}
f'(\mathbf{y}) = \sum_{i \in [n]} \mathbf{w}'_i \cdot \mathbf{y}_i + b' = \sum_{i \in S} \mathbf{w}' \cdot \mathbf{y}_i + \sum_{i \in \bar{S}} \mathbf{w}' \cdot \mathbf{y}_i + (b + \sum_{i \in S, \mathbf{x}_i = 1} \mathbf{w}_i) = \\
\sum_{i \in \bar{S}} \mathbf{w} \cdot \mathbf{y}_i + (b + \sum_{i \in S, \mathbf{x}_i = 1} \mathbf{w}_i) = \\
\sum_{i \in \bar{S}} \mathbf{w} \cdot \mathbf{y}_i + (b + \sum_{i \in S, \mathbf{x}_i = 1} \mathbf{w}_i \cdot \mathbf{x}_i + \sum_{i \in S, \mathbf{x}_i = 0} \mathbf{w}_i \cdot \mathbf{x}_i) = f(\mathbf{x}_S; \mathbf{y}_{\bar{S}})
\end{aligned}
\tag{28}
$$

$\square$

Now, we only need to prove that when the weights and biases of a perceptron $f$ are given in unary, $H_{\mathcal{D}_p}(S, k)$ can be computed in polynomial time. This will complete our proof that the *SHAP* query can be solved in pseudo-polynomial time for perceptrons. We use the notation $sim(\mathbf{z}, \mathbf{x})$ to denote the set of features $S$ such that for all $i \in S$, it holds that $\mathbf{z}_i = \mathbf{x}_i$. We begin by demonstrating the following relations:

$$H_{\mathcal{D}_p}(S,k) := \sum_{S \subseteq [n], |S|=k} \mathbb{E}_{\mathbf{z} \sim \mathcal{D}_p}[f(\mathbf{z})|\mathbf{z}_S = \mathbf{x}_S] =$$

$$\sum_{\mathbf{z} \in \mathbb{F}, |sim(\mathbf{z},\mathbf{x})| \geq k} \binom{|sim(\mathbf{z},\mathbf{x})|}{k} \cdot \Big( \prod_{i \in [n], \mathbf{z}_i=1} \big(p(i)\big) \prod_{i \in [n], \mathbf{z}_i=0} \big(1-p(i)\big)\Big) \cdot f(\mathbf{z}) =$$

$$\sum_{\mathbf{z} \in \mathbb{F}, |sim(\mathbf{z},\mathbf{x})| \geq k, f(\mathbf{z})=1} \binom{|sim(\mathbf{z},\mathbf{x})|}{k} \cdot \Big( \prod_{i \in [n], \mathbf{z}_i=1} \big(p(i)\big) \prod_{i \in [n], \mathbf{z}_i=0} \big(1-p(i)\big)\Big) =$$

$$\sum_{j=k}^{j=n} \Big( \sum_{\mathbf{z} \in \mathbb{F}, |sim(\mathbf{z},\mathbf{x})|=j, f(\mathbf{z})=1} \binom{|sim(\mathbf{z},\mathbf{x})|}{k} \cdot \Big( \prod_{i \in [n], \mathbf{z}_i=1} \big(p(i)\big) \prod_{i \in [n], \mathbf{z}_i=0} \big(1-p(i)\big)\Big)\Big) = \tag{29}$$

$$\sum_{j=k}^{j=n} \binom{j}{k} \Big( \sum_{\mathbf{z} \in \mathbb{F}, |sim(\mathbf{z},\mathbf{x})|=j, f(\mathbf{z})=1} \cdot \Big( \prod_{i \in [n], \mathbf{z}_i=1} \big(p(i)\big) \prod_{i \in [n], \mathbf{z}_i=0} \big(1-p(i)\big)\Big)\Big) =$$

$$\sum_{j=k}^{j=n} \binom{j}{k} \Big( \sum_{\mathbf{z} \in \mathbb{F}, |sim(\mathbf{z},\mathbf{x})|=j} \cdot \Big( \prod_{i \in [n], \mathbf{z}_i=1} \big(p(i)\big) \prod_{i \in [n], \mathbf{z}_i=0} \big(1-p(i)\big)\Big)\Big) \cdot \mathbf{1}_{\{f(\mathbf{z})=1\}} =$$

$$\sum_{j=k}^{j=n} \binom{j}{k} \Big( \sum_{\mathbf{z} \in \mathbb{F}, |sim(\mathbf{z},\mathbf{x})|=j} \cdot \Big( \prod_{i \in [n], \mathbf{z}_i=1} \big(p(i)\big) \prod_{i \in [n], \mathbf{z}_i=0} \big(1-p(i)\big)\Big)\Big) \cdot \mathbf{1}_{\{\sum_{i=1}^n \mathbf{w}_i \cdot \mathbf{z}_i + b \geq 0\}}$$

We define $M := \max(\mathbf{w}) + 1$. Next, we define the vector $\mathbf{w}'$ as follows:

$$\mathbf{w}'_i := \begin{cases} -\mathbf{w}_i + M & if \ \mathbf{x}_i = 1 \\ \mathbf{w}_i + M & if \ \mathbf{x}_i = 0 \end{cases} \tag{30}$$

We finally define $T := b + \big( \sum_{i \in [n], \mathbf{x}_i=0} \mathbf{w}_i \big)$. We now prove the following relation:

**Claim 5.** *Given some integer $1 \leq j \leq n$, the following relation between $\mathbf{w}$ and $\mathbf{w}'$ holds:*

$$\sum_{\mathbf{z} \in \mathbb{F}, |sim(\mathbf{z},\mathbf{x})|=j} \mathbf{1}_{\{\sum_{i=1}^n \mathbf{w}_i \cdot z_i + b \geq 0\}} = \sum_{S \subseteq [n], |S|=j} \mathbf{1}_{\{\sum_{i \in S} \mathbf{w}'_i \leq T + M \cdot j\}} \tag{31}$$

*Proof.* We start by noting that the iterating over all vectors $\mathbf{z} \in \mathbb{F}$ for which $|sim(\mathbf{z}, \mathbf{x})| = j$ is equivalent to iterating over all subsets $S \subseteq [n], |S| = j$ and taking the values of the features in $S$ to be those of $\mathbf{x}$ and those of $\overline{S}$ to be $\neg \mathbf{x}$. In other words, iterating over all vectors of the form $(\mathbf{x}_S; \neg\mathbf{x}_{\overline{S}}$ for which $|S| = j$. From this, we can derive the following relation:

$$\sum_{\mathbf{z} \in \mathbb{F}, |sim(\mathbf{z},\mathbf{x})|=j} \mathbf{1}_{\{\sum_{i=1}^n \mathbf{w}_i \cdot z_i + b \geq 0\}} = \sum_{S \subseteq [n], |S|=j} \mathbf{1}_{\{\sum_{i=1}^n \mathbf{w}_i \cdot (\mathbf{x}_i; \neg\mathbf{x}_i) + b \geq 0\}} =$$

$$\sum_{S \subseteq [n], |S|=j} \mathbf{1}_{\{(\sum_{i \in S} \mathbf{w}_i \cdot \mathbf{x}_i) + (\sum_{i \in \overline{S}} \mathbf{w}_i \cdot (\neg\mathbf{x}_i)) + b \geq 0\}} \tag{32}$$

We can continue and show that the new relation holds the following:

$$\sum_{S \subseteq [n], |S|=j} \mathbf{1}_{\{(\sum_{i \in S} \mathbf{w}_i \cdot \mathbf{x}_i) + (\sum_{i \in \overline{S}} \mathbf{w}_i \cdot (\neg\mathbf{x}_i)) + b \geq 0\}} =$$

$$\sum_{S \subseteq [n], |S|=j} \mathbf{1}_{\{(\sum_{i \in S, \mathbf{x}_i=1} \mathbf{w}_i) + (\sum_{i \in \overline{S}, \mathbf{x}_i=0} \mathbf{w}_i) + b \geq 0\}} \tag{33}$$

We move to the second term and prove that the following holds:

$$\sum_{S \subseteq [n], |S|=j} \mathbf{1}_{\{\sum_{i \in S} \mathbf{w}'_i \leq T + M \cdot j\}} =$$

$$\sum_{S \subseteq [n], |S|=j} \mathbf{1}_{\{\sum_{i \in S, \mathbf{x}_i=1}(-\mathbf{w}_i + M) + \sum_{i \in S, \mathbf{x}_i=0}(\mathbf{w}_i + M) \leq T + M \cdot j\}} =$$

$$\sum_{S \subseteq [n], |S|=j} \mathbf{1}_{\{\sum_{i \in S, \mathbf{x}_i=1}(-\mathbf{w}_i) + \sum_{i \in S, \mathbf{x}_i=0} \mathbf{w}_i \leq T\}} = \tag{34}$$

$$\sum_{S \subseteq [n], |S|=j} \mathbf{1}_{\{\sum_{i \in S, \mathbf{x}_i=1}(-\mathbf{w}_i) + \sum_{i \in S, \mathbf{x}_i=0} \mathbf{w}_i \leq b + (\sum_{i \in [n], \mathbf{x}_i=0} \mathbf{w}_i)\}} =$$

$$\sum_{S \subseteq [n], |S|=j} \mathbf{1}_{\{\sum_{i \in S, \mathbf{x}_i=1}(-\mathbf{w}_i) + \sum_{i \in S, \mathbf{x}_i=0} \mathbf{w}_i - \sum_{i \in S, \mathbf{x}_i=0} \mathbf{w}_i - \sum_{i \in \bar{S}, \mathbf{x}_i=0} \mathbf{w}_i \leq b\}} =$$

$$\sum_{S \subseteq [n], |S|=j} \mathbf{1}_{\{\sum_{i \in S, \mathbf{x}_i=1} \mathbf{w}_i + \sum_{i \in \bar{S}, \mathbf{x}_i=0} \mathbf{w}_i + b \geq 0\}}$$

Hence proving the equivalence.

$\square$

We hence can equivalently derive in that $H_{\mathcal{D}_p}(S, k)$ is equivalent to:

$$\sum_{j=k}^{j=n} \binom{j}{k} \Big( \sum_{|S|=j} \big( \prod_{i \in [n], (\mathbf{x}_S; \neg \mathbf{x}_{\bar{S}})_i=1} (p(i)) \prod_{i \in [n], (\mathbf{x}_S; \neg \mathbf{x}_{\bar{S}})_i=0} (1 - p(i)) \big) \big) \cdot \mathbf{1}_{\{\sum_{i \in S} \mathbf{w}'_i \leq T + M \cdot j\}} \tag{35}$$

$\square$

We now will conclude the proof by proving the following claim:

**Claim 6.** $H_{\mathcal{D}_p}(S, k)$ *can be computed in polynomial time.*

*Proof.* We will demonstrate how $H_{\mathcal{D}_p}(S, k)$ can be computed in polynomial time. This will be done using a dynamic programming algorithm that computes a portion of this sum, utilizing the notation of $DP$. Specifically, for all $i \in \mathbb{N}, \ell \in \mathbb{N}_0$ such that $1 \leq i - \ell \leq i \leq n$, and for all $C \in \mathbb{Z}$, we define:

$$DP[i][C][i-\ell] :=$$

$$\begin{cases} \sum_{S \subseteq [n], |S| \leq i-\ell} \big( \prod_{r \in [i], (\mathbf{x}_S; \neg \mathbf{x}_{\bar{S}})_r=1} (p(r)) \prod_{r \in [i], (\mathbf{x}_S; \neg \mathbf{x}_{\bar{S}})_r=0} (1 - p(r)) \big) & if \ \sum_{i \in S} \mathbf{w}'_i \leq C \\ 0 & otherwise \end{cases} \tag{36}$$

where, for any $i, \ell \in \mathbb{N}$ such that it does *not* hold that $1 \leq \ell \leq i \leq n$, we define $DP[i][C][i-\ell] = 0$. Given some $1 \leq j \leq n$, if we take $i := n$, $C := T + Mj$, and $i - \ell := j$ (i.e., $\ell := i - j$), we find that $DP[n][T + Mj][j]$ is equal to:

$$\sum_{|S| \leq j} \big( \prod_{i \in [n], (\mathbf{x}_S; \neg \mathbf{x}_{\bar{S}})_i=1} (p(i)) \prod_{i \in [n], (\mathbf{x}_S; \neg \mathbf{x}_{\bar{S}})_i=0} (1 - p(i)) \big) \cdot \mathbf{1}_{\{\sum_{i \in S} \mathbf{w}'_i \leq T + M \cdot j\}} \tag{37}$$

Thus, we ultimately obtain that:

$$DP[n][T + Mj][j] - DP[n][T + M(j-1)][j-1] =$$

$$\sum_{|S|=j} \big( \prod_{i \in [n], (\mathbf{x}_S; \neg \mathbf{x}_{\bar{S}})_i=1} (p(i)) \prod_{i \in [n], (\mathbf{x}_S; \neg \mathbf{x}_{\bar{S}})_i=0} (1 - p(i)) \big) \cdot \mathbf{1}_{\{\sum_{i \in S} \mathbf{w}'_i \leq T + M \cdot j\}} \tag{38}$$

Therefore, assuming that $DP[i][C][j]$ can be computed in polynomial time, we can iterate over all $j \leq k \leq n$ and compute $DP[i][C][j]$ for each $j$, thereby calculating $H_{\mathcal{D}_P}$ overall. To simplify the presentation, we define:

$$R(\mathbf{x}, S, i) := \Big( \prod_{r \in [i], (\mathbf{x}_S; \neg \mathbf{x}_{\bar{S}})_r = 1} \big(p(r)\big) \prod_{r \in [i], (\mathbf{x}_S; \neg \mathbf{x}_{\bar{S}})_r = 0} \big(1 - p(r)\big) \Big) \tag{39}$$

Thus, we can simply express it as:

$$DP[i][C][i - \ell] := \begin{cases} \sum_{S \subseteq [n], |S| \le i - \ell} R(\mathbf{x}, S, i) & if \ \sum_{i \in S} \mathbf{w}'_i \le C \\ 0 & otherwise \end{cases} \tag{40}$$

For our final step, we will now prove the correctness of our dynamic programming algorithm:

**Lemma 12.** *There exists a polynomial dynamic programming algorithm that computes $DP[i][C][j]$ for any $C \in \mathbb{Z}$ and any $1 \le \ell \le i \le n$.*

*Proof.* We present the following inductive relation:

$$DP[i + 1][C][i + 1 - \ell] =$$
$$\begin{cases} DP[i][C][i - \ell] \cdot (1 - p(i)) + DP[i][C - \mathbf{w}'_{i+1}][i - \ell] \cdot p(i) & if \ \mathbf{x}_i = 1 \\ DP[i][C][i - \ell] \cdot p(i) + DP[i][C - \mathbf{w}'_{i+1}][i - \ell] \cdot (1 - p(i)) & if \ \mathbf{x}_i = 0 \end{cases} \tag{41}$$

We also define that $DP[i][C][j] = 0$ for any $i, C, j < 0$ and further define that when $i = 0$, then $DP[i][C][j] = 1$. We begin with the induction base, i.e., when $i = 1$. If $\ell \ge 1$, then we have $DP[1][C][1 - \ell] = 0$, which satisfies our conditions. The only remaining case is when $\ell = 0$. In the case that $\mathbf{x}_i = 1$:

$$DP[1][C][1 - \ell] = DP[1][C][1] = DP[0][C][0] \cdot (1 - p(1)) + DP[0][C - s_1][0] \cdot p(1) = \\ DP[0][C - s_1][0] \cdot p(1) \tag{42}$$

If it also holds that $s_1 \le C$, we obtain:

$$DP[1][C][1 - \ell] = DP[0][C - s_1][0] \cdot p(1) = p(1) = R(\mathbf{x}, \{s_1\}, 1) \tag{43}$$

However, if $s_1 > C$ we get that:

$$DP[1][C][1 - \ell] = DP[0][C - s_1][0] \cdot p(1) = 0 \tag{44}$$

As required. For the other scenario, we get that if $\mathbf{x}_i = 0$ it holds that:

$$DP[1][C][1 - \ell] = DP[1][C][1] = DP[0][C][0] \cdot (p(1)) + DP[0][C - s_1][0] \cdot p(1) = \\ DP[0][C - s_1][0] \cdot (1 - p(1)) \tag{45}$$

If it additionally holds that $s_1 \le C$, then we have:

$$DP[1][C][1 - \ell] = DP[0][C - s_1][0] \cdot (1 - p(1)) = (1 - p(1)) = R(\mathbf{x}, \{s_1\}, 1) \tag{46}$$

However, if $s_1 > C$ we get that:

$$DP[1][C][1 - \ell] = DP[0][C - s_1][0] \cdot (1 - p(1)) = 0 \tag{47}$$

as required. For the inductive step, we assume correctness holds for some $i \in \mathbb{N}$ and show that:

$$DP[i+1][C][i+1-\ell] :=$$
$$DP[i][C][j] \cdot p(i) + DP[i][C - \mathbf{w}'_{i+1}][j-1] \cdot (1 - p(i)) =$$
$$\sum_{|S| \leq i-\ell, |\mathbf{x}_S|_1 \leq C} R(k, \mathbf{x}, S, i) \cdot p(i+1) +$$
$$\sum_{|S| \leq i-\ell, |\mathbf{x}_S|_1 \leq C - \mathbf{w}'_{i+1}} R(k, \mathbf{x}, S, i) \cdot (1 - p(i+1)) = \quad (48)$$
$$\sum_{|S| \leq i-\ell+1, |\mathbf{x}_S|_1 \leq C, i+1 \notin S} R(k, \mathbf{x}, S, i) \cdot p(i+1) +$$
$$\sum_{|S| \leq i-\ell+1, |\mathbf{x}_S|_1 \leq C, i+1 \in S} R(k, \mathbf{x}, S, i) \cdot (1 - p(i+1)) =$$
$$R(k, \mathbf{x}, S, i+1)$$

$\square$

We conclude by presenting the complete algorithm described for computing $H_{\mathcal{D}_P}(S, k)$ with the following pseudocode:

---
**Algorithm 1** Computing $H_{\mathcal{D}_P}(S, k)$

---
**Input** $f$, $\mathbf{x}$, $S$, $k$

1: $T \leftarrow b + \left( \sum_{i \in [n], \mathbf{x}_i = 0} \mathbf{w}_i \right)$
2: $H_{\mathcal{D}_P} \leftarrow 0$
3: **for each** $k \leq j \leq n$ **do**
4: $\quad H_{\mathcal{D}_P} \leftarrow H_{\mathcal{D}_P} + \binom{j}{k} \cdot (DP[n][T + Mj][j] - DP[n][T + M(j-1)][j-1])$
5: **end for**
6: **return** $H_{\mathcal{D}_P}$

---

The following proof demonstrates that solving the *SHAP* query for Perceptrons can be done in polynomial time, assuming the weights and biases are provided in unary.

## G  PROOF OF PROPOSITION 3

**Proposition 3.** *There is no explainability query $Q$ for which the class of MLPs is strictly more c-interpretable than the class of ensemble-MLPs.*

*Proof.* We will specifically prove this claim for a much larger set of functions that we will refer to as models that are *closed under ensemble construction*:

**Definition 5.** *We say that a class of models $\mathcal{C}$ is **closed under ensemble construction** if given an ensemble $f$ containing models from $\mathcal{C}$, we can construct in polynomial time a model $g \in \mathcal{C}$ for which $\forall \mathbf{x} \in \mathbb{F}, f(\mathbf{x}) = g(\mathbf{x})$.*

We first note that from the definition of a model that is closed under ensemble construction, the following claim holds: Let there be a class of models $\mathcal{C}$, which is closed under ensemble construction. Let us denote $\mathcal{C}_E$ as the class of ensemble models that consist of base-models from class $\mathcal{C}$. Then for any explainability query $Q$: then it holds that $Q(\mathcal{C}_E) \leq_P Q(\mathcal{C})$. Now, if $Q(\mathcal{C})$ belongs to a complexity class that is closed under polynomial reductions, it hence must hold that $Q(\mathcal{C}_E)$ does not belong to a strictly harder complexity class. We are now only left to prove the following claim:

**Claim 7.** *Let $\mathcal{C}_{MLP}$ denote the class of models that are represented by MLPs. Then $\mathcal{C}_{MLP}$ is closed under ensemble construction.*

*Proof.* Consider a model $f$ comprising $k$ individual MLPs denoted as $f_1, \ldots, f_k$ We can devise a new MLP, $f'$, by integrating the hidden layers from each $f_i$. Specifically, $f'$'s first hidden layer is constructed by concatenating the first hidden layers of $f_1, \ldots, f_k$, and similarly for subsequent layers up to the highest number of layers present in any $f_i$. For models with fewer layers, we introduce

"dummy" layers equipped with weights of 1 and biases of 0, effectively passing their last actual output through unchanged. In this initial setup, the layers in $f'$ corresponding to each $f_i$ are only linked to their respective preceding layers within the same $f_i$, thus lacking full connectivity across the different models $f_j$ such that $j \neq i$. To amend this, connectivity can be enhanced by adding inter-layer connections with weights of 1 and biases of 0, ensuring each layer does not influence the next across the different sub-models.

Finally, we observe that the constructed $f'$ outputs $k$ distinct values corresponding to the outputs of each model $f_1, \ldots, f_k$ (the output value prior to the step function). We need to introduce a new output layer to $f'$ to implement a majority voting mechanism. This is conceptualized as a weighted voting process where each model $f_i$ is assigned a specific weight $\phi_i$. This can be realized by adding a fully connected layer that consolidates the $k$ outputs into a single output in the final layer of $f'$. In this layer, each output $i \in \{1, \ldots, k\}$ is assigned a weight $w_i := \phi_i$, and we set the bias of this last layer to 0. Consequently, applying a step function over $f'$ results in an output that represents a weighted majority vote of the ensemble $f$. Additionally, a non-weighted majority vote can be modeled in the setting where all weights $\phi_1 = \phi_2 = \ldots = \phi_k$ are equal. We thus establish that $f' \in \mathcal{C}_{MLP}$, and that for all $\mathbf{z} \in \mathbb{F}$, $f(\mathbf{z}) = f'(\mathbf{z})$ thereby confirming that $\mathcal{C}_{MLP}$ is closed under ensemble construction.

□

## H    PROOF OF PROPOSITION 4

**Proposition 4.** *An ensemble consisting of either Perceptrons, FBDDs, or MLPs, parameterized by the maximal base-model size is (i) para-coNP Complete with respect to CSR, (ii) para-NP-Complete with respect to MCR, (iii) para-$\Sigma_2^P$-Complete with respect to MSR, and (iv) para-$\#P$-Complete with respect to CC, (v) para-$\#P$-Hard with respect to SHAP.*

*Proof Sketch.* The full proof appears in Appendix L. To prove para-NP/para-coNP hardness for MCR and CSR, respectively, we reduce from the well-known NP-Complete Subset-Sum Problem (SSP) to the problem of solving CSR/MCR for ensembles consisting of only two Perceptrons. To establish para-$\Sigma_2^P$-Hardness for the MSR query, we employ a more intricate reduction from the lesser-known Generalized Subset-Sum Problem (GSSP) (Schaefer & Umans (2002); Berman et al. (2002)), a $\Sigma_2^P$-Complete problem. This reduction demonstrates that solving the MSR query in an ensemble of only five perceptrons is already $\Sigma_2^P$-Hard.

*Full Proof.* Membership is straightforward from the definition of para-$\mathcal{K}$ and the completeness of all the non-paramaterized versions to each corresponding complexity class $\mathcal{K}$, as proven in Proposition 1. For example, the non-parameterized version of *CSR* for an ensemble of Perceptrons, FBDDs, or MLPs is coNP-Complete. The same holds for the other complexity classes.

**Hardness.** The reduction for the *CSR* query was provided as a direct reduction from the *TAUT* problem. Since *TAUT* is hard when restricted to *3-DNF*s as well, then hardness for an ensemble consisting of *constant* sized-base lines is straightforward (for any $k \geq 3$). In the case of the *MCR* reduction, the result already holds for any $k = 2$ since the reductino inherently produces ensembles consisting of input dimensions of at most 2. For the *CC* query.

Finally, in the case of *MSR*, we note that the Shortest Implicant Core problem in its classic form presented by Umans (2001) describes general DNFs and not restricted DNFs. However, in a consequent work Dick et al. (2009), it was proven that the Shortest implicant core problem for DNFs with *constant* term size is also $\Sigma_2^P$-Hard.

□

## I    PROOF OF PROPOSITION 5

**Proposition 5.** *For ensembles of $k$-FBDDs (i) the CSR query is coW[1]-Complete, (ii) the MCR query is W[1]-Hard and in W[P], (iii) the CC query is $\#W[1]$-Complete, and (iv) the SHAP query is W[1]-Hard and in XP.*

*Proof Sketch.* Membership for the CSR query is established through a many-to-one FPT reduction to the complementary version of the $k$-Clique problem, while hardness is demonstrated with a many-to-

one FPT reduction from the complementary of the $k$-Multicolored-Clique problem, both of which are known to be W[1]-Complete. The CC query extends these results, as the counting versions of these tasks are #W[1]-Complete. #$W[1]$-Hardness for SHAP can be inferred from the complexity of CC, given that the tractability of SHAP is linked to model counting (Van den Broeck et al. (2022)). For membership, a more detailed proof places SHAP within XP, which is made possible by the assumption of feature independence. Specifically, when the distribution is uniform, SHAP is proven to be #$W[1]$-Complete (see Lemma 17). Lastly, the hardness of the MCR query is demonstrated through a reduction similar to that used for CSR, and its membership is established by reducing it to the weighted circuit satisfiability (WCS) problem for circuits with arbitrary depth. The full proof appears in Appendix I.

*Proof.* We begin by demonstrating the initial complexity result stated in the proposition:

**Lemma 13.** *For ensembles of $k$-FBDDs, the CSR query is coW[1]-Complete.*

*Proof.* We execute a many-one FPT reduction from the complement of the Multi-Color Clique problem to the corresponding CSR query, and from the CSR query to the complement of the $k$-Clique problem. Both the Multi-Color Clique and the $k$-Clique problems are known to be W[1]-Complete. We will begin with the first direction to establish membership in coW[1]. We start by defining the $k$-Clique problem:

---

$k$-**Clique**:
**Input**: A graph $G = \langle V, E \rangle$.
**Parameter:** Integer $k$.
**Output**: *Yes*, if there exists a clique of size larger than $k$ in $G$.

---

**Membership.** We initiate the reduction by demonstrating membership in coW[1]. As discussed in Section A, our aim is to establish membership for the weighted voting scenario. However, for clarity, we will initially focus on the simpler case of regular majority voting. Subsequently, we will elaborate on how this proof can be extended to the weighted version.

**Membership for standard majority vote.** We will specifically demonstrate a reduction from $\overline{\text{CSR}}$ to $k$-Clique, establishing that *CSR* resides in coW[1]. Given an instance $\langle f, \mathbf{x}, S \rangle$, $\overline{\text{CSR}}$ inquires whether $S$ is *not* a sufficient reason with respect to $\mathbf{x}$, or in other words, if $\overline{S}$ is contrastive. We will begin by establishing the following claim, which concerns the fact that ensembles of FBDDs are closed under conditioning (refer to Definition 4):

**Claim 8.** *Given a model $f$ which is an ensemble of $k$ FBDDs, some input $\mathbf{x}$, and subset $S \subseteq [n]$, then $f$ is closed under conditioning.*

*Proof.* We first observe that any individual FBDD $f_i$ in the ensemble can be conditioned over $\mathbf{x}_S$. This is accomplished by creating a modified model $f_i'$ as follows: we start by duplicating $f_i$, that is, $f_i' := f_i$. We then iterate through the tree from the top downwards, examining all splits. Each split corresponds to an assignment to a feature $j$, which can be set to either 0 or 1. If $j \in S$, we will retain all paths that extend from the subtree and follow the assignment $\mathbf{x}_j$, while deleting all paths that follow the assignment $\neg\mathbf{x}_j$. Ultimately, the resulting tree $f_i$ adheres to the following condition:

$$\forall \mathbf{z} \in \mathbb{F} \quad [f_i(\mathbf{x}_S; \mathbf{z}_{\bar{S}}) = f_i'(\mathbf{z}] \tag{49}$$

We observe that following the modifications, any split in the tree $f_i'$ concerning the features $j \in S$ results in only one viable path, as the subtree associated with the contrary assignment was removed earlier. Consequently, $f_i'$ effectively becomes a tree defined solely over the features in $\bar{S}$. Therefore, the following assertion is valid:

$$\forall \mathbf{z} \in \mathbb{F} \quad [f_i(\mathbf{x}_S; \mathbf{z}_{\bar{S}}) = f_i'(\mathbf{z}_{\bar{S}})] \tag{50}$$

We have thus demonstrated that a single FBDD is closed under conditioning. By applying this procedure to each tree within the $k$-ensemble $f$, we arrive at the following conclusion:

$$\forall \mathbf{z} \in \mathbb{F} \quad [f(\mathbf{x}_S; \mathbf{z}_{\bar{S}}) = f'(\mathbf{z}_{\bar{S}})] \tag{51}$$

□

We will establish a secondary minor claim that will be beneficial for our reduction:

**Claim 9.** *Given a model $f$ which is an ensemble of $k$ FBDDs, then $f' := \neg f$ can be constructed in polynomial time.*

*Proof.* We first note that negating a single FBDD can be achieved by switching each leaf node's assignment from "1" to "0" and vice versa. Applying this modification to any FBDD $f_i$ within the ensemble results in negating the entire model $f$.

Now, employing Claim 8, an ensemble $f$ can be conditioned on a partial assignment of features, allowing us to develop a new model $f'$ that is conditioned on $\mathbf{x}_S$. In simpler terms, $f'$ retains only the features from $\overline{S}$ and satisfies the following condition:

$$\forall \mathbf{z} \in \mathbb{F} \quad [f(\mathbf{x}_S; \mathbf{z}_{\bar{S}}) = f'(\mathbf{z}_{\bar{S}})] \tag{52}$$

Given the ensemble $f = (f_1, f_2, \ldots, f_k)$, the reduction constructs $f' = (f'_1, f'_2, \ldots, f'_k)$ and uses it to construct a graph $G$. The final setup of the reduction is $\langle G, k' := \lceil \frac{k}{2} \rceil \rangle$. We will now detail the construction of the graph $G$. First, we compute $f'(\mathbf{x}_{\bar{S}})$, which is equivalent to $f(\mathbf{x}_S; \mathbf{x}_{\bar{S}}) = f(\mathbf{x})$. If $f'(\mathbf{x}_{\bar{S}}) = 1$, we negate the ensemble $f'$. This negation is carried out using Lemma 9. Thus, we can generally assume that $f'(\mathbf{x}_{\bar{S}}) = 0$.

The graph we construct will be a $k$-partite graph, with each part corresponding to each tree in the ensemble $f'$. The vertices and edges of the graph are constructed as follows: We iterate over the leaf nodes in each tree of $f'$, and every leaf node corresponding to the assignment of $\neg f'(\mathbf{x})$ is designated as a vertex $v_i$ in the graph (associated with the specific tree this node is part of). We will now proceed to describe the construction of the edges of the graph.

First, it is important to note that any two vertices within the same part of the graph (i.e., associated with the same tree) will not be connected by an edge. We iterate over any two paths in two *distinct* trees —— and hence associated with two different parts of the graph. We consider two paths, $\alpha$ and $\alpha'$, from different trees in $f'$ to "match" if they do not "collide" on any variable. Specifically, there should be no feature $i$ associated with both $\alpha$ and $\alpha'$ where the assignment of $i$ in $\alpha$ is $\mathbf{x}_i \in \{0, 1\}$ and the assignment in $\alpha'$ is $\neg \mathbf{x}_i$. The two paths "match" if there is no such collision.

Now, each vertex $v_i$ in the graph is linked to a particular leaf node in one of the trees (and consequently to the path $\alpha$ that leads to this leaf node). The reduction will establish an edge between any two vertices $v_i$ and $v_j$, which correspond to paths $\alpha$ and $\alpha'$ from two separate trees, if and only if the paths $\alpha$ and $\alpha'$ "match" and both paths conclude at a terminating node with a True assignment, that is, classified as 1.

We will now prove that $S$ is not a sufficient reason with respect to $\langle f, \mathbf{x} \rangle$ if and only if there exists a clique in the graph of size larger than $k' := \lceil \frac{k}{2} \rceil$. The proof will be divided into several distinct claims. First, we will establish the following claim:

**Claim 10.** *For the aforementioned reduction construction, $S$ is not a sufficient reason concerning $\langle f, \boldsymbol{x} \rangle$ if and only if there exists a partial assignment $\boldsymbol{z}_{\bar{S}}$ for which $f'(\boldsymbol{z}_{\bar{S}}) = 1$.*

*Proof.* By definition, $S$ is not a sufficient reason for $\langle f, \mathbf{x} \rangle$ if and only if:

$$\exists \mathbf{z} \in \mathbb{F} \quad [f(\mathbf{x}_S; \mathbf{z}_{\bar{S}}) \neq f(\mathbf{x})] \tag{53}$$

This equivalently means that $\overline{S}$ is contrastive with respect to $\langle f, \mathbf{x} \rangle$. Given that $f'$ is conditioned on $\mathbf{x}_S$ and includes only the features from $\bar{S}$, it equivalently follows that $S$ is not a sufficient reason concerning $\langle f, \mathbf{x} \rangle$ iff:

$$\exists \mathbf{z} \in \mathbb{F} \quad [f'(\mathbf{z}_{\bar{S}}) \neq f'(\mathbf{x}_{\bar{S}}) = 0] \iff \exists \mathbf{z} \in \mathbb{F} \quad [f'(\mathbf{z}_{\bar{S}}) = 1] \tag{54}$$

This equation holds based on our assumption about the negation of $f'$ during its construction. Additionally, the condition where $f'(\mathbf{z}_{\bar{S}}) = 1$ occurs if and only if at least $\lceil \frac{k}{2} \rceil$ models in the ensemble $f'$, specifically $f'1, \ldots, f'\lceil \frac{k}{2} \rceil$, have $f'i(\mathbf{z}_{\bar{S}}) = 1$.

We will begin by establishing a smaller claim.

**Claim 11.** *For the aforementioned reduction construction, the following condition is satisfied: there exists some $\mathbf{z} \in \mathbb{F}$ for which $f'(\mathbf{z}_{\bar{S}}) = 1$ if and only if there exists a subset of $j \geq \lceil \frac{k}{2} \rceil$ trees: $f'_1, \ldots, f'_j$, in which it is possible to select one path $\alpha_i$ in each tree $f'_i$ such that each path terminates at a "True" (classification = 1) node, and each pair of distinct paths "match".*

*Proof.* We first observe that upon proving this lemma, it will be established that the claim (the existence of $j \geq \lceil \frac{k}{2} \rceil$ trees satisfying the aforementioned property) holds if and only if $S$ is not a sufficient reason for $\langle f, \mathbf{x} \rangle$. This is a direct outcome of the equivalence property mentioned in equation 54.

Let us assume we have a set of $j$ different paths $\alpha_1, \ldots, \alpha_j$ within $j$ distinct trees: $f''_1, \ldots, f''_j$, where each $f''_i$ is a model from the ensemble $f_1, \ldots, f_k$, with each path chosen within one of the trees. All paths are selected such that they terminate on a "True" node (assignment 1) and every pair of paths from two distinct trees "matches". According to the definition of "matching" paths, this means that there is no feature $i$ for which paths in two different trees disagree on the assignment of that feature. Consequently, we can adopt the partial assignment $\mathbf{z}_i$ for each feature $i$ that is assigned a value in one of these paths. Let us denote this partial assignment as $\mathbf{z}_{S'}$ for some $S' \subseteq \bar{S}$. It therefore, follows that if we fix the features in $S'$ to their values in $\mathbf{z}$, the prediction of each one of the distinct trees: $f''_1, \ldots, f''_j$ will be classified as 1. This is equivalent to stating that for any $S'' \subseteq \bar{S}'$, the following holds:

$$\forall \mathbf{z}' \in \mathbb{F} \quad [f''_1(\mathbf{z}_{S''}; \mathbf{z}'_{\bar{S} \setminus S''}) = f''_2(\mathbf{z}_{S''}; \mathbf{z}'_{\bar{S} \setminus S''}) = \ldots = f''_j(\mathbf{z}_{S''}; \mathbf{z}'_{\bar{S} \setminus S''}) = 1] \implies$$
$$\exists \mathbf{z}' \in \mathbb{F} \quad [f''_1(\mathbf{z}_{S''}; \mathbf{z}'_{\bar{S} \setminus S''}) = f''_2(\mathbf{z}_{S''}; \mathbf{z}'_{\bar{S} \setminus S''}) = \ldots = f''_j(\mathbf{z}'_{S''}; \mathbf{z}'_{\bar{S} \setminus S''}) = 1] \tag{55}$$

If we consider $j \geq \lceil \frac{k}{2} \rceil$, then it clearly follows that:

$$\exists \mathbf{z}' \in \mathbb{F} \quad [f'(\mathbf{z}_{S''}; \mathbf{z}'_{\bar{S} \setminus S''}) = 1] \tag{56}$$

Therefore, we can assign $\mathbf{y}_{\bar{S}} := (\mathbf{z}_{S''}; \mathbf{z}'_{\bar{S} \setminus S''})$ and it will be established that:

$$\exists \mathbf{y} \in \mathbb{F} \quad [f'(\mathbf{y}_{\bar{S}}) = 1] \tag{57}$$

For the other direction, let us assume that there is *no* set of $j \geq \lceil \frac{k}{2} \rceil$ trees that terminate at a 1 "True" assignment, such that there is a viable choice of path $\alpha_i$ in each tree $f''_i$ where each pair of distinct paths in this set of trees "matches". From this assumption, it follows that there is *no* group of $j \geq \lceil \frac{k}{2} \rceil$ trees for which a partial assignment $\mathbf{z}_{S'}$, with $S' \subseteq \bar{S}$, can be fixed to $\mathbf{z}$ ensuring that the prediction of all the trees $f''_1, \ldots, f''_j$ will remain 1. More specifically, this indicates that there is no set of $j \geq \lceil \frac{k}{2} \rceil$ trees for which an assignment to $\bar{S}$ guarantees that the prediction of all $f''_1, \ldots, f''_j$ trees will remain 1.

Since there are no $j \geq \lceil \frac{k}{2} \rceil$ trees where an assignment to $\bar{S}$ leads to all these trees predicting 1, the value of $f'$ will consistently be 0 for any possible assignment to the features in $\bar{S}$. Therefore, it is established that:

$$\forall \mathbf{y} \in \mathbb{F} \quad [f'(\mathbf{y}_{\bar{S}}) = 0] \tag{58}$$

$\square$

This concludes this segment of the proof. We have now demonstrated that $S$ is not a sufficient reason concerning $\langle f, \mathbf{x} \rangle$ if and only if there exists a subset of $j \geq \lceil \frac{k}{2} \rceil$ trees, each with a different path that finishes at a positive terminal node, and where each pair of paths "match". We will now proceed to prove the following claim, which will conclude our general proof:

**Claim 12.** *In the aforementioned reduction construction, there is a clique of size greater than $k' \geq \lceil \frac{k}{2} \rceil$ in $G$ if and only if there exist $k'$ distinct trees in $f'$ with $k'$ distinct paths (one per tree) that end at a True "1" node, and each pair of distinct paths "match".*

*Proof.* Assuming that there are $k' \geq \lceil \frac{k}{2} \rceil$ paths in $k'$ distinct trees that "match" and end at a 1 (True) node, the reduction construction implies that each of these paths corresponds to a vertex in $G$ (as they terminate on a 1 node). Furthermore, given that these paths "match" according to our construction, there will be an edge connecting each pair of vertices. Consequently, the set of these $k'$ vertices forms a clique in $G$, establishing the existence of a clique of size $k'$ in $G$.

For the second direction, suppose there is no subset of $k'$ trees or more. This equivalently means that for any subset of $j \geq \lceil \frac{k}{2} \rceil$ paths chosen from $j$ distinct trees that end on a 1 node, there exists at least one pair of distinct paths that do *not* match. According to our construction, this implies that the subgraph $G' \subseteq G$, corresponding to these vertices (each representing one of the paths), is not a clique. Therefore, it follows that for any subgraph $G' \subseteq G$ with $\lceil \frac{k}{2} \rceil$ or more vertices, $G'$ is not a clique. This completes the reduction.

**Membership for weighted Vote.** In our previous proof, we conditioned $f$ on the partial assignment $\mathbf{x}_S$ to derive the model $f'$. We demonstrated that $S$ not being sufficient with respect to $\langle f, \mathbf{x} \rangle$ equates to a satisfying assignment in $f'$. We will now extend this proof to the weighted version, where a satisfying assignment does not necessarily correspond to a set of $\lceil \frac{k}{2} \rceil$ trees with a "1" classification. In this version, each tree $f'_i$ is associated with a weight $\phi_i$, and $f'$ is defined as follows:

$$f'(\mathbf{x}) := step(\sum_{1 \leq i \leq n} \phi_i \cdot f'_i(\mathbf{x})) \tag{59}$$

Therefore, we can implement the following procedure: Iterate over the power set of all possible trees (representing all potential sets of different trees), which includes iterating over subsets $S' \subseteq \{1, \ldots, k\}$. This enumeration is bounded by $O(2^k)$ and thus can be performed in FPT time. For each selected combination $S'$, representing a choice of $|S'|$ distinct trees, we check whether:

$$step(\sum_{i \in S'} \phi_i \cdot f'_i(\mathbf{x})) > 0 \tag{60}$$

This corresponds to checking whether:

$$step(\sum_{i \in S'} \phi_i \cdot f'_i(\mathbf{x})) = step(\sum_{i \in S'} \phi_i \cdot f'_i(\mathbf{x}) + \sum_{i \in \bar{S}'} \phi_i \cdot f'_i(\mathbf{x})) = step(\sum_{1 \leq i \leq n} \phi_i \cdot f'_i(\mathbf{x})) > 0 \tag{61}$$

Thus, we only need to "check" subsets $S'$ where equation 62 is satisfied (as these alone correspond to a positive instance of $f'$). In our reduction, for each subset, $S'$ satisfying equation 62, we construct a subgraph $G'$ in the same manner as in the previous reduction: each path for each tree associated with $S'$ becomes a vertex in $G'$, and an edge between two vertices is formed between two edges iff two distinct paths "match". Additionally, we add $k - |S'|$ extra vertices to each such graph, which are connected to all other vertices in $G'$. We then construct $G$ as the union of all such sub-graphs $G'$ that were derived from each $S'$, and the reduction results in $\langle G, k \rangle$.

Previously, in the classic majority-vote scenario, we demonstrated that a positive assignment in $f'$ corresponds to a clique in $G$, with each vertex in the clique representing its associated path in $f'$. Thus, in our current construction, if there is a positive assignment to $f'$, then there exists some subset $S'$ such that:

$$step(\sum_{i \in S'} \phi_i \cdot f'_i(\mathbf{x})) > 0 \tag{62}$$

This implies that there is a subset $S'$ of distinct trees within a subgraph $G'$ that forms a clique of size $|G'|$ (where each vertex in $G'$ corresponds to a path in $f'$). Since these vertices are also connected to an additional $k - |S'|$ vertices included in our construction, there is also a corresponding clique of size $k$ within $G'$, and thus in $G$ as well.

However, if a positive assignment for $f'$ does not exist, it indicates that for any subset of trees $S'$ in $f'$, there is no corresponding clique of size $|S'|$ where each vertex corresponds to a path in $f'$. This implies that any clique is of size less than $k - |S'| + |S'|$. Since this holds for any subgraph $G'$ within $G$, it follows that there is no clique of size $k$ in $G$, thereby concluding the reduction.

□

**Hardness.** To demonstrate that CSR for a $k$-ensemble of FBDDs is coW[1]-Hard, we will establish a many-to-one FPT reduction from the complementary version of the multi-color clique problem, which is known to be W[1]-Complete.

---

**Multi Color Clique**:
**Input**: A graph $G = \langle V, E \rangle$, such that $V := \langle V_1, \ldots, V_k \rangle$ where each $V_i$ denotes a set of distinct vertices of some color, for which any two vertices associated with a color are not neighbors (for all $i$ there is no edge $(u, v) \in E$ where $u, v \in V_i$).
**Parameter:** $k$ (the number of colors).
**Output**: *Yes*, if there exists a clique of size $k$ in $G$.

---

Let us consider an instance $\langle G, k \rangle$, where $G$ is a multi-colored graph with $k$ distinct colors. We can assume that $|V_1| = |V_2| = \ldots = |V_k| = m$. This assumption is valid because we can take the set with the maximum number of vertices, denoted $maxV$, and "pad" the parts of the graph with fewer than $maxV$ vertices by adding extra vertices that are unconnected (thereby not affecting the size of any potential clique). Consequently, the total number of vertices is $m \cdot k$.

The reduction constructs a model $f$, which is an ensemble of FBDDs, in the following manner: Initially, each vertex is associated with a unique binary string of length $log(m)$, ensuring that vertices of the same color have different strings. We then iterate over pairs of distinct colors, ranging from $1 \le i < j \le k$. For each pair, we create a tree $f_{i,j}$, which is a complete binary tree with a depth of $2log(m)$. This tree comprises features $x_1^i, x_2^i, \ldots, x_{log(m)}^i$ (representing all possible assignments for the tree associated with color $i$) and $x_1^j, x_2^j, \ldots, x_{log(m)}^j$ (representing all possible assignments for the tree associated with color $j$). Each of the $2log(m)$ nodes in the tree corresponds to an assignment of all features of $x^i$ and $x^j$. Each terminal node in the tree, which corresponds to some path, represents a pair of vertices in colors $i$ and $j$. If there is an edge between these two vertices, this terminal node is marked with a 1 (a "True" assignment); if not, it is marked with a 0 (a "False" assignment).

Now, for each constructed tree (totaling $\binom{k}{2}$ trees), we also construct an additional "dummy" tree that consistently returns "False" (assignment 0). Consequently, the total number of trees in the constructed ensemble is $\binom{k}{2} \cdot 2$.

We initially observe that we can assume the existence of at least one pair of vertices $(u, v)$ from different colors $i, j$ that are *not* adjacent (if all pairs were adjacent, there would trivially exist a clique of size $k$). Therefore, we select an assignment where $f_{i,j}$ reaches a "False" (0) terminal node. Arbitrary assignments can be chosen for all other features. Let us denote this complete feature assignment by $\mathbf{x}$. Given that there is at least one tree among the first $\binom{k}{2}$ trees that results in a "False" (0) classification, and all the additional $\binom{k}{2}$ "dummy" trees are designed to reach a "False" (0) outcome, the majority of trees in $f$ will classify $\mathbf{x}$ as 0. Thus, it is established that $f(\mathbf{x}) = 0$. The final structure of the reduction is $\langle f, \mathbf{x}, S := \emptyset \rangle$. Notably, the number of models in $f$ is $2 \cdot \binom{k}{2}$, setting the parameter for the instance $\langle f, \mathbf{x}, S := \emptyset \rangle$ at $k' := 2 \cdot \binom{k}{2}$, which is within the bounds of a computable function $g(k)$, thereby maintaining the FPT reduction.

We first note that this reduction operates in FPT time, as the size of $f$ is capped by $log(n) \cdot O(k^2)$, which is naturally bounded by $O(g(k) \cdot n^k)$ for some computable function $g$. We will now demonstrate that a multi-colored clique of size $k$ or greater exists in $G$ if and only if $S := \emptyset$ is not a sufficient reason with respect to $\langle f, \mathbf{x} \rangle$.

Assume there exists a multi-colored clique of size $k$, denoted as $G'$, within $G$. This implies that for any two vertices $u, v$ in $G'$, $(u, v) \subseteq E$. Given that no edges exist between vertices of the same color, selecting two distinct colors $i, j$ ensures that there is at least one edge connecting a vertex from color $i$ to a vertex from color $j$. Therefore, for each tree $f_{i,j}$ associated with a pair of two different colors $i, j$, we can select the corresponding path that aligns with the edge connecting these two vertices, and which terminates at a "True" (1) leaf node, as per our construction.

We will now prove that when we select these specific paths, any two pairs of distinct paths "match". This occurs due to the following reason: Consider two paths chosen from two distinct trees. Initially,

assume the first tree $f_{i,j}$ is associated with a pair of colors $i, j$ and the second tree $f_{k,l}$ is associated with a pair $k, l$, where $i \neq j \neq k \neq l$. Since the features of the tree $f_{i,j}$ are not shared with those of the tree $f_{k,l}$, any two paths from these trees "match" (they do not conflict over any feature assignment). The more complex scenario arises when two distinct trees, $f_{i,j}$ and $f_{j,k}$, involve the colors $i, j$ and $j, k$, where $i \neq j \neq k$. In this case, the features corresponding to $i, k$ are different, but those associated with color $j$ are shared. However, because a clique of size $k$ in a $k$-partite graph includes exactly one vertex from each color, the vertex corresponding to color $j$ associated with the paths chosen for both $f_{i,j}$ and $f_{j,k}$ is the same. Consequently, the binary string representing the path associated with these features is identical, ensuring these two paths do not conflict over any feature assignment. Thus, in any scenario, any pair of distinct paths selected in this manner for these trees "match".

Therefore, we can assign each of the features in these paths to their respective values within the path. This is feasible because all these paths "match" and have no conflicting assignments. Given that all of these trees are complete, all features can be assigned (i.e., this constitutes a full assignment, not a partial one). We will denote this assignment by $\mathbf{z} \in \mathbb{F}$. As previously noted, and according to our construction, all of these paths terminate at a "True" (1) leaf node. Consequently, the assignment $\mathbf{z}$ results in the first $\binom{k}{2}$ trees receiving a (1) assignment. Since exactly half of the trees in the ensemble are assigned a value of 1, the overall classification by $f$ is 1. In summary, there exists an assignment $\mathbf{z}$ for which:

$$\exists \mathbf{z} \in \mathbb{F} \quad [f(\mathbf{z}) = 1 \neq f(\mathbf{x}) = 0] \tag{63}$$

If no clique of size $k$ exists in $G$, then any subgraph $G'$ of $G$ containing $k$ vertices is *not* a clique. Consider such a subgraph $G'$ with $k$ vertices. Since it is not a clique, this means there must be at least one pair of vertices $(u, v)$ within it that are not connected by an edge.

Consider some assignment $\mathbf{z} \in \mathbb{F}$. When examining the specific path associated with $f_{i,j}(\mathbf{x})$ (for $1 \leq i, j \leq k$), it follows a designated path leading to a specific terminal node of the tree $f_{i,j}$. Additionally, any pair of distinct paths from two distinct trees associated with $\mathbf{z}$ necessarily "match" (since otherwise, they would not correspond to a non-contradicting assignment). Suppose, for the sake of contradiction, that each of these paths ends at a "True" ("1" classification) node.

For each tree $f_{i,j}$, we can identify the pair of vertices $(u, v)$ that correspond to the path representing the binary string of that path. This involves selecting vertices $(u, v)$ associated with two distinct colors $i, j$. Consequently, this specific assignment $\mathbf{z}$ gives rise to a subgraph $G'$, which includes at least one vertex from each color (since every tree $f_{i,j}$ is associated with two vertices —— one for each color). However, we will now prove that there must be *exactly* one vertex from each color in $G'$. Assume, for the sake of contradiction, that there are two vertices $v_1, v_2$ of the same color $i$ in $G'$. Since $v_1, v_2$ are in $G'$, there exist (without loss of generality) two distinct trees $f_{i,j}$ and $f_{i,k}$ where $i \neq k$, and the path associated with $f_{i,j}$ includes vertex $v_1$, while the path associated with $f_{i,k}$ includes vertex $v_2$. This configuration implies that the paths for $f_{i,j}$ and $f_{i,k}$ do not "match" (as the vertex chosen for color $i$ in these two trees differs and is associated with a different binary string).

We have established that $G'$, the graph associated with a specific assignment $\mathbf{z}$, contains exactly one vertex from each color in $G$, and therefore has a size of $k$. From the assumption of this direction in the reduction, this indicates that $G'$ is *not* a clique. Consequently, this also means there must be two vertices $(u, v)$ from two different colors $i \neq j$ where $(u, v) \not\subseteq E$. In terms of our reduction construction, this means that examining $f_{i,j}$, the binary string associated with vertices $u, v$ leads to a "False" (0) terminal node, which contradicts our initial assumption.

Thus, we have demonstrated that if there is no clique in $G$ of size $k$, then for any arbitrary assignment $\mathbf{z}$, not all of the first $\lceil \binom{k}{2} \rceil$ trees in $f$ receive a 1 assignment. In simpler terms, at least one tree receives a 0 assignment. Consequently, in the ensemble, any assignment results in at least $\lceil \binom{k}{2} \rceil + 1$ trees being assigned 0, ensuring that the ensemble classification is always 0. In other words:

$$\forall \mathbf{z} \in \mathbb{F} \quad [f(\mathbf{z}) = 0 = f(\mathbf{x})] \tag{64}$$

This indicates that $S := \emptyset$ is a sufficient reason with respect to $\langle f, \mathbf{x} \rangle$, thereby concluding our reduction.

$\square$

**Lemma 14.** *The CC query for a $k$-ensemble of FBDDs is #W[1]-Complete.*

*Proof.* We note that the CC query is the counting version for the CSR query, for which we already proved coW[1]-Completeness. The proofs of membership and hardness where from the $k$-Clique, and $k$-Multicolored-Clique problems. It is well known that the counting version of $k$-Clique is #W[1]-Complete Flum & Grohe (2004). Moreover, there exist FPT reductions to and from $k$ Multi-Color Clique to $k$-Clique, which shows us that Multicolored clique is also #W[1]-Complete. Finally, we arrive at that the CC query for a $k$ ensemble of FBDDs is #W[1]-Complete.

□

**Lemma 15.** *The MCR query for a $k$-ensemble of FBDDs is W[1]-Hard and in W[P].*

*Proof.* **Hardness**. Specifically, the hardness results are consistent with those presented by Ordyniak et al. (2024). However, we can also directly prove hardness by presenting a reduction from the previous (complement of) the *CSR* problem for ensembles of FBDDs, which we have shown to be coW[1]-Complete (via FPT reductions from $k$-Clique and $k$-Multicolored Clique). The complement of the *CSR* problem is equivalent to validating whether, given some $f, \mathbf{x}$, and a subset $S$, it can be checked whether $S$ is *not* a sufficient reason for $\langle f, \mathbf{x} \rangle$. This is equivalent to checking whether $\overline{S}$ is a contrastive reason for $\langle f, \mathbf{x} \rangle$.

Hence, given an instance $\langle f, \mathbf{x}, S \rangle$, we can apply Lemma 8, which states that ensembles of FBDDs are closed underr conditioning. We will condition $f$ on $\mathbf{x}_S$ to construct $f'$. The resulting model $f'$ will have $|\overline{S}|$ features, and it holds that:

$$\forall \mathbf{z} \in \mathbb{F} \quad [f'(\mathbf{z}_{\bar{S}}) = f(\mathbf{x}_S; \mathbf{z}_{\bar{S}})] \tag{65}$$

Now, given the instance $\langle f, \mathbf{x}, S \rangle$, the reduction will construct: $\langle f', \mathbf{x}, k := |\overline{S}| \rangle$. If $S$ is not a sufficient reason for $\langle f, \mathbf{x} \rangle$, this implies that $\overline{S}$ is a contrastive reason for $\langle f, \mathbf{x} \rangle$, and therefore:

$$\exists \mathbf{z} \in \mathbb{F} \quad [f(\mathbf{x}_S; \mathbf{z}_{\bar{S}}) \neq f(\mathbf{x})] \tag{66}$$

This further implies that:

$$\exists \mathbf{z} \in \mathbb{F} \quad [f'(\mathbf{z}_{\bar{S}}) = f(\mathbf{x}_S; \mathbf{z}_{\bar{S}}) \neq f(\mathbf{x}) = f'(\mathbf{x}_{\bar{S}})] \tag{67}$$

which indicates that $\overline{S}$ is a contrastive reason for $\langle f', \mathbf{x} \rangle$. Therefore, there exists a contrastive reason of size $|\overline{S}|$ for $\langle f', \mathbf{x} \rangle$. If we assume that $S$ is a sufficient reason for $\langle f, \mathbf{x} \rangle$, then it holds that:

$$\forall \mathbf{z} \in \mathbb{F} \quad [f(\mathbf{x}) = f(\mathbf{x}_S; \mathbf{z}_{\bar{S}})] \iff$$
$$\forall \mathbf{z} \in \mathbb{F} \quad [f'(\mathbf{x}_{\bar{S}}) = f(\mathbf{x}) = f(\mathbf{x}_S; \mathbf{z}_{\bar{S}}) = f'(\mathbf{z}_{\bar{S}})] \tag{68}$$

This indicates that $\overline{S}$ is not a contrastive reason for $\langle f', \mathbf{x} \rangle$, and that any subset $S' \subseteq S$ is also not a contrastive reason for $\langle f', \mathbf{x} \rangle$. Hence, it follows that there is no contrastive reason of size $|\overline{S}|$ or smaller for $\langle f', \mathbf{x} \rangle$. This completes the reduction, thereby proving that the *MCR* query for an ensemble of $k$ FBDDs is W[1]-Hard.

**Membership.** We will prove membership in W[P] by reducing the MCR query for $k$-ensemble FBDDs to the WCS[$C_{d,t}$] problem, as described in Section B. Given an instance $\langle f, \mathbf{x}, D \rangle$, where $D$ represents the size of the contrastive reason we are looking for, we construct a Boolean circuit $C$. Although the weft of $C$ is 2 (and could be reduced to 1 with a more refined construction), its depth will depend on a parameter $D$ and will not be bounded by a constant.

Our reduction begins by creating a modified model $f'$ based on the original model $f$ and the input $\mathbf{x}$. Essentially, $f$ and $f'$ will maintain the same structure; however, the vector $\mathbf{1}_n$ (consisting solely of 1s) in $f'$ will correspond to the vector $\mathbf{x}$ in $f$, and conversely, the vector $\neg\mathbf{x}$ in $f$ will correspond to the vector $\mathbf{0}_n$ (consisting solely of 0s) in $f'$.

To carry out this construction, we start by replicating $f$ to create $f'$. For each FBDD $f_i$ in the ensemble $f$, we examine every node $v$ within $f_i$. For each node assignment $v_i \in \{0, 1\}$ where

$v_i \neq \mathbf{x}_i$, we reverse the 0 and 1 assignments, with the typical convention that 1 represents the right branch and 0 the left branch. This flipping will be done such that 1 will now correspond to the left branch. If $v_i = \mathbf{x}_i$, we retain the original order. This process is repeated across all paths in each FBDD of the ensemble. As a result, we generate a new model $f'$ where each assignment to a value of $\mathbf{x}_i$ in $f$ corresponds to an assignment to 1 in $f'$. If $f(\mathbf{x}) = 1$, we can apply the negation principle using Lemma 9 to negate $f'$, and if $f(\mathbf{x}) = 0$, we can leave it as is. Consequently, any vector $\mathbf{z} \in \mathbb{F}$ where $f(\mathbf{z}) \neq f(\mathbf{x})$ translates to a vector $\mathbf{z}' \in \mathbb{F}$ for which $f'(\mathbf{z}) \neq 1$. Therefore, the problem of determining whether there exists a subset $S$ of size $D$ such that $f(\mathbf{x}_{\bar{S}}; \mathbf{z}_S) \neq f(\mathbf{x})$ equates to finding a vector $\mathbf{z}' \in \mathbb{F}$ with $D$ assignments to 1 (i.e., of Hamming weight $D$) where $f'(\mathbf{z}') \neq 1$.

Now, with $f'$ in place, we will develop a Boolean circuit $C$, as outlined earlier. Specifically, we will create a Boolean circuit $C$ with a weft of 2 and arbitrary depth, designed to return True if $f'$ has a positive assignment of Hamming weight $D$, and False otherwise. The construction will proceed through the following steps:

1. In our ensemble of $k$ trees, for each leaf $v$ in every tree $f'_i$, we will designate $y_{\{v, f'_i\}}$ as an input node in the circuit. We will consider two input nodes, $y_{\{v, f'_i\}}$ and $y_{\{v', f'_j\}}$, as inconsistent if $i = j$ and $v \neq v'$. In such cases, we will introduce a node: $[\neg y_{\{v, f'_i\}}] \vee [\neg y_{\{v', f'_j\}}]$, which is equivalent to $\neg[y_{\{v, f'_i\}} \wedge y_{\{v', f'_i\}}]$, where both $y_{\{v, f'_i\}}$ and $y_{\{v', f'_i\}}$ are input nodes. This setup essentially encodes the $k$-*Clique* problem, except for the final AND encoding involving all input nodes, which we will address in the last step. Recalling our proof for the *CSR* query, the reduction to the $k$-*Clique* problem assists in determining whether a positive assignment exists for $f'$. We are now tasked with a more challenging problem: determining whether there is a positive assignment to $f'$ with a Hamming weight of $d$. This necessitates the inclusion of additional constraints.

2. To this circuit $C$, currently encoding the Clique problem, we will introduce more nodes. For each feature $i \in [n]$, we will create a new node $u_i$ functioning as an OR gate. This node will take inputs from any $y_{\{v, f'_j\}}$ where the assignment represented by the leaf $v$ assigns "True" (i.e., a 1 assignment) to feature $i$.

3. For the final component, we will add another layer to our circuit. For every $1 \leq j \leq n$ and for every $0 \leq d' \leq D$, we define a variable $u_{\{j, d'\}}$. This variable is configured to be set to True if and only if exactly $d'$ of the features $1, \ldots, j$ are set to True. Specifically, $u_{\{1, 0\}}$ will take $\neg u_1$ as its input, and $u_{\{1, 1\}}$ will take $u_1$ as its input. All other $u_{\{1, d'\}}$ variables are set to False. For $j > 1$, we construct $u_{\{j, d'\}}$ to take the input:

$$[\, u_{\{j-1, d'-1\}} \wedge u_j \,] \vee [\, u_{\{j-1, d'-1\}} \wedge \neg u_j \,] \tag{69}$$

4. Finally, the output of the circuit is derived from a large AND node that collects inputs from all nodes established in step 2 and those from step 4 of the form $u_{n, d'}$, where $d'$ ranges from 0 to $D$.

Since that circuit $C$ effectively encodes a scenario where $f'$ receives a positive assignment of Hamming weight $D$, and considering that $C$ possesses a weft of 2 and arbitrary depth, we conclude that the *MCR* query for ensembles of FBDDs is in W[P].

$\square$

**The W[1]-W[P] gap for the MCR query.** We note that while we have proven the *MCR* query for ensembles of FBDDs is W[1]-Hard and belongs to W[P], the exact complexity class for which this problem is complete remains unknown. Unfortunately, the current definition of the W-hierarchy is not well-suited to capture the complexity of this problem. Specifically, containment in W[t] for a fixed constant $t$ means that the problem instance at hand should be encoded using a Boolean circuit of constant depth (as well as weft $t$), so that the instance at hand is a yes-instance if and only if the circuit has a satisfying assignment having exactly $k$ variables assigned true, where $k$ is the parameter (or $k$ is bounded by a function of the parameter) of the instance at hand. We refer to Chapter 13.3 in Cygan et al. (2015) for more information. However, our problem involves a "weight measure" $d$ (the Hamming distance) that may be arbitrarily larger than $k$. In particular, encoding that a potential solution to the instance at hand assigns true to at most $d$ variables cannot be done with a constant-depth circuit, as this requires the implementation of a counter.

The problem described above is a fundamental one in the definition of the W-hierarchy in the context of *weighted problems in general*, where the desired total weight of the solution is not bounded by the parameter. Indeed, it is not known how to classify basic W[1]-hard problems in the field such as Weighted $k$-Clique, where given a vertex-weighted graph, along with integers $k$ and $w$, we seek a clique of size exactly $k$ and weight at least/at most $W$, and the parameter is $k$. The same situation holds for problems where the "weight measure" is implicit similarly to our problem, such as the Partial Vertex Cover problem, where given a (non-weighted) graph $G$ and integers $k$ and $t$, we seek a set of vertices of size exactly $k$ that altogether cover at least $t$ edges. From personal communications with other researchers in the field, we have gathered that the definition and study of a "Weighted W-Hierarchy" can be a topic of independent interest in parameterized complexity, but this is outside the scope of this paper.

**Lemma 16.** *The SHAP query for a $k$-ensemble of FBDDs is #W[1]-Hard, and in XP.*

*proof.* **Hardness.** We begin by proving that this problem is #W[1]-Hard. Our approach follows the proof outlined by (Arenas et al. (2023)), which demonstrates a link between the computation of Shapley values and the model counting problem. First, we provide a definition of the model counting problem. Given a model $f : \{0,1\}^n \to \{0,1\}$, we denote $\#f$ as the number of assignments where $f$ outputs 1. In other words:

$$\#f := |\{\mathbf{z} \in \mathbb{F}, f(\mathbf{z}) = 1\}| \tag{70}$$

Now, the work of (Arenas et al. (2023)) demonstrates the following relationship when the feature distribution $\mathcal{D}_p$ is assumed to be the *uniform distribution*, which is a specific case of the fully factorized distribution considered in this paper. For any $f, \mathbf{x}$, it holds that:

$$\#f := f(\mathbf{x}) - 2^n \cdot \sum_{i \in \{1,\ldots,n\}} \phi_i(f, \mathbf{x}) \tag{71}$$

This provides direct proof that computing the Shapley value under a uniform distribution (and consequently for the fully factorized distribution) is as difficult as the model counting problem. It remains to be shown that the model counting problem for an ensemble of $k$ FBDDs is #W[1]-Complete, when parameterized by $k$. Since we have already established that the $CC$ query for a $k$-ensemble of FBDDs is #W[1]-Complete, we can demonstrate an FPT many-one reduction from the model counting problem to the computation of the $CC$ query.

Given a model $f$, we start by selecting an arbitrary vector $\mathbf{x} \in \mathbb{F}$. If $f(\mathbf{x}) = 1$, the reduction calculates the count for $\langle f, \mathbf{x}, S := \emptyset \rangle$, which we denote as $\#CC(f, \mathbf{x}, S)$. If $f(\mathbf{x}) = 0$, the reduction computes $2^n - \#CC(f, \mathbf{x}, S)$. We note that $2^n$ can be computed in polynomial time, as $n$ (representing the number of input assignments) is provided in unary.

The completion count seeks the number of assignments where the complement of $S$ (which, in this case, includes all possible assignments) results in a classification of 1. Thus, if $f(\mathbf{x}) = 1$, we have $\#f = \#CC(f, \mathbf{x}, S)$, and if $f(\mathbf{x}) = 0$, we have $\#f = 2^n - \#CC(f, \mathbf{x}, S)$, concluding the reduction.

**Membership.** We will prove membership in *XP* by presenting an algorithm that computes Shapley values for ensembles of $k$-FBDDs in $O(|\mathcal{X}|^k)$ time. We utilize the following relation, established by (Van den Broeck et al. (2022)), concerning Shapley values under conditional expectations and assuming feature independence. The following relation has been proven:

$$\text{SHAP}(f, i, \mathcal{D}_p, \mathbf{x}) =_P \mathbb{E}_{\mathbf{z} \sim \mathcal{D}_p}[f(\mathbf{z})] \tag{72}$$

In other words — the computational complexity of obtaining a Shapley value under this formalization is equivalent (under polynomial reductions) to the complexity of computing $\mathbb{E}_{\mathbf{z} \sim \mathcal{D}_p}[f(\mathbf{z})]$. This means we can focus on determining the complexity of obtaining $\mathbb{E}\mathbf{z} \sim \mathcal{D}_p[f(\mathbf{z})]$, which can sometimes be easier to handle.

We now present the following algorithm for computing $\mathbb{E}_{\mathbf{z} \sim \mathcal{D}_p}[f(\mathbf{z})]$ for ensembles of FBDDs in $O(|\mathcal{X}|^k)$ time. The algorithm iterates over every combination of selecting one path from each tree

in the ensemble $f$. We assume that $j$ of these selected paths correspond to trees that classify as "1", while the remaining $k - j$ trees classify as "0". We first check whether this selection of $j$ trees $f'_1, \ldots, f'_j$ that classify as "1" can result in a positive classification for $f$. For unweighted majority voting, this is equivalent to verifying whether $j \geq \lceil \frac{k}{2} \rceil$, and for weighted voting, it involves checking whether the weights $\phi_i$ associated with each tree $f'i$ satisfy $\sum 1 \leq i \leq j\phi_i > 0$.

We can then verify whether each pair of paths chosen from the set of paths across all trees satisfies that any two paths "match" (i.e., they do not contain any features with conflicting assignments). For each combination of trees, we examine the partial assignment to all features involved in these paths (there must be only one such assignment since the paths "match"). In practice, this is equivalent to iterating over all possible positive assignments of the ensemble $f$. However, we note that a single iteration over a selection of paths in each tree may yield only a partial assignment to some features, denoted $\mathbf{z}_{S'}$ for some $S' \subseteq [n]$. As a result, this assignment corresponds to any assignment of the form $(\mathbf{z}_{S'}; \mathbf{z}'_{\bar{S}'})$ for any $\mathbf{z}' \in \mathbb{F}$ (all of which are also classified as "1").

We note that this entire process is bounded by $O(m^k)$, where $m$ represents the maximum number of leaves in a tree and $k$ is the number of trees. This is also bounded by $O(|\mathcal{X}|^{O(k)})$. We now proceed to prove the following claim:

**Claim 13.** *For two distinct partial assignments $\mathbf{z}_S$ and $\mathbf{z}'_{S'}$, obtained by iterating through the aforementioned procedure of selecting $j$ paths in $j$ distinct trees, it holds that there exists a feature $i \in S, S'$ such that $z_i \neq z'_i$.*

*Proof.* Since $\mathbf{z}_S$ and $\mathbf{z}'S'$ are selected from iterating over two distinct choices of paths from trees, there must be at least one tree where the paths chosen for $\mathbf{z}_S$ and $\mathbf{z}'S'$ differ. Any two distinct paths in a tree contain at least one feature with differing assignments for some feature $i$. Therefore, based on the previous construction, it follows that there exists at least one feature $i$ where the partial assignments of $\mathbf{z}_S$ and $\mathbf{z}'_{S'}$ differ.

By definition, the following sum holds:

$$\mathbb{E}_{\mathbf{z} \sim \mathcal{D}_p}[f(\mathbf{z})] = \sum_{\mathbf{z} \in \mathbb{F}, f(\mathbf{z})=1} \mathcal{D}_p(\mathbf{z}) = \sum_{\mathbf{z} \in \mathbb{F}, f(\mathbf{z})=1} \left( \prod_{i \in [n], \mathbf{z}_i=1} p(i) \right) \cdot \left( \prod_{i \in [n], \mathbf{z}_i=0} (1 - p(i)) \right) \tag{73}$$

That is, computing the expectation involves summing each $\mathcal{D}p(\mathbf{z})$ for every positive assignment. Now, assume we have some partial assignment $\mathbf{z}_S$ obtained in the previous phase. Since this is only a partial assignment, we need to account for all possible completions of $\mathbf{z}_S$, or, in other words, any vector of the form $(\mathbf{z}_S; \mathbf{z}'\bar{S})$. Let the set of all partial assignments computed using the aforementioned procedure be denoted by $\mathbb{S}$. Since any two partial assignments have a conflicting feature that does not "match", there is no "overlap" in the assignment completions corresponding to two distinct partial assignments. In other words, for two distinct partial assignments $\mathbf{z}_S$ and $\mathbf{z}'S'$, it holds that for all $\mathbf{z}'' \in \mathbb{F}$, $(\mathbf{z}_S; \mathbf{z}''\bar{S}) \neq (\mathbf{z}'S'; \mathbf{z}''\bar{S}')$. This leads to the following equivalence:

$$\mathbb{E}_{\mathbf{z} \sim \mathcal{D}_p}[f(\mathbf{z})] = \sum_{\mathbf{z}_S \in \mathbb{S}} \left( \sum_{\mathbf{z}'_{\bar{S}} \in \{0,1\}^{|\bar{S}|}} \mathcal{D}_p(\mathbf{z}_S; \mathbf{z}'_{\bar{S}}) \right) \tag{74}$$

We do note, however, that for a fixed partial assignment $\mathbf{z}_S$, the following holds:

$$\sum_{\mathbf{z}'_{\bar{S}} \in \{0,1\}^{|\bar{S}|}} \mathcal{D}_p(\mathbf{z}_S; \mathbf{z}'_{\bar{S}}) = \sum_{\mathbf{z}'_{\bar{S}} \in \{0,1\}^{|\bar{S}|}} \left( \prod_{i \in [n], (\mathbf{z}_S; \mathbf{z}'_{\bar{S}})_i=1} p(i) \right) \cdot \left( \prod_{i \in [n], (\mathbf{z}_S; \mathbf{z}'_{\bar{S}})_i=0} (1 - p(i)) \right) =$$

$$\sum_{\mathbf{z}'_{\bar{S}} \in \{0,1\}^{|\bar{S}|}} \left( \prod_{i \in S, \mathbf{z}_i=1} p(i) \right) \left( \prod_{i \in S, \mathbf{z}_i=0} (1 - p(i)) \right) \left( \prod_{i \in \bar{S}, \mathbf{z}'_i=1} p(i) \right) \left( \prod_{i \in \bar{S}, \mathbf{z}'_i=0} (1 - p(i)) \right) = \tag{75}$$

$$\left( \prod_{i \in S, \mathbf{z}_i=1} p(i) \right) \cdot \left( \prod_{i \in S, \mathbf{z}_i=0} (1 - p(i)) \right)$$

Overall, we obtain that:

$$\mathbb{E}_{\mathbf{z}\sim\mathcal{D}_p}[f(\mathbf{z})] = \sum_{\mathbf{z}_S\in\mathbb{S}} \Big( \prod_{i\in S, \mathbf{z}_i=1} p(i) \Big) \cdot \Big( \prod_{i\in S, \mathbf{z}_i=0} (1-p(i)) \Big) \tag{76}$$

Hence, after computing $\mathbb{S}$ in $O(|\mathcal{X}|^{O(k)})$ time, we can iterate over each partial assignment, compute its corresponding weight, and sum all the weights to obtain $\mathbb{E}_{\mathbf{z}\sim\mathcal{D}_p}[f(\mathbf{z})]$. This proves that the complexity of computing $\mathbb{E}\mathbf{z}\sim\mathcal{D}_p[f(\mathbf{z})]$ is in XP.

$\square$

As explained earlier, this also establishes that the complexity of computing SHAP for some $\langle f, \mathbf{x}, i, \mathcal{D}_p\rangle$ is likewise in XP.

$\square$

We have demonstrated that solving *SHAP* when $\mathcal{D}_p$ represents any fully factorized distribution is #W[1]-Hard and in XP. However, if we specifically set $\mathcal{D}_p$ to the *uniform* distribution (i.e., where for any $i\in[n]$, $p(i) = \frac{1}{2}$), a specific type of fully factorized distribution, then this query becomes #W[1]-Complete.

**Lemma 17.** *Assuming that $\mathcal{D}_p$ is the uniform distribution, then the SHAP query for ensembles of $k$-FBDDs is #W[1]-Complete.*

*Proof.* The hardness is derived directly from our proof concerning fully factorized distributions, assuming a uniform distribution. For membership, we leverage the analysis by (Arenas et al. (2021c)), which demonstrates that when $\mathcal{D}_p$ is uniform, computing *SHAP* can be polynomially reduced to the problem of model counting (i.e., computing $\#f$). As outlined in Proposition 14, the task of model counting for a $k$-ensemble of FBDDs is $\#W[1]$-Complete. Thus, we conclude that computing *SHAP* under the uniform distribution for $k$-ensembles of FBDDs also achieves #W[1]-Complete status.

$\square$

## J   PROOF OF PROPOSITION 6

**Proposition 6.** *The MSR query for a $k$-ensemble of FBDDs is para-NP-Hard and ix in XNP.*

*Proof Sketch.* To prove membership, we devise an algorithm that initially computes all minimal contrastive reasons during a preprocessing phase in $O(|\mathcal{X}|^k)$ time. In the second phase, the algorithm leverages the Minimal-Hitting-Set (MHS) duality between sufficient and contrastive reasons (Ignatiev et al. (2020)), allowing the algorithm to non-deterministically identify the MHS among all minimal contrastive reasons.

*Proof.* **Hardness.** Para-NP hardness is straightforward since the *MSR* query for a single FBDD is already NP-Hard Barceló et al. (2020) and hence is obtained for $k = 1$.

**Membership.** For membership in XNP we need to show that there exists a non-deterministic algorithm that solves this problem in $O(|\mathcal{X}|^k)$ time. Specifically, we will make use of the minimum-hitting-set (MHS) duality between sufficient and contrastive reasons to prove this claim Ignatiev et al. (2020). First, we will define the MHS:

**Definition 6.** *Given a collection $\mathbb{S}$ of sets from a universe U, a hitting set $h$ for $\mathbb{S}$ is a set such that $\forall S\in\mathbb{S}, h\cap S\neq\emptyset$. A hitting set $h$ is said to be* minimum *when it has the smallest possible cardinality among ll hitting sets.*

We note that a subset minimal contrastive (sufficient) reason $S$ of $\langle f, \mathbf{x}\rangle$ is a contrastive (sufficient) reason that ceases to be a contrastive (Sufficient reason) when any feature $i$ is removef from it. In other words, for all $i$, it holds that $S\setminus i$ is not sufficient (contrastive). We now are in a position to use the following MHS duality between sufficient and contrastive reasons Ignatiev et al. (2020):

**Lemma 18.** *The MHS of all subset minimal contrastive reasons with respect to $\langle f, \boldsymbol{x}\rangle$ is a cardinally minimal sufficient reason of $\langle f, \boldsymbol{x}\rangle$. Moreover, the MHS of all subset minimal sufficient reasons with respect to $\langle f, \boldsymbol{x}\rangle$ is a cardinally minimal contrastive reason of $\langle f, \boldsymbol{x}\rangle$.*

Now, we describe the following preprocessing stage which runs in time $O(|\mathcal{X}|^k)$ and computes all of the subset minimal contrastive reasons of $\langle f, \mathbf{x}\rangle$.

**Claim 14.** *Given an ensemble of FBDDs $f$ with $k$ FBDDs, there exists an algorithm that computes all of the subset minimal contrastive reasons of $\langle f, \boldsymbol{x} \rangle$ in time $O(m^k)$, where $m$ denotes the maximal number of leaf nodes in an FBDD within $f$.*

*Proof.* We iterate over combinations of choosing one leaf (which corresponds to one path) from every distinct tree in $f$. This process can be done in $O(m^k)$ time. We check whether two conditions hold. First, we check whether every pair of two distinct paths that belong to two distinct trees "matches" and whether more than $\lceil \frac{k}{2} \rceil$ of these trees terminate over a leaf with a $\neg f(\mathbf{x})$ assignment. For any combination of $k$ distinct paths in which any two paths "match", it means that there is some partial assignment $\mathbf{z}_{S'}$, which describes the corresponding assignments in each one of these paths (there is necessarily one such assignment since each two paths "match") and it holds that:

$$\forall \mathbf{y} \in \mathbb{F} \quad [f'(\mathbf{z}_{S'}; \mathbf{y}_{\bar{S}'}) \neq f(\mathbf{x})] \tag{77}$$

We now will denote by $S''$, the subset of features in $\mathbf{z}_{S'}$ that do not match with $\mathbf{x}$. More specifically:

$$S'' := |\{i \in \{1, \ldots S'\} \ \mathbf{z}_i \neq \mathbf{x}_i| \tag{78}$$

For each combination of $k$ paths, for which the corresponding two conditions hold, we compute $S''$. We denote the set of all such subsets as $\mathbb{S}$. We will now prove the following claim, which will finish the proof of our lemma:

**Claim 15.** *Any subset minimal contrastive reason of $\langle f, \boldsymbol{x} \rangle$ is contained in $\mathbb{S}$*

*Proof.* Let us assume towards contradiction that this claim does not hold. In other words, there exists a subset $S$ which is a subset minimal contrastive reason of $\langle f, \mathbf{x} \rangle$ and that is not chosen to be in $\mathbb{S}$ by our algorithm. Since $S$ is a contrastive reason it holds that:

$$\exists \mathbf{z} \in \mathbb{F} \quad [f(\mathbf{x}_{\bar{S}}; \mathbf{z}_S) \neq f(\mathbf{x})] \tag{79}$$

However, we will prove a stronger property that holds if $S$ is a *subset minimal* contrastive reason of $\langle f, \mathbf{x} \rangle$. Specifically, it holds that:

$$[f(\mathbf{x}_{\bar{S}}; \neg\mathbf{x}_S) \neq f(\mathbf{x})] \tag{80}$$

or in other words the specific vector $\mathbf{z}$ for which fixing $S$ to, changes the classification is $\neg\mathbf{x}$. The proof to this claim is straightforward — let us assume towards contradiction that this is not the case, or in other words, it holds that:

$$[f(\mathbf{x}_{\bar{S}}; \neg\mathbf{x}_S) = f(\mathbf{x})] \tag{81}$$

Since $S$ is a contrastive reason, this means that there exists some other assignment to the features of $S$ (which is not $\neg\mathbf{x}$, which causes the classification to change. In other words, there exists some $\mathbf{y} \neq \neg\mathbf{x}$ for which:

$$[f(\mathbf{x}_{\bar{S}}; \mathbf{y}_S) \neq f(\mathbf{x})] \tag{82}$$

Since $\mathbf{y} \neq \neg\mathbf{x}$ over the features in $S$, this means that there is at least one feature $i \in S$ for which $\mathbf{y}_i = \mathbf{x}_i$. Let us now denote $S_0 := S \setminus \{i\}$. We hence get that:

$$[f(\mathbf{x}_{\bar{S}}; \mathbf{y}_S) = f(\mathbf{x}_{\bar{S_0}}; \neg\mathbf{x}_{S_0}) \neq f(\mathbf{x})] \tag{83}$$

This implies that $S_0$ is a contrastive reason of $\langle f, \mathbf{x} \rangle$, hence contradicting the subset minimality of $S$. This concludes the proof of this claim, and we hence derive in the fact that since $S$ is a subset minimal contrastive reason of $\langle f, \mathbf{x} \rangle$, it must hold that:

$$[f(\mathbf{x}_{\bar{S}}; \neg\mathbf{x}_S) \neq f(\mathbf{x})] \tag{84}$$

We hence can take the assignment $(\mathbf{x}_{\bar{S}}; \neg\mathbf{x}_S)$ and propagate it through $f$. The assignment $f_i(\mathbf{x}_{\bar{S}}; \neg\mathbf{x}_S)$ will lead to some path, for which there are at least $\lceil \frac{k}{2} \rceil$ which terminate on a $\neg f(\mathbf{x})$ node. These paths, of course "match" (meaning any pair of two distinct paths "match"), because they correspond to *one* assignment of features, and hence will be chosen as a combination of paths by our algorithm.

We recall that our algorithm chooses the partial assignment which assigns a value to each one of the features in each path of each tree (there exists one such assignment). In our case this will be some partial assignment of $(\mathbf{x}_{\bar{S}}; \neg\mathbf{x}_S)$. The algorithm then chooses the subset $S''$ which is the subset of features whose values (in our case of $(\mathbf{x}_{\bar{S}}; \neg\mathbf{x}_S)$) that are different than those of $\mathbf{x}$. Hence, it necessarily holds that $S'' \subseteq S$.

We note that $S''$ is a contrastive reason concerning $\langle f, \mathbf{x} \rangle$, since by construction, it describes a subset of features such that if we set them to values that are not of $\mathbf{x}$ (specifically $\neg\mathbf{x}$), $\lceil \frac{k}{2} \rceil$ or more trees in the ensemble terminate on a $\neg f_i(\mathbf{x})$ assignment. Since we have proven that $S'' \subseteq S$, then it must hold that $S'' = S$, since otherwise, this will contradict the subset minimality of $S$. We hence derive in the fact that $S'' = S$ is chosen during our algorithm to be in $\mathbb{S}$, contradicting the assumption that $S$ is not in $\mathbb{S}$. This concludes the proof that any subset minimal contrastive reason of $\langle f, \mathbf{x} \rangle$ is in $\mathbb{S}$.

$\square$

We will now use the MHS duality to conclude our proof regarding membership in XNP. We recall that if $\mathbb{S}$ is the set containing all subset minimal contrastive reasons of $\langle f, \mathbf{x} \rangle$, then the MHS of $\mathbb{S}$ is the cardinally minimal sufficient reason of $\langle f, \mathbf{x} \rangle$. We note that if we add to $\mathbb{S}$ (which contains all subset minimal contrastive reasons of $\langle f, \mathbf{x} \rangle$ other (non-subset-minimal) contrastive reasons of $\langle f, \mathbf{x} \rangle$ the MHS of $\mathbb{S}$ will remain the same. This is true since the hitting set of any two subsets $S$ and $S'$, when $S \subseteq S'$, is equal to $S$.

We have proven that the set $\mathbb{S}$ that is obtained by our algorithm contains all subset minimal contrastive reasons of $\langle f, \mathbf{x} \rangle$, as well as perhaps other contrastive reasons. Hence, the MHS of $\mathbb{S}$ is a cardinally minimal sufficient reason of $\langle f, \mathbf{x} \rangle$.

Hence, we can simply non-deterministically guess some subset $S_1 \subseteq \{1, \ldots, n\}$, and check whether $S_1$ intersects with all subsets in $\mathbb{S}$ (and hence is the MHS of $\mathbb{S}$, and a cardinally minimal sufficient reason of $\langle f, \mathbf{x} \rangle$. If $|S_1| \leq d$, our algorithm can return true, and otherwise, it will return false. Overall, the entire algorithm that we described performs a preprocessing step in $O(m^k)$ time (and hence is bounded by $O(|\mathcal{X}|^k)$ time), and then performs a non-deterministic step, also bounded by $O(|\mathcal{X}|^k)$ time. This concludes the proof that solving the MHS query for an ensemble of $k$ FBDDs is contained in XNP.

$\square$

# K    PROOF OF PROPOSITION 7

**Proposition 7.** *Obtaining a subset-minimal sufficient reason for a $k$-ensemble of FBDDs is in XP.*

*Proof.* The specific result aligns with a result demonstrated by Ordyniak et al. (2023). However, we can further establish this result as a direct extension of the complexity result for the CSR query (Lemma 13). To show this, we use a common greedy algorithm that computes a subset-minimal sufficient reason by invoking a linear number of queries, each checking whether a given subset is a sufficient reason Ignatiev et al. (2019b). Intuitively, the algorithm attempts to "free" a feature from the subset $S$ at each iteration, until finally converging to a subset-minimal sufficient reason.

Lemma 13 proved that the *CSR* query for an ensemble of FBDDs is in coW[1] (and hence is also in XP). In essesnse, algorithm K implies that a linear number of queries to the *CSR* query produces a subset-minimal sufficient reason. It hence, directly follows that obtaining some subset minimal sufficient reason for some $\langle f, \mathbf{x} \rangle$ is also in XP.

$\square$

---

**Algorithm 2** Greedy Subset Minimal Sufficient Reason Search

---

**Input** $f$, $\mathbf{x}$

1: $S \leftarrow \{1, \ldots, n\}$
2: **for each** $i \in \{1, ..., n\}$ by some arbitrary ordering **do**
3:     **if** $S \setminus \{i\}$ is a sufficient reason w.r.t $\langle f, \mathbf{x} \rangle$ **then**
4:         $S \leftarrow S \setminus \{i\}$
5:     **end if**
6: **end for**
7: **return** $S$               $\triangleright$ $S$ is a *subset minimal* sufficient reason

---

## L   PROOF OF PROPOSITION 8

**Proposition 8.** *$k$-ensemble-Perceptrons are (i) para-coNP Complete with respect to CSR, (ii) para-NP-Complete with respect to MCR, (iii) para-$\Sigma_2^P$-Complete with respect to MSR, and (iv) para-#P-Hard with respect to CC and SHAP.*

*Proof.* All *membership* results are a direct result from the non-parameterized complexity results 1, and from the reasons described under the proof of Proposition 4. We now will prove each hardness result seperattley.

**Lemma 19.** *The CSR query for a $k$-ensemble of Perceptrons is para-coNP Hard.*

*Proof.* We will equivalently prove that the *CSR* query for an ensemble containing only 2 perceptrons is coNP Complete. We will reduce from the *Subset-sum problem* (*SSP*), which is a classic NP-Complete problem.

---

**SSP (Subset Sum)**:
**Input**: $(z_1, z_2, \ldots, z_n)$ set of positive integers and a positive (target) integer $T$
**Output**: *Yes*, if there exists a subset $S \subseteq (1, 2, \ldots, n)$ such that $\sum_{i \in S} z_i = T$, and *No* otherwise.

---

We reduce $CSR$ for an ensemble with $k = 2$ of Perceptrons from $\overline{SSP}$. Given some $\langle (z_1, z_2, \ldots, z_n), T \rangle$, the reduction constructs the two following Perceptrons $f_1 := \langle \mathbf{w^1}, b_1 \rangle$ and $f_2 := \langle \mathbf{w^2}, b_2 \rangle$, where $\mathbf{w^1} := (-z_1, -z_2, \ldots, -z_n)$, $b_1 := T - \frac{1}{2}$, $\mathbf{w^2} := (z_1, z_2, \ldots, z_n)$, and $b_2 := -T - \frac{1}{2}$. The reduction constructs $\langle f := (f_1, f_2), \mathbf{x} := \mathbf{0}_n, S := \emptyset \rangle$, and $\mathbf{0}_n$ denotes a vector of size $n$ where all values are set to 0.

First, we notice that:

$$f_1(\mathbf{x}) = f_1(\mathbf{0}_n) = T - \frac{1}{2} > 0 \; \wedge$$
$$f_2(\mathbf{x}) = f_2(\mathbf{0}_n) = -T - \frac{1}{2} < 0 \tag{85}$$

Since $f_1(\mathbf{x})$ is positive and $f_2(\mathbf{x})$ is negative, then the ensemble $f = \langle f_1, f_2 \rangle$ returns 1 for the input $\mathbf{x}$ (is positive), by definition.

If $\langle (z_1, z_2, \ldots, z_n), T \rangle \in \overline{SSP}$ then there does not exist a subset of features $S' \subseteq (1, 2, \ldots, n)$ such that $\sum_{i \in S'} z_i = T$. Since these are *integers*, this implies that any subset $S' \subseteq (1, 2, \ldots, n)$ is either equal or greater than $T + 1$ or equal and smaller than $T - 1$. Let us mark $S'$ as some subset for which it holds that $\sum_{i \in S'} z_i \geq T + 1$. Let us denote $\mathbf{x}' = (\mathbf{1}_{S'}; \mathbf{0}_{\bar{S}'})$. Or in other words $\mathbf{x}'$ denotes a vector where all the values in $S'$ are set to 1 and the rest to 0. It clearly holds that:

$$f_1(\mathbf{x}') = \sum_{i \in \bar{S}'} \mathbf{x}'_i \cdot \mathbf{w^1_i} + b_1 =$$
$$-\sum_{i \in S'} z_i + b_1 \leq \tag{86}$$
$$-(T + 1) + T - \frac{1}{2} = -\frac{3}{2} < 0$$

It also holds that:

$$f_2(\mathbf{x}') = \sum_{i \in \bar{S}'} \mathbf{x}'_i \cdot \mathbf{w_i^2} + b_2 =$$
$$\sum_{i \in S'} z_i + b_2 \geq \tag{87}$$
$$(T+1) + (-T - \frac{1}{2}) = \frac{1}{2} > 0$$

This implies $f_1(\mathbf{x}')$ is negative and $f_2(\mathbf{x}')$ is positive, and overall we get that for the ensemble $f$ it holds that $f(\mathbf{x}')$ is positive.

Now, let us assume that $\sum_{i \in S'} z_i \leq T - 1$. Again, let us denote $\mathbf{x}' = (\mathbf{1}_{S'}; \mathbf{0}_{\bar{S}'})$. It clearly holds that:

$$f_1(\mathbf{x}') = \sum_{i \in \bar{S}'} \mathbf{x}'_i \cdot \mathbf{w_i^1} + b_1 =$$
$$-\sum_{i \in S'} z_i + b_1 \geq \tag{88}$$
$$-(T-1) + T - \frac{1}{2} = \frac{1}{2} > 0$$

It also holds that:

$$f_2(\mathbf{x}') = \sum_{i \in \bar{S}'} \mathbf{x}'_i \cdot \mathbf{w_i^2} + b_2 =$$
$$\sum_{i \in S'} z_i + b_2 \leq \tag{89}$$
$$(T-1) + (-T - \frac{1}{2}) = -\frac{3}{2} < 0$$

Implying, that under this scenario the opposite case occurs: $f_1(\mathbf{x}')$ is positive and $f_2(\mathbf{x}')$ is negative, and overall we get again that for the ensemble $f$ it holds that $f(\mathbf{x}')$ is positive.

This shows, that for any feasible value of $\mathbf{x}'$, $f(\mathbf{x}')$ is positive, and this implies that $S = \emptyset$ is a sufficient reason of $\langle f, \mathbf{x} = \mathbf{0}_n \rangle$.

If $\langle (z_1, z_2, \ldots, z_n), T \rangle \notin \overline{SSP}$, this means that there does exist a subset $S' \subseteq \{1, \ldots, n\}$ for which $\sum_{i \in S'} z_i = T$. We again denote $\mathbf{x}' = (\mathbf{1}_{S'}; \mathbf{0}_{\bar{S}'})$. It holds that:

$$f_1(\mathbf{x}') = \sum_{i \in \bar{S}'} \mathbf{x}'_i \cdot \mathbf{w_i^1} + b_1 =$$
$$-\sum_{i \in S'} z_i + b_1 = \tag{90}$$
$$-(T) + T - \frac{1}{2} = -\frac{1}{2} < 0$$

It also holds that:

$$f_2(\mathbf{x}') = \sum_{i \in \bar{S}'} \mathbf{x}'_i \cdot \mathbf{w_i^2} + b_2 =$$
$$\sum_{i \in S'} z_i + b_2 = \tag{91}$$
$$(T) + (-T - \frac{1}{2}) = -\frac{1}{2} < 0$$

This means that both $f_1(\mathbf{x}')$ and $f_2(\mathbf{x}')$ are negative and hence the ensemble $f(\mathbf{x}')$ is also negative. Since $f(\mathbf{x}) = f(\mathbf{0}_n)$ is positive, it holds that $S = \emptyset$ is not a sufficient reason of $\langle f, \mathbf{x} \rangle$, which concludes the reduction.

$\square$

**Lemma 20.** *The MCR query for a $k$-ensemble of Perceptrons is para-NP Hard.*

We will equivalently show that the *MCR* query for an ensemble of k=2 Perceptrons is NP-Complete. We use a refined version of the SSP problem — *kSSP*, which is also known to be NP-Complete.

---

**kSSP (k Subset Sum)**:
**Input**: $(z_1, z_2, \ldots, z_n)$ set of positive integers, a positive integer $k$, and a positive (target) integer $T$.
**Output**: *Yes*, if there exists a subset $S \subseteq (1, 2, \ldots, n)$ such that $|S| = k$ and $\sum_{i \in S} z_i = T$, and *No* otherwise.

---

We reduce $MCR$ for an ensemble with 2 Perceptrons from $kSSP$. Given some $\langle (z_1, z_2, \ldots, z_n), k, T \rangle$, the reduction constructs the two following Perceptrons $f_1 := \langle \mathbf{w^1}, b_1 \rangle$ and $f_2 := \langle \mathbf{w^2}, b_2 \rangle$, where $\mathbf{w^1} := (-z_1, -z_2, \ldots, -z_n)$, $b_1 := T - \frac{1}{2}$, $\mathbf{w^2} := (z_1, z_2, \ldots, z_n)$, and $b_2 := -T - \frac{1}{2}$. The reduction constructs $\langle f := (f_1, f_2), \mathbf{x} := \mathbf{0}_n, k \rangle$, and $\mathbf{0}_n$ denotes a vector of size $n$ where all values are set to 0.

First, we notice that:

$$f_1(\mathbf{x}) = f_1(\mathbf{0}_n) = T - \frac{1}{2} > 0 \ \wedge$$
$$f_2(\mathbf{x}) = f_2(\mathbf{0}_n) = -T - \frac{1}{2} < 0 \tag{92}$$

Since $f_1(\mathbf{x})$ is positive and $f_2(\mathbf{x})$ is negative, then the ensemble $f = \langle f_1, f_2 \rangle$ returns 1 for the input $\mathbf{x}$ (is positive), by definition.

If $\langle (z_1, z_2, \ldots, z_n), k, T \rangle \notin SSP$ then there does not exist a subset of features $S' \subseteq (1, 2, \ldots, n)$ such that $\sum_{i \in S'} z_i = T$. Since these are *integers*, this implies that any subset $S' \subseteq (1, 2, \ldots, n)$ *of size $k$* is either equal or greater than $T + 1$ or equal and smaller than $T - 1$. Let us mark $S'$ as some subset of size $k$ for which it holds that $\sum_{i \in S'} z_i \geq T + 1$. Let us denote $\mathbf{x}' = (\mathbf{1}_{S'}; \mathbf{0}_{\bar{S}'})$. Or in other words $\mathbf{x}'$ denotes a vector where all the values in $S'$ are set to 1 and the rest to 0. It clearly holds that:

$$f_1(\mathbf{x}') = \sum_{i \in \bar{S}'} \mathbf{x}'_i \cdot \mathbf{w^1_i} + b_1 =$$
$$-\sum_{i \in S'} z_i + b_1 \leq \tag{93}$$
$$-(T + 1) + T - \frac{1}{2} = -\frac{3}{2} < 0$$

It also holds that:

$$f_2(\mathbf{x}') = \sum_{i \in \bar{S}'} \mathbf{x}'_i \cdot \mathbf{w^2_i} + b_2 =$$
$$\sum_{i \in S'} z_i + b_2 \geq \tag{94}$$
$$(T + 1) + (-T - \frac{1}{2}) = \frac{1}{2} > 0$$

This implies $f_1(\mathbf{x}')$ is negative and $f_2(\mathbf{x}')$ is positive, and overall we get that for the ensemble $f$ it holds that $f(\mathbf{x}')$ is positive.

Now, let us assume that $\sum_{i \in S'} z_i \leq T - 1$. Again, let us denote $\mathbf{x}' = (\mathbf{1}_{S'}; \mathbf{0}_{\bar{S}'})$. It clearly holds that:

$$f_1(\mathbf{x}') = \sum_{i \in \bar{S}'} \mathbf{x}'_i \cdot \mathbf{w_i^1} + b_1 =$$

$$-\sum_{i \in S'} z_i + b_1 \geq \tag{95}$$

$$-(T-1) + T - \frac{1}{2} = \frac{1}{2} > 0$$

It also holds that:

$$f_2(\mathbf{x}') = \sum_{i \in \bar{S}'} \mathbf{x}'_i \cdot \mathbf{w_i^2} + b_2 =$$

$$\sum_{i \in S'} z_i + b_2 \leq \tag{96}$$

$$(T-1) + (-T - \frac{1}{2}) = -\frac{3}{2} < 0$$

Implying, that under this scenario the opposite case occurs: $f_1(\mathbf{x}')$ is positive and $f_2(\mathbf{x}')$ is negative, and overall we get again that for the ensemble $f$ it holds that $f(\mathbf{x}')$ is positive.

This shows that for any subset $S'$ of size $k$ the value of $f(\mathbf{x}')$ remains positive. Since the value of $f(\mathbf{x})$ is also positive, this implies that there is no contrastive reason $S'$ of size $k$ with respect to $\langle f, \mathbf{x} \rangle$. This also clearly implies that there is no contrastive reason $S'$ of size smaller than $k$ with respect to $\langle f, \mathbf{x} \rangle$.

If $\langle (z_1, z_2, \ldots, z_n), T \rangle \in SSP$, this means that there does exist a subset $S' \subseteq \{1, \ldots, n\}$ of size $k$ for which $\sum_{i \in S'} z_i = T$. We again denote $\mathbf{x}' = (\mathbf{1}_{S'}; \mathbf{0}_{\bar{S}'})$. It holds that:

$$f_1(\mathbf{x}') = \sum_{i \in \bar{S}'} \mathbf{x}'_i \cdot \mathbf{w_i^1} + b_1 =$$

$$-\sum_{i \in S'} z_i + b_1 = \tag{97}$$

$$-(T) + T - \frac{1}{2} = -\frac{1}{2} < 0$$

It also holds that:

$$f_2(\mathbf{x}') = \sum_{i \in \bar{S}'} \mathbf{x}'_i \cdot \mathbf{w_i^2} + b_2 =$$

$$\sum_{i \in S'} z_i + b_2 = \tag{98}$$

$$(T) + (-T - \frac{1}{2}) = -\frac{1}{2} < 0$$

This means that both $f_1(\mathbf{x}')$ and $f_2(\mathbf{x}')$ are negative and hence the ensemble $f(\mathbf{x}')$ is also negative. Since $f(\mathbf{x}) = f(\mathbf{0}_n)$ is positive, it holds that there exists a contrastive reason $(S')$ of size $k$ with respect to $\langle f, \mathbf{x} \rangle$, concluding the reduction.

$\square$

**Lemma 21.** *The CC and SHAP queries are para-#P-Hard for a $k$ ensemble of Perceptrons.*

*Proof.* The Hardness of the *CC* query follows from tbe fact that this problem is already #P-Hard for a single perceptron Barceló et al. (2020) ($k = 1$). For the *SHAP* query, we follow the proof suggested by Arenas et al. (2023) who showed a connection between the exact computation of shapley values computations and the model counting problem. Given some model $f$, we define $\#f$ as the number of assignments which output 1. More formally: $\#f : |\{\mathbf{z} \mid f(\mathbf{z}) = 1\}|$. Arenas et al. (2023) showed

the following connection for the specific case where $\mathcal{D}_p$ is taken to be the uniform distribution. It holds that for *all* $\mathbf{x} \in \mathbb{F}$ it holds that:

$$\#f = 2^n \big( f(\mathbf{x}) - \sum_{i \in [n]} \phi_i(f, \mathbf{x}) \big) \tag{99}$$

where the following relation is a consequence of the well-known *efficiency* property of shapley values. This result establishes a direct reduction from the *SHAP* query to the model counting problem. Moreover, the model counting problem is simply a private case of the *CC* query where $S := \emptyset$. This establishes a polynomial reduction from the *CC* to the *SHAP* query. We hence conclude that *SHAP* is also $\#P$-Hard for an ensemble consisting of only a single Perceptron.

$\square$

**Lemma 22.** *The MSR query for a $k$-ensemble of Perceptrons is $\Sigma_2^P$-Complete.*

*Proof.* **Membership.** We establish membership in para-$\Sigma_2^P$ using the direct definition of the complexity class since we can non-deterministically guess a subset $S$ and then utilize a coNP oracle to verify whether $S$ is sufficient.

**Hardness.** We will equivalently show that the *MSR* query for an ensemble of $k = 5$ Perceptrons is $\Sigma_2^P$-Hard. First, we will present the *Generalized Subset Sum* problem, which is known to be $\Sigma_2^P$-complete-complete Schaefer & Umans (2002).

---

**GSSP (Generalized Subset Sum)**:
**Input**: Two vectors of positive integers $\mathbf{u} = (\mathbf{u}_1, \mathbf{u}_2, \ldots, \mathbf{u}_l)$ and $\boldsymbol{b} = (\boldsymbol{b}_1, \boldsymbol{b}_2, \ldots, \boldsymbol{b}_m)$, and a positive (target) integer $T$.
**Output**: *Yes*, if there *exists* a binary vector $\mathbf{x} \in \{0, 1\}^l$ such that for *any* binary vector $\mathbf{y} \in \{0, 1\}^m$, it holds that: $\Sigma_{i=1}^l (\mathbf{x}_i \cdot \mathbf{u}_i) + \Sigma_{j=1}^m (\mathbf{y}_j \cdot \boldsymbol{b}_j) \neq T$; and *No* otherwise.

---

We will actually use a very close modified version of this problem which we term *$k$-Generalized Subset Sum*, which requires an additional constraint that the size of the vector $\mathbf{x}$ is equal to some input integer $k$. More formally, the problem is defined as follows:

---

**$k$-GSSP ($k$-Generalized Subset Sum)**:
**Input**: Two vectors of positive integers $\mathbf{u} = (\mathbf{u}_1, \mathbf{u}_2, \ldots, \mathbf{u}_l)$ and $\boldsymbol{b} = (\boldsymbol{b}_1, \boldsymbol{b}_2, \ldots, \boldsymbol{b}_m)$, an integer $k$, and a positive (target) integer $T$.
**Output**: *Yes*, if there *exists* a binary vector $\mathbf{x} \in \{0, 1\}^l$ such that $\|\mathbf{x}\|_1 = k$, and for *any* binary vector $\mathbf{y} \in \{0, 1\}^m$, it holds that: $\Sigma_{i=1}^l (\mathbf{x}_i \cdot \mathbf{u}_i) + \Sigma_{j=1}^m (\mathbf{y}_j \cdot \boldsymbol{b}_j) \neq T$; and *No* otherwise.

---

We next prove the following claim:

**Claim 16.** *The query $k$-GSSP is $\Sigma_2^P$-complete-hard.*

*Proof.* We present a polynomial-time reduction from *GSSP* to *$k$-GSSP*, enabling us to conclude that $k$-GSSP is $\Sigma_2^P$-hard. Given $\langle \mathbf{u} = (\mathbf{u}_1, \mathbf{u}_2, \ldots, \mathbf{u}_l), \boldsymbol{b} = (\boldsymbol{b}_1, \boldsymbol{b}_2, \ldots, \boldsymbol{b}_m), T \rangle$, we define some $G > 0$. The reduction then constructs $\langle \mathbf{u}' = (\mathbf{u}_1 + G, \mathbf{u}_2 + G, \ldots, \mathbf{u}_l + G, G_1, \ldots, G_l), \mathbf{v'} := \mathbf{v} = (\boldsymbol{b}_1, \boldsymbol{b}_2, \ldots, \boldsymbol{b}_m), k' := l, T' := T + (l \cdot G) \rangle$ (where $G_i$ is the $i$-th occurrence of the value $G$).

First, assume that $\langle \mathbf{u}, \boldsymbol{b}, T \rangle \in GSSP$. By construction, this implies that:

$$\exists \mathbf{x} \in \{0, 1\}^l \quad \forall \mathbf{y} \in \{0, 1\}^m \quad \Sigma_{i=1}^l (\mathbf{x}_i \cdot \mathbf{u}_i) + \Sigma_{j=1}^m (\mathbf{y}_j \cdot \boldsymbol{b}_j) \neq T \tag{100}$$

Let $\mathbf{x}$ be an element of $\{0, 1\}^l$. Therefore, for this $\mathbf{x}$, the following holds:

$$\forall \mathbf{y} \in \{0, 1\}^m \quad \Sigma_{i=1}^l (\mathbf{x}_i \cdot \mathbf{u}_i) + \Sigma_{j=1}^m (\mathbf{y}_j \cdot \boldsymbol{b}_j) \neq T \tag{101}$$

We express: $\|\mathbf{x}\|_1 = r \leq l$. Next, we divide the $l$ summations into two parts: the first sum will include the corresponding values in $\mathbf{u}$ with non-zero coordinates in $\mathbf{x}$, and the other part will include

the remaining values in $\mathbf{u}$. For simplicity and w.l.o.g., we also reindex the coordinates accordingly so that, w.l.o.g., all $r$ nonzero coordinates are the first ones of $\mathbf{x}$ (this is w.l.o.g. since we can reorder the corresponding coordinates in $\mathbf{u}$ accordingly). Next, given this $\mathbf{x}$, we can construct a new binary vector $\mathbf{x}' \in \{0,1\}^{2l}$ where the first $l$ coordinates are identical to the given vector $\mathbf{x}$, and the remaining $l$ coordinates are constructed as follows: the first $(l-r)$ coordinates of the second half, i.e., coordinates $(l+1)$ to $(2l-r)$ of $\mathbf{x}'$, will be "1", while the remaining coordinates of $\mathbf{x}'$, i.e., from $(2l-r+1)$ to $(2l)$, will be "0".

$$
\begin{aligned}
\Sigma_{i=1}^{2l}(\mathbf{x}_i' \cdot \mathbf{u}_i') = \\
\Sigma_{i=1}^{r}(\mathbf{x}_i' \cdot \mathbf{u}_i') + \Sigma_{i=r+1}^{l}(\mathbf{x}_i' \cdot \mathbf{u}_i') + \Sigma_{i=l+1}^{2l-r}(\mathbf{x}_i' \cdot \mathbf{u}_i') + \Sigma_{i=2l-r+1}^{2l}(\mathbf{x}_i' \cdot \mathbf{u}_i') = \\
\Sigma_{i=1}^{r}(\mathbf{x}_i \cdot (\mathbf{u}_i + G)) + \Sigma_{i=r+1}^{l}(0 \cdot (\mathbf{u}_i + G)) + \Sigma_{i=l+1}^{2l-r}(1 \cdot G) + \Sigma_{i=2l-r+1}^{2l}(0 \cdot G) = \\
\Sigma_{i=1}^{r}(\mathbf{x}_i \cdot \mathbf{u}_i) + (r \cdot G) + ((l-r) \cdot G)
\end{aligned}
\tag{102}
$$

And as (w.l.o.g.) the *last* $(l-r)$ coordinates for $\mathbf{x}$ are "0", it follows that for this constructed $\mathbf{x}'$, we can sum up to $l$ (and not "only" $r$):

$$
\Sigma_{i=1}^{2l}(\mathbf{x}_i' \cdot \mathbf{u}_i') = \Sigma_{i=1}^{l}(\mathbf{x}_i \cdot \mathbf{u}_i) + (l \cdot G)
\tag{103}
$$

Now, suppose, by contradiction, that for this constructed $\mathbf{x}'$, it holds that:

$$
\exists \mathbf{y} \in \{0,1\}^m \quad \Sigma_{i=1}^{2l}(\mathbf{x}_i' \cdot \mathbf{u}_i') + \Sigma_{j=1}^{m}(\mathbf{y}_j \cdot \boldsymbol{b}_j) = T'
\tag{104}
$$

Then, according to Eq. 103, it follows that:

$$
\exists \mathbf{y} \in \{0,1\}^m \quad \Sigma_{i=1}^{l}(\mathbf{x}_i \cdot \mathbf{u}_i) + (l \cdot G) + \Sigma_{j=1}^{m}(\mathbf{y}_j \cdot \boldsymbol{b}_j) = T'
\tag{105}
$$

And from the definition $T' := T + (l \cdot G)$, we infer from our assumption that:

$$
\begin{aligned}
\exists y \in \{0,1\}^m \quad \Sigma_{i=1}^{l}(\mathbf{x}_i \cdot \mathbf{u}_i) + (l \cdot G) + \Sigma_{j=1}^{m}(\mathbf{y}_j \cdot \boldsymbol{b}_j) = T + (l \cdot G) \iff \\
\exists \mathbf{y} \in \{0,1\}^m \quad \Sigma_{i=1}^{l}(\mathbf{x}_i \cdot \mathbf{u}_i) + \Sigma_{j=1}^{m}(\mathbf{y}_j \cdot \boldsymbol{b}_j) = T
\end{aligned}
\tag{106}
$$

However, this contradicts the outcome from Eq. 101 concerning this specific $\mathbf{x}$. Therefore, our assumption is incorrect, and thus for the constructed $\mathbf{x}' \in \{0,1\}^{2l}$, with $\|\mathbf{x}'\|_1 = l = k'$, it follows that:

$$
\forall \mathbf{y} \in \{0,1\}^m \quad \Sigma_{i=1}^{2l}(\mathbf{x}_i' \cdot \mathbf{u}_i') + \Sigma_{j=1}^{m}(\mathbf{y}_j \cdot \boldsymbol{b}_j) \neq T'
\tag{107}
$$

Therefore, we can conclude that: $\langle \mathbf{u}', \boldsymbol{b}', k', T' \rangle \in k\text{-GSSP}$.

Now, suppose $\langle \mathbf{u}, \mathbf{v}, T \rangle \notin GSSP$. This implies that $\forall \mathbf{x} \in \{0,1\}^l$:

$$
\exists \mathbf{y} \in \{0,1\}^m \quad \Sigma_{i=1}^{l}(\mathbf{x}_i \cdot \mathbf{u}_i) + \Sigma_{j=1}^{m}(\mathbf{y}_j \cdot \boldsymbol{b}_j) = T
\tag{108}
$$

We need to prove that this implies the following:

$$
\begin{aligned}
\forall \mathbf{x}' \in \{0,1\}^{2l} \quad \text{s.t} \quad \|\mathbf{x}'\|_1 = k', \\
\exists \mathbf{y} \in \{0,1\}^m, \quad \Sigma_{i=1}^{2l}(\mathbf{x}_i' \cdot \mathbf{u}_i') + \Sigma_{j=1}^{m}(\mathbf{y}_j \cdot \boldsymbol{b}_j') = T'
\end{aligned}
\tag{109}
$$

Let us assume, for the sake of contradiction, that this is not the case, i.e.:

$$
\begin{aligned}
\exists \mathbf{x}' \in \{0,1\}^{2l} \quad \text{s.t} \quad \|\mathbf{x}'\|_1 = k', \\
\forall \mathbf{y} \in \{0,1\}^m \quad \Sigma_{i=1}^{2l}(\mathbf{x}_i' \cdot \mathbf{u}_i') + \Sigma_{j=1}^{m}(\mathbf{y}_j \cdot \boldsymbol{b}_j') \neq T'
\end{aligned}
\tag{110}
$$

Let $\mathbf{x}' \in \{0,1\}^{2l}$ represent a binary input $\mathbf{x}' \in \{0,1\}^{2l}$. Therefore, $\|\mathbf{x}'\|_1 = k' := l$. Assume $1 \leq s \leq l$ corresponds to the number of "1" entries in $\mathbf{x}'$ located in the first $l$ coordinates of $\mathbf{u}'$, with the remaining $(k'-s) = (l-s)$ "1" entries of $\mathbf{x}'$ located in the second half of $\mathbf{u}'$. Further, assume without loss of generality that the *first* $s$ coordinates of $\mathbf{x}'_{[1\ldots l]}$ are "1", and similarly, without loss of generality, that the *first* $(l-s)$ coordinates of $\mathbf{x}'_{[(l+1)\ldots 2l]}$ (i.e., the second half of $\mathbf{x}'$'s $l$ coordinates) represent the locations of the remaining $(l-s)$ "1" values. Therefore, we establish that for this $\mathbf{x}'$:

$$
\begin{aligned}
\forall \mathbf{y} \in \{0,1\}^m, \quad & \Sigma_{i=1}^s (\mathbf{x}'_i \cdot \mathbf{u}'_i) + \Sigma_{i=s+1}^l (\mathbf{x}'_i \cdot \mathbf{u}'_i) + \Sigma_{i=l+1}^{2l-s} (\mathbf{x}'_i \cdot \mathbf{u}'_i) + \\
& \Sigma_{i=2l-s+1}^{2l} (\mathbf{x}'_i \cdot u'_i) + \Sigma_{j=1}^m (\mathbf{y}_j \cdot \mathbf{b}'_j) \neq T' \iff \\
\forall \mathbf{y} \in \{0,1\}^m, \quad & \Sigma_{i=1}^s (\mathbf{x}'_i \cdot \mathbf{u}'_i) + \Sigma_{i=s+1}^l (0 \cdot \mathbf{u}'_i) + \Sigma_{i=l+1}^{2l-s} (\mathbf{x}'_i \cdot \mathbf{u}'_i) + \\
& \Sigma_{i=2l-s+1}^{2l} (0 \cdot \mathbf{u}'_i) + \Sigma_{j=1}^m (\mathbf{y}_j \cdot \mathbf{b}'_j) \neq T' \iff \\
\forall \mathbf{y} \in \{0,1\}^m, \quad & \Sigma_{i=1}^s (\mathbf{x}'_i \cdot \mathbf{u}'_i) + \Sigma_{i=l+1}^{2l-s} (\mathbf{x}'_i \cdot \mathbf{u}'_i) + \Sigma_{j=1}^m (\mathbf{y}_j \cdot \mathbf{b}'_j) \neq T' \iff \\
\forall \mathbf{y} \in \{0,1\}^m, \quad & \Sigma_{i=1}^s (\mathbf{x}'_i \cdot (\mathbf{u}_i + G)) + \Sigma_{i=l+1}^{2l-s} (\mathbf{x}'_i \cdot G) + \Sigma_{j=1}^m (\mathbf{y}_j \cdot \mathbf{b}'_j) \neq T'
\end{aligned}
\tag{111}
$$

Similarly, by partitioning the summation and considering the coordinates of $\mathbf{x}'$, we can deduce that the aforementioned equation can be rewritten as:

$$
\begin{aligned}
\forall \mathbf{y} \in \{0,1\}^m, \quad & \Sigma_{i=1}^s (1 \cdot \mathbf{u}_i) + \Sigma_{i=1}^s (1 \cdot G) + \\
& \Sigma_{i=l+1}^{2l-s} (1 \cdot G) + \Sigma_{j=1}^m (\mathbf{y}_j \cdot \mathbf{b}'_j) \neq T' \iff \\
\forall \mathbf{y} \in \{0,1\}^m \quad & \Sigma_{i=1}^s (1 \cdot \mathbf{u}_i) + (s + (l-s)) \cdot G + \\
& \Sigma_{j=1}^m (\mathbf{y}_j \cdot \mathbf{b}'_j) \neq T' = T + (l \cdot G) \iff \\
\forall \mathbf{y} \in \{0,1\}^m \quad & \Sigma_{i=1}^s (1 \cdot \mathbf{u}_i) + \Sigma_{j=1}^m (\mathbf{y}_j \cdot \mathbf{b}'_j) \neq T
\end{aligned}
\tag{112}
$$

Equivalently:

$$
\forall \mathbf{y} \in \{0,1\}^m, \quad \Sigma_{i=1}^s (1 \cdot \mathbf{u}_i) + \Sigma_{i=s+1}^{2l} (0 \cdot \mathbf{u}_i) + \Sigma_{j=1}^m (\mathbf{y}_j \cdot \mathbf{b}'_j) \neq T
\tag{113}
$$

Let us define an input $\mathbf{x}'' \in \{0,1\}^l$, such that the first $1 \leq s \leq l$ coordinates are "1", and the remaining ones are "0". In other words, $\mathbf{x}'' := \mathbf{x}'_{[1\ldots l]}$. Therefore, given Eq. 113 and the specified $\mathbf{x}'' \in \{0,1\}^l$, it follows that:

$$
\forall \mathbf{y} \in \{0,1\}^m \quad \Sigma_{i=1}^l (\mathbf{x}''_i \cdot \mathbf{u}_i) + \Sigma_{j=1}^m (\mathbf{y}_j \cdot \mathbf{b}_j) \neq T
\tag{114}
$$

However, this contradicts Eq. 108, and therefore we determine that our initial assumption is erroneous (i.e., no such $\mathbf{x}' \in \{0,1\}^{2l}$ exists), and:

$$
\begin{aligned}
\forall \mathbf{x}' \in \{0,1\}^{2l} \quad & \text{s.t} \quad \|\mathbf{x}'\|_1 = k', \\
\exists \mathbf{y} \in \{0,1\}^m \quad & \Sigma_{i=1}^{2l} (\mathbf{x}'_i \cdot \mathbf{u}'_i) + \Sigma_{j=1}^m (\mathbf{y}_j \cdot \mathbf{b}'_j) = T'
\end{aligned}
\tag{115}
$$

Thus, $\langle \mathbf{u}', \mathbf{b}', k', T' \rangle \notin$ *k-GSSP*. To conclude, we have proved that: $\langle \mathbf{u}, \mathbf{b}, T \rangle \in$ *GSSP* $\iff$ $\langle \mathbf{u}', \mathbf{b}', k', T' \rangle \in$ *k-GSSP*, completing our reduction.

$\square$

Next, we introduce a variant of the *k-GSSP* problem, which we refer to as the *k-GSSP** problem.

---

**k-GSSP* (Constrained k Generalized Subset Sum)**:
**Input**: A set of positive integers $(\mathbf{z}_1, \mathbf{z}_2, \ldots, \mathbf{z}_n)$, a subset $S_0$, an integer $k$, and a positive (target) integer $T$.
**Output**: *Yes*, if there *exists* a subset of features (indices) $S \subseteq S_0$, such that, $|S| = k$, and for *all* subsets $S' \subseteq \bar{S}$ it holds that $\Sigma_{i \in S} (\mathbf{z}_i) + \Sigma_{j \in S'} (\mathbf{z}_j) \neq T$; and *No* otherwise.

---

Now, we will establish the hardness of this refined query by demonstrating the following claim:

**Claim 17.** *The query k-GSSP$^*$ is $\Sigma_2^P$-complete-hard.*

*Proof.* We will demonstrate $\Sigma_2^P$-complete-hardness through a reduction from the *k-GSSP* problem. Starting with $\langle \mathbf{u} = (\mathbf{u}_1, \mathbf{u}_2, \ldots, \mathbf{u}_l), \boldsymbol{b} = (\boldsymbol{b}_1, \boldsymbol{b}_2, \ldots, \boldsymbol{b}_m), k, T \rangle$, we produce $\langle \mathbf{z} = (\mathbf{z}_1, \mathbf{z}_2, \ldots, \mathbf{z}_n), S_0 = (1, \ldots, l), k', T' \rangle$. The vector $\mathbf{z}$ is formed by concatenating $\mathbf{u}$ and $\boldsymbol{b}$ in the following manner:

- $\mathbf{z} := (\mathbf{u}' {}^\frown \boldsymbol{b}') \in \mathbb{N}^{n:=(l+m)}$, i.e., for any $1 \leq i \leq l$: $\mathbf{z}_i = \mathbf{u}_i$, and for any $(l+1) \leq j \leq (l+m)$: $\mathbf{z}_j = \boldsymbol{b}_{j-l}$.
- $\mathbf{u}' := [(2n+1) \cdot \mathbf{u}] + \mathbb{1}$, i.e., $\mathbf{u}'_j = [(2n+1) \cdot \mathbf{u}_j] + 1$
- $\boldsymbol{b}' := (2n+1) \cdot \boldsymbol{b}$, i.e., $\boldsymbol{b}'_i = (2n+1) \cdot \boldsymbol{b}_i$

We also set $T' := T(2n+1) + k$ and $k' := k$. We then will prove that: $\langle \mathbf{u}, \boldsymbol{b}, k, T \rangle \in k\text{-}GSSP \iff \langle \mathbf{z}, S_0, k', T' \rangle \in k\text{-}GSSP^*$.

First, suppose $\langle \mathbf{u}, \boldsymbol{b}k, T \rangle \in k\text{-}GSSP$. Then, there *exists* a binary vector $\mathbf{x} \in {0, 1}^l$ such that $\|\mathbf{x}\|_1 = k$, and for *any* binary vector $\mathbf{y} \in {0, 1}^m$, the following condition is met: $\Sigma_{i=1}^l (\mathbf{x}_i \cdot \mathbf{u}_i) + \Sigma_{j=1}^m (\mathbf{y}_j \cdot \boldsymbol{b}_j) \neq T$. Assuming, without loss of generality, that the *first $k$* entries of $\mathbf{u}$ align with this particular binary vector $\mathbf{x} \in \{0,1\}^l$ (note that $\|\mathbf{x}\|_1 = k$), we define the subset $S$ to include all corresponding indices, thus $S := \{i | \mathbf{x}_i = 1\} = \{1, \ldots, k\} \subseteq S_0$. We will now establish that for *all* subsets $S' \subseteq \bar{S} := \{(k+1), \ldots, l, (l+1), \ldots, n := (l+m)\}$, the following holds true for our chosen set $S$:

$$\Sigma_{i \in S}(\mathbf{z}_i) + \Sigma_{j \in S' \subseteq \bar{S}}(\mathbf{z}_j) \neq T' \tag{116}$$

We will demonstrate this in two parts: initially by considering subsets of $\bar{S}$ that include only features from $\{(l+1), \ldots, (l+m)\}$, and subsequently by considering subsets of $\bar{S}$ that intersect with $\{(k+1), \ldots, l\}$. In the first scenario, consider all $S' \subseteq \bar{S}$ where $S' \subseteq \{(l+1), \ldots, (l+m) = n\}$, meaning the features correspond exclusively to those in the original $\boldsymbol{b}$ vector. For this particular $\mathbf{x} \in \{0,1\}^l$, it is established that for *any* $\mathbf{y} \in \{0,1\}^m$:

$$\Sigma_{i=1}^l (\mathbf{x}_i \cdot \mathbf{u}_i) + \Sigma_{j=1}^m (\mathbf{y}_j \cdot \boldsymbol{b}_j) \neq T \iff$$
$$(2n+1)\Sigma_{i=1}^l (\mathbf{x}_i \cdot \mathbf{u}_i) + (2n+1)\Sigma_{j=1}^m (\mathbf{y}_j \cdot \boldsymbol{b}_j) \neq T(2n+1) \iff \tag{117}$$
$$(2n+1)\Sigma_{i=1}^l (\mathbf{x}_i \cdot \mathbf{u}_i) + (2n+1)\Sigma_{j=1}^m (\mathbf{y}_j \cdot \boldsymbol{b}_j) + k \neq T(2n+1) + k = T'$$

Given that $\|\mathbf{x}\|_1 = k$, we proceed under the assumption (w.l.o.g.) that the *first $k$* indices of $\mathbf{x}$ are set to "1", while the remaining $(l-k)$ coordinates are 0". Therefore, for *any* $\mathbf{y} \in \{0,1\}^m$:

$$(2n+1)\Sigma_{i=1}^k (\mathbf{x}_i \cdot \mathbf{u}_i) + (2n+1)\Sigma_{i=k+1}^l (\mathbf{x}_i \cdot \mathbf{u}_i) +$$
$$(2n+1)\Sigma_{j=1}^m (\mathbf{y}_j \cdot \boldsymbol{b}_j) + k \neq T' \iff$$
$$\Sigma_{i=1}^k ((2n+1) \cdot \mathbf{x}_i \cdot \mathbf{u}_i) + \Sigma_{i=k+1}^l ((2n+1) \cdot \mathbf{x}_i \cdot \mathbf{u}_i) + \tag{118}$$
$$\Sigma_{j=1}^m ((2n+1) \cdot \mathbf{y}_j \cdot \boldsymbol{b}_j) + k \neq T'$$

And, assuming (w.l.o.g.) that the *first $k$* coordinates of $\mathbf{x}$ are "1", with the remaining ones set to "0", it follows that for *any* $\mathbf{y} \in \{0,1\}^m$:

$$\Sigma_{i=1}^k ((2n+1) \cdot 1 \cdot \mathbf{u}_i) + \Sigma_{i=k+1}^l ((2n+1) \cdot 0 \cdot \mathbf{u}_i) +$$
$$\Sigma_{j=1}^m ((2n+1) \cdot \mathbf{y}_j \cdot \boldsymbol{b}_j) + k \neq T' \iff \tag{119}$$
$$\Sigma_{i=1}^k ((2n+1) \cdot 1 \cdot \mathbf{u}_i) + \Sigma_{j=1}^m ((2n+1) \cdot \mathbf{y}_j \cdot \boldsymbol{b}_j) + k \neq T' \iff$$
$$\Sigma_{i=1}^k ([(2n+1) \cdot \mathbf{u}_j] + 1) + \Sigma_{j=1}^m (\mathbf{y}_j \cdot (2n+1) \cdot \boldsymbol{b}_j) \neq T'$$

Given our definitions of $\mathbf{u}'$ and $\boldsymbol{b}'$, it follows that for *any* $\mathbf{y} \in \{0,1\}^m$:

$$\Sigma_{i=1}^k (\mathbf{u}'_i) + \Sigma_{j=1}^m (\mathbf{y}_j \cdot \boldsymbol{b}'_j) \neq T' \tag{120}$$

Since $S$ encompasses the first $k$ coordinates of **u'** (which align with the $k$ "1" coordinates of the specified **x**, by design), it results that for *any* $\mathbf{y} \in \{0,1\}^m$:

$$\Sigma_{i \in S}(\mathbf{z}_i) + \Sigma_{j=1}^m (\mathbf{y}_j \cdot \mathbf{z}_{l+j}) \neq T' \tag{121}$$

Therefore, for all $S' \subseteq \bar{S}$ where $S' \subseteq \{(l+1), \ldots, (l+m)\}$, it is established that:

$$\Sigma_{i \in S}(\mathbf{z}_i) + \Sigma_{i \in S' \subseteq \bar{S}}(\mathbf{z}_i) \neq T' \tag{122}$$

In the second scenario, we consider any subset $S' \subseteq \bar{S}$ that intersects with $\{(k+1), \ldots, l\}$, meaning there is at least one feature $i \in S'$ such that $(k+1) \leq i \leq l$. We note that, according to our construction, $S$ contains *exactly* $k$ indices, each corresponding to a value of $[(2n+1) \cdot \mathbf{u}_i] + 1$. In this second case, $S' \subseteq \bar{S}$ includes *at least* one index that corresponds to a value of $[(2n+1) \cdot \mathbf{u}_i] + 1$ (corresponding to **u'**) and potentially other values of $(2n+1) \cdot \boldsymbol{b}_i$ (corresponding to $\boldsymbol{b}'$). Consequently, when adding the values associated with the indices from $S$ and $S'$, we obtain *at least* $(k+1)$ values (and at most, $n$), with *at least* $k+1$ values $t$ being such that $t \mod (2n+1) = 1$. Therefore, the total sum will yield a value with a modulus of *at least* $k+1$ over $(2n+1)$, implying that:

$$[\Sigma_{i \in S}(\mathbf{z}_i)] mod(2n+1) = k \implies$$
$$1 \leq [\Sigma_{i \in S' \subseteq \bar{S}}(\mathbf{z}_i)] mod(2n+1) \leq l - k \implies \tag{123}$$
$$(\Sigma_{i \in S}(\mathbf{z}_i) + \Sigma_{i \in S' \subseteq \bar{S}}(\mathbf{z}_i)) mod(2n+1) \neq k$$

Lastly, given that $T' := (2n+1)T + k$, it follows that $(T') \mod (2n+1) = k$, therefore:

$$\Sigma_{i \in S}(\mathbf{z}_i) + \Sigma_{i \in S' \subseteq \bar{S}}(\mathbf{z}_i) \neq T' = (2n+1)T + k \tag{124}$$

Thus, we have demonstrated in both scenarios that for our defined $S \subseteq S_0$, it is true that $|S| = k$ and for *all* subsets $S' \subseteq \bar{S} := \{(k+1), \ldots, l, (l+1), \ldots, n := (l+m)\}$, it holds that:

$$\Sigma_{i \in S}(\mathbf{z}_i) + \Sigma_{j \in S' \subseteq \bar{S}}(\mathbf{z}_j) \neq T' \tag{125}$$

Hence, $\langle \mathbf{z}, S_0, k', T' \rangle \in k\text{-}GSSP^*$

For the other direction, if $\langle \mathbf{u}, \boldsymbol{b}, k, T \rangle \notin k\text{-}GSSP$, then for *every* binary vector $\mathbf{x} \in \{0,1\}^l$ with $\|\mathbf{x}\|_1 = k$, there *exists* a binary vector $\mathbf{y}' \in \{0,1\}^m$ such that: $\Sigma_{i=1}^l(\mathbf{x}_i \cdot \mathbf{u}_i) + \Sigma_{j=1}^m(\mathbf{y}'j \cdot \boldsymbol{b}j) = T$. For every $k$-sized subset $S \subseteq S_0$, we define $\mathbf{x}' := \mathbb{1}_{i \in S}$. Additionally, for each fixed $S$, we define the set $S' \subseteq \bar{S}$ to be:

$$S' := \{(|\mathbf{u}| + i)|\mathbf{y}'_i = 1\} \tag{126}$$

for the corresponding $\mathbf{y}'$ that aligns with the indicator $\mathbf{x}'$ associated with $S'$. Next, we observe that:

$$\Sigma_{i \in S}(\mathbf{z}_i) + \Sigma_{j \in S' \subseteq \bar{S}}(\mathbf{z}_j) \underset{(*)}{=}$$
$$\Sigma_{i \in S}(\mathbf{u}'_i) + \Sigma_{j \in S' \subseteq \bar{S}}(\boldsymbol{b}'_j) =$$
$$\Sigma_{i \in S}((2n+1)\mathbf{u}_i + 1) + \Sigma_{j \in S' \subseteq \bar{S}}((2n+1)\boldsymbol{b}_i) =$$
$$(2n+1)[\Sigma_{i \in S}(\mathbf{u}_i) + \Sigma_{j \in S' \subseteq \bar{S}}(\boldsymbol{b}_i)] + |S| \cdot 1 \underset{(**)}{=}$$
$$(2n+1)[\Sigma_{i=1}^l(\mathbf{x}'_i \cdot \mathbf{u}_i) + \Sigma_{j=1}^m(\mathbf{y}'_j \cdot \boldsymbol{b}_j)] + k = \tag{127}$$
$$(2n+1)[T] + k =$$
$$(2n+1)[\frac{T'-k}{2n+1}] + k =$$
$$[T'-k] + k$$

Where (*) arises because we have selected $S' \subseteq \bar{S}$ to exclusively contain indices of values pertaining to **v'** (i.e., $S' \cap 1, \ldots, l \neq \emptyset$), and (**) is due to considering any subset $S \subseteq S_0$ with a size of $|S| = k$.

Therefore, we conclude that for *all* $k$-sized subsets $S \subseteq S_0$, it is established that there *exists* a subset $S' \subseteq \bar{S}$ such that:

$$\Sigma_{i \in S}(\mathbf{z}_i) + \Sigma_{j \in S' \subseteq \bar{S}}(\mathbf{z}_j) = T' \tag{128}$$

Thus, $\langle \mathbf{z}, S_0, k', T' \rangle \notin k\text{-GSSP}^*$, as required.

$\square$

We will now present the final component of this proof by establishing the following claim:

**Claim 18.** *The MSR query for an ensemble of $k$-Perceptrons is $\Sigma_2^P$-complete-hard for $k = 5$.*

*Proof.* We will outline a polynomial-time reduction from *k-GSSP*$^*$ to k-perceptron-MSR, specifically for $k = 5$. Given $\langle \mathbf{z}, S_0, k, T \rangle$, our polynomial-time reduction produces $\langle (f_1, \ldots, f_5), \mathbf{x}', k \rangle$, in the following manner: Initially, the reduction verifies (in polynomial time) whether: $\Sigma_{i=1}(\mathbf{z}_i) = T$. If this condition is met, then since $\mathbf{z}$ consists solely of strictly positive integers, any *strict* $k$-sized subset of these will not sum to $T$. Consequently, $\langle \mathbf{z}, S_0, k, T \rangle$ qualifies as belonging to k-GSSP*. As a result, the reduction returns, within polynomial time, $\langle (f_1 := \text{True}, \ldots, f_5 := \text{True}), \mathbf{x}', k \rangle$ for k-Perceptron-MSR, as any $k$-sized set of features sufficiently justifies the condition. Otherwise, if $\Sigma_{i=1}(\mathbf{z}_i) \neq T$, the following steps are taken: we initialize $\mathbf{x}' := 1^n$ and $k' := k$. For each $1 \leq i \leq 5$, the perceptron $f_i$ is defined as follows:

- $f_1 := (\mathbf{w}_1, b_1)$, for $\mathbf{w}_1 := (-\mathbf{z}_1, \ldots, -\mathbf{z}_n)$, and $b_1 := T - \frac{1}{2}$

- $f_2 := (\mathbf{w}_2, b_2)$, for $\mathbf{w}_2 := (z_1, \ldots, z_n)$, and $b_2 := -T - \frac{1}{2}$

- $f_3 := (\mathbf{w}_3, b_3)$, for $\mathbf{w}_3 := (\mathbf{1}_{S_0}; \mathbf{0}_{\bar{S}_0})$ and $b_3 := -k$ it holds that: $f_3(\mathbf{x}) = 1 \iff \Sigma_{i=1}^n(\mathbf{x}_i) - k \geq 0 \iff \Sigma_{i=1}^n(\mathbf{x}_i) \geq k \iff \Sigma_{i=1}^{|S_0|}(\mathbf{1}_{S_0} \wedge \mathbf{x}_i) + \Sigma_{i=|S_0|+1}^n(\mathbf{1}_{S_0} \wedge \mathbf{x}_i) \geq k \iff \Sigma_{i=1}^{|S_0|}(\mathbf{1}_{S_0} \wedge \mathbf{x}_i) \geq k \iff |\{\mathbf{x}_i | \mathbf{x}_i = 1 \wedge \mathbf{x}_i \in S_0\}| \geq k$, i.e., $f_3$ classifies as "1" if and only if the input has *at least* $k$ "1" values in $S_0$.

- $f_4 := (\mathbf{w}_4, b_4)$, for $\mathbf{w}_4 := (0, \ldots, 0)$, and $b_4 := 1$, i.e., $f_4$=True

- $f_5 := \mathbf{1}_{1^n}$, i.e., $f_5(\mathbf{x}) = 1 \iff \mathbf{x} := 1^n$ (i.e., $f_5$ acts as a Perceptron serving as an indicator function for the constructed input $\mathbf{x} := 1^n$. This setup can be implemented in polynomial time, as demonstrated in **?**.)

First, we observe that the ensemble $(f_1, \ldots, f_5)$ classifies $\mathbf{x}' := 1^n$ as "1". This classification results from a majority of at least three (out of five) ensemble members designating the input $1^n$ as "1":

$$f_3(\mathbf{x'}) = step(\Sigma_{i=1}^{|S_0|}(\mathbf{x}'_i - k) = step(\Sigma_{i=1}^{|S_0|}(1) - k) = step(|S_0| - k) = 1 \tag{129}$$

Thus, $f_3(\mathbf{x'}) = f_3(1^n) = 1$. Additionally, $f_4(1^n) = 1$, as it classifies every input as "1", and $f_5(1^n) = 1$ as it functions as an indicator for this very input. We will prove that: $\langle \mathbf{z}, S_0, k, T \rangle \in k\text{-GSSP}^* \iff \langle (f_1, \ldots, f_5), \mathbf{x}', k' \rangle \in \text{MSR}$ for $k$-ensembles of Perceptrons when $k = 5$.

Initially, assuming that $\langle \mathbf{z}, S_0, k, T \rangle \in k\text{-GSSP}^*$, there *exists* a subset $S \subseteq S_0$ with $|S| = k$ and for every $S' \subset \bar{S}$ it holds that $\Sigma_{i \in S}(\mathbf{z}_i) + \Sigma_{j \in S'}(\mathbf{z}_j) \neq T$. We then contend that $S$ serves as a $k$-sized explanation for input $\mathbf{x}' := 1^n$ within the 5-Perceptron ensemble $(f_1, \ldots, f_5)$. When considering $S$, we conclude that for every $S' \subseteq \bar{S}$: $\Sigma_{i \in S}(\mathbf{z}_i) + \Sigma_{j \in S'}(\mathbf{z}_j) \neq T$, or equivalently, for each $S'$:

$$[\Sigma_{i \in S}(\mathbf{z}_i) + \Sigma_{j \in S'}(\mathbf{z}_j) \geq T + 1] \vee [\Sigma_{i \in S}(\mathbf{z}_i) + \Sigma_{j \in S'}(\mathbf{z}_j) \leq T - 1] \tag{130}$$

We will demonstrate that in both cases, *exactly* one of $f_1$ and $f_2$ classifies an input as "1", while the other classifies it as "0". In the scenario where $\Sigma_{i \in S}(\mathbf{z}_i) + \Sigma_{j \in S'}(\mathbf{z}_j) \geq T + 1$, for any input $\mathbf{x} \in \{0, 1\}^n$ with $\mathbf{x}_S = 1S$, we construct $(\mathbf{x}_S; \mathbf{y}_{\bar{S}})$, applicable for $\mathbf{x} \in \{0, 1\}^n$ and for *every* $\mathbf{y} \in \{0, 1\}^n$, ensuring that both $f_2((\mathbf{x}_S; \mathbf{y}_{\bar{S}})) = 1$ and $f_1((\mathbf{x}_S; \mathbf{y}_{\bar{S}})) = 0$. Moreover, $f_2((\mathbf{x}_S; \mathbf{y}_{\bar{S}})) = 1$', based on the following:

$$f_2((\mathbf{x}_S; \mathbf{y}_{\bar{S}})) =$$
$$step([(\mathbf{x}_S; \mathbf{y}_{\bar{S}}) \cdot \mathbf{w}_2] + b_2) =$$
$$step([(\mathbf{x}_S; \mathbf{y}_{\bar{S}}) \cdot (z_1, \ldots, z_n)] + (-T - \frac{1}{2})) = \quad (131)$$
$$step(\Sigma_{i \in S}(\mathbf{x}_i \cdot \mathbf{z}_i) + \Sigma_{j \in \bar{S}}(\mathbf{y}_i \cdot \mathbf{z}_j) - T - \frac{1}{2}) \geq$$
$$step((T + 1) - T - \frac{1}{2}) = step(\frac{1}{2}) = 1$$

In this case, it also true that $f_1((\mathbf{x}_S; \mathbf{y}_{\bar{S}})) = 0$, as the following holds:

$$f_1((\mathbf{x}_S; \mathbf{y}_{\bar{S}})) =$$
$$step([(\mathbf{x}_S; \mathbf{y}_{\bar{S}}) \cdot \mathbf{w}_1] + b_1) =$$
$$step([(\mathbf{x}_S; \mathbf{y}_{\bar{S}}) \cdot (-\mathbf{z}_1, \ldots, -\mathbf{z}_n)] + (T - \frac{1}{2})) = \quad (132)$$
$$step(-\Sigma_{i \in S}(z_i) - \Sigma_{j \in \bar{S}}(z_j) + (T - \frac{1}{2})) \leq$$
$$step(-(T + 1) + (T - \frac{1}{2})) = step(-\frac{3}{2}) = 0$$

With the last transition justified by the assumption that: $\Sigma_{i \in S}(\mathbf{z}_i) + \Sigma_{j \in S'}(\mathbf{z}_j) \geq T + 1$, and we also observe that we have selected $S' := \bar{S}$ (which is valid since trivially $\bar{S} \subseteq \bar{S}$). In the situation where $\Sigma_{i \in S}(\mathbf{z}_i) + \Sigma_{j \in S'}(\mathbf{z}_j) \leq T - 1$, for any input $\mathbf{x} \in \{0, 1\}^n$ with $\mathbf{x}_S = \mathbf{1}$, we construct $(\mathbf{x}_S; \mathbf{y}_{\bar{S}})$, suitable for $\mathbf{x} \in \{0, 1\}^n$ and for *every* $\mathbf{y} \in \{0, 1\}^n$, ensuring that both $f_1((\mathbf{x}_S; \mathbf{y}_{\bar{S}})) = 1$ and $f_2((\mathbf{x}_S; \mathbf{y}_{\bar{S}})) = 0$. Additionaly, $f_1((\mathbf{x}_S; \mathbf{y}_{\bar{S}})) = 1$, based on the following:

$$f_1((\mathbf{x}_S; \mathbf{y}_{\bar{S}})) =$$
$$step([(\mathbf{x}_S; \mathbf{y}_{\bar{S}}) \cdot \mathbf{w}_1] + b_1) =$$
$$step([(\mathbf{x}_S; \mathbf{y}_{\bar{S}}) \cdot (-\mathbf{z}_1, \ldots, -\mathbf{z}_n)] + (T - \frac{1}{2})) = \quad (133)$$
$$step(-[\Sigma_{i \in S}(\mathbf{x}_i \cdot \mathbf{z}_i) + \Sigma_{j \in \bar{S}}(\mathbf{y}_i \cdot \mathbf{z}_j)] + T - \frac{1}{2}) \geq$$
$$step((-1)(T - 1) + T - \frac{1}{2}) = step(1 - T + T - \frac{1}{2}) = step(\frac{1}{2}) = 1$$

In this scenario, it also holds that $f_2((\mathbf{x}_S; \mathbf{y}_{\bar{S}})) = 0$, as the following holds:

$$f_2((\mathbf{x}_S; \mathbf{y}_{\bar{S}})) =$$
$$step([(\mathbf{x}_S; \mathbf{y}_{\bar{S}}) \cdot \mathbf{w}_2] + b_2) =$$
$$step([(\mathbf{x}_S; \mathbf{y}_{\bar{S}}) \cdot (\mathbf{z}_1, \ldots, \mathbf{z}_n)] + (-T - \frac{1}{2})) = \quad (134)$$
$$step(\Sigma_{i \in S}(\mathbf{z}_i) + \Sigma_{j \in \bar{S}}(\mathbf{z}_j) + (-T - \frac{1}{2})) \leq$$
$$step(T - 1 + (-T - \frac{1}{2})) = step(-\frac{3}{2}) = 0$$

The last transition remains valid under the assumption: $\Sigma_{i \in S}(\mathbf{z}_i) + \Sigma_{j \in S'}(\mathbf{z}_j) \leq T - 1$. We further observe that $S' := \bar{S}$, which is valid since $\bar{S} \subseteq \bar{S}$ by definition. Therefore, when $\Sigma_{i \in S}(\mathbf{z}_i) + \Sigma_{j \in S'}(\mathbf{z}_j) \neq T$, it follows that *exactly* one of the Perceptrons in the pair $(f_1, f_2)$ classifies the input $(\mathbf{x}_S; \mathbf{y}_{\bar{S}})$ as "1", while the other classifies it as "0".

Furthermore, $f_3((\mathbf{x}_S; \mathbf{y}_{\bar{S}})) = 1$ because $(\mathbf{x}_S; \mathbf{y}_{\bar{S}})$ contains $k$ "1" values, which is the threshold required by $f_3$ to activate. It is also observed that $f_4((\mathbf{x}_S; \mathbf{y}_{\bar{S}})) \geq 0$, implying that it evaluates to True (classifying all inputs as 1"). Therefore, when $k$ values of "1" are set in $S$, one of two outcomes occurs: either the majority $(f_2, f_3, f_4)$ or the majority $(f_1, f_3, f_4)$ classifies every input $(\mathbf{x}_S, \mathbf{y}_{\bar{S}})$ as

"1" for *every* possible $\mathbf{y} \in \{0,1\}^n$. Consequently, $S$ serves as a $k$-sized sufficient reason for input $\mathbf{1}^n$ with respect to $(f_2, \ldots, f_5)$, thus $\langle (f_1, \ldots, f_5), \mathbf{x'}, k' \rangle \in$ MSR for $k$-ensemble Perceptrons when $k = 5$.

For the second direction, assume that $\langle \mathbf{z}, S_0, k, T \rangle \notin k\text{-}GSSP^*$. We aim to demonstrate that this leads to $\langle (f_1, \ldots, f_5), \mathbf{x'}, k' \rangle \notin$ MSR for $k$-ensemble Perceptrons when $k = 5$. For the sake of contradiction, suppose the opposite is true, i.e., $\langle (f_1, \ldots, f_5), \mathbf{x'}, k' \rangle \in$ MSR for $k$-ensemble Perceptrons when $k = 5$. In other words, we posit that there are $k$ features in $\mathbf{x} = \mathbf{1}^n$ which, when fixed, result in the ensemble consistently classifying as "1". We will prove this to be impracticable by examining the only two scenarios: (i) that these features are all contained within $S_0$; (ii) that at least one of the fixed $k$ features resides in $\bar{S}_0$.

In the first scenario, assume that all $k$ features are within $S_0$. Yet, since $\langle \mathbf{z}, S_0, k, T \rangle \notin k\text{-}GSSP^*$, it follows that for *any* input $(\mathbf{x}_S; \mathbf{y}_{\bar{S}})$ (for *any* $k$-sized subset $S \subseteq S_0$ and *any* $\mathbf{x} \in \{0,1\}^n$), there *exists* a corresponding $\mathbf{y} \in \{0,1\}^n$ such that:

$$\Sigma_{i \in S}(\mathbf{z}_i) + \Sigma_{j \in S'}(\mathbf{z}_j) = T \tag{135}$$

This suggests that *both* $f_1$ and $f_2$ classify the input $(\mathbf{x}_S; \mathbf{y}_{\bar{S}})$ as "0", which we will demonstrate below. Additionally, $f_1((\mathbf{x}_S; \mathbf{y}_{\bar{S}})) = 0$, as evidenced by the following:

$$\begin{aligned}
f_1((\mathbf{x}_S; \mathbf{y}_{\bar{S}})) = \\
step([(\mathbf{x}_S; \mathbf{y}_{\bar{S}}) \cdot \mathbf{w}_1] + b_1) = \\
step([(\mathbf{x}_S; \mathbf{y}_{\bar{S}}) \cdot (-\mathbf{z}_1, \ldots, -\mathbf{z}_n)] + (T - \frac{1}{2})) = \\
step(-[\Sigma_{i \in S}(\mathbf{x}_i \cdot \mathbf{z}_i) + \Sigma_{j \in \bar{S}}(\mathbf{y}_i \cdot \mathbf{z}_j)] + T - \frac{1}{2}) = \\
step(-T + T - \frac{1}{2}) = step(-\frac{1}{2}) = 0
\end{aligned} \tag{136}$$

Additionally, $f_2((\mathbf{x}_S; \mathbf{y}_{\bar{S}})) = 0$, as the following holds:

$$\begin{aligned}
f_2((\mathbf{x}_S; \mathbf{y}_{\bar{S}})) = \\
step([(\mathbf{x}_S; \mathbf{y}_{\bar{S}}) \cdot \mathbf{w}_2] + b_2) = \\
step([(\mathbf{x}_S; \mathbf{y}_{\bar{S}}) \cdot (\mathbf{z}_1, \ldots, \mathbf{z}_n)] + (-T - \frac{1}{2})) = \\
step(\Sigma_{i \in S}(\mathbf{x}_i \cdot \mathbf{z}_i) + \Sigma_{j \in \bar{S}}(\mathbf{y}_i \cdot \mathbf{z}_j) - T - \frac{1}{2}) = \\
step(T - T - \frac{1}{2}) = step(-\frac{1}{2}) = 0
\end{aligned} \tag{137}$$

Therefore, under the assumption that *all* $k$ fixed features are within $S_0$, there *exists* an input $\mathbf{y} \in \{0,1\}^n$ such that for the input $(\mathbf{x}_S; \mathbf{y}_{\bar{S}})$, both $f_1$ and $f_2$ classify $f_1((\mathbf{x}_S; \mathbf{y}_{\bar{S}})) = 0$ and $f_2((\mathbf{x}_S; \mathbf{y}_{\bar{S}})) = 0$. Next, we will demonstrate that the input $(\mathbf{1}_S; \mathbf{0}_{\bar{S}}) \neq \mathbf{1}^n$, i.e., it contains *at least* one feature with value "0". This is because, for the defined $f_1$ and $f_2$, it is not possible that both $f_1(\mathbf{1}^n) = 0$ and $f_2(\mathbf{1}^n) = 0$, as this would require:

$$\begin{aligned}
f_1(\mathbf{1}^n) = 0 \wedge f_2(\mathbf{1}^n) = 0 \iff \\
[step(\Sigma_{i=1}^n (1 \cdot -\mathbf{z}_i) + (T - \frac{1}{2})) = 0] \wedge [step(\Sigma_{i=1}^n (1 \cdot \mathbf{z}_i) + (-T - \frac{1}{2})) = 0] \iff \\
[\Sigma_{i=1}^n (1 \cdot -\mathbf{z}_i) + (T - \frac{1}{2}) < 0] \wedge [\Sigma_{i=1}^n (1 \cdot \mathbf{z}_i) + (-T - \frac{1}{2}) < 0] \iff \\
[\Sigma_{i=1}^n (1 \cdot \mathbf{z}_i) > T - \frac{1}{2}] \wedge [\Sigma_{i=1}^n (1 \cdot \mathbf{z}_i) < T + \frac{1}{2}] \iff \\
T - \frac{1}{2} < \Sigma_{i=1}^n (1 \cdot \mathbf{z}_i) < T + \frac{1}{2}
\end{aligned} \tag{138}$$

Given that the $\mathbf{z}_i$ values are positive integers, this is only true if $\Sigma_{i=1}^n (1 \cdot \mathbf{z}_i) = T$, which contradicts our assumption. Therefore, for $\mathbf{1}^n$, it is not the case that both $f_1(\mathbf{1}^n) = 0$ and $f_2(\mathbf{1}^n) = 0$. Consequently,

the *existing* input $\mathbf{y} \in \{0,1\}^n$ such that both $f_1((\mathbf{x}_S; \mathbf{y}_{\bar{S}})) = 0$ and $f_2((\mathbf{x}_S; \mathbf{y}_{\bar{S}})) = 0$ is not $\mathbf{1}^n$, i.e., $(\mathbf{1}_S; \mathbf{0}_{\bar{S}}) \neq (\mathbf{1}_S; \mathbf{1}_{\bar{S}}) = \mathbf{1}^n$. In this scenario, it also holds that for this input $f_5((\mathbf{1}_S; \mathbf{0}_{\bar{S}})) = 0$, since $f_3$ is an indicator for the input $\mathbf{1}^n$. Therefore, for *any* $k$ fixed inputs originating solely from $S_0$, there always *exists* an $(\mathbf{1}_S; \mathbf{0}_{\bar{S}})$, such that the perceptron majority $(f_1, f_2, f_5)$, and thus the ensemble overall, classifies this input $(\mathbf{1}_S; \mathbf{0}_{\bar{S}})$ as "0", contrary to the initial classification. Consequently, given our premise that $\langle \mathbf{z}, S_0, k, T \rangle \notin k\text{-}GSSP^*$, no $k$ features from $S_0$ qualify as a sufficient reason with respect to $\mathbf{1}^n$.

In the second scenario, we examine the case where *not all* $k$ fixed features are in $S_0$, which, under our assumption, means the MSR (Minimum Sufficient Reason) consists of *at most* $(k-1)$ fixed values of "1" in $S_0$. We will show why this is also infeasible. We consider any completion with exactly $|S| - k$ "0" values in the features of $S$ to illustrate this point. Using the same logic as previously, based on the configuration of $f_1$ and $f_2$, it is established that for any input –--- *at least* one of $f_1$ or $f_2$ classifies it as "0". As demonstrated, *exactly* one Perceptron of the pair classifies it as "0" when the summation of $\mathbf{z}_i$-s does not equal $T$, and both of them classify it as "0" when the summation equals $T$.

Furthermore, any input with at most $k - 1$ "0" values in features corresponding to $S$ is classified by $f_5$ as "0", since $f_3$ activates only when it detects *at least* $k$ of $S$'s features as "1". Additionally, any input that includes more than a single zero value is clearly not equal to $\mathbf{1}^n$ and, therefore, will also result in $f_5$ classifying it as "0", given that $f_5$ serves as an indicator for $\mathbf{1}^n$ (classifying any other input as 0"). Thus, in this scenario as well, there is a majority of $(f_1, f_3, f_5)$ or $(f_2, f_3, f_5)$, and consequently the entire ensemble, classifying any such input as "0".

This indicates that any sufficient reason for our ensemble must have a size of at least $(k + 1)$, and therefore: $\langle (f_1, \ldots, f_5), \mathbf{x}', k' \rangle \notin$ MSR for $k$-ensembles of Perceptrons when $k = 5$. In summary, we have demonstrated that $\langle \mathbf{z}, S_0, k, T \rangle \in k\text{-}GSSP^* \iff \langle (f_1, \ldots, f_5), \mathbf{x}', k' \rangle \in$ MSR for $k$-ensembles of Perceptrons when $k = 5$. Thus, solving the MSR query for $k$-ensemble Perceptrons is para-$\Sigma_2^P$-hard.

$\square$