# OpenReview forum: "On the (un) interpretability of Ensembles: A Computational Analysis"
_ICLR.cc/2025/Conference — Submitted to ICLR 2025_

### Official Review · Reviewer_dQgK · 2024-11-01

**Soundness:** 3
**Presentation:** 3
**Contribution:** 3
**Rating:** 6
**Confidence:** 3

**Summary:**

The paper studies the computational complexity of computing explanations of ensembles.

**Strengths:**

The paper is mostly well written and easy to follow even if you do not have a strong background in computational complexity theory -- the last time I had to deal with this was when I was a student. Given that, I would not consider myself as an expert in this area -- i.e. you should take all my comments with a pinch of salt.

However, checking the proofs for correctness is beyond my level of expertise.

**Weaknesses:**

Given that the paper is purely theoretical, I think it would be nice to highlight the consequences/recommendations for practitioners—e.g., limiting the number of models or the size of the base models, etc. Also, I would be interested in having an algorithm (even a naive one) that can be applied to ensembles for which the stated properties are fulfilled—although this might be too much for a conference paper.


Minor:
"while ensembles consisting of a limited number of decision trees can be interpreted efficiently, ensembles that consist of a small (even constant) number of linear models are computationally intractable to interpret" -- when first reading this (abstract) I had trouble understanding the difference between "limited number" vs. "small". Eventually it became clear throughout the paper -- I think rephrasing this statement would make it more clear for the average reader (e.g. "... , but limiting the number of linear models does not make the problem tractable" or smth. similar).

**Questions:**

What do you consider the major takeaway or recommendation for practitioners?

---

> ### Author Response · Authors · 2024-11-19
>
> We appreciate the reviewer's insightful comments and have provided our responses below.
>
>
> **What are the major takeaways and recommendations for practitioners?**
>
> We thank the reviewer for emphasizing the need to better highlight this point, which will enhance the paper’s applicability. We will provide a brief description here, and provide an elaborate discussion of this matter in our final draft. In our response, we will slightly simplify complexity notations by referring to problems like NP-Hard, $\Sigma^P_2$-Hard, etc. as “exponentially hard”. While these classes differ in difficulty, it is widely believed none admit polynomial-time solutions.
>
>
> **(1) When a practitioner attempts to provide explanations over an arbitrary ensemble**:
>
> In general terms, before getting into more specific parameterized results, we prove that computing many types of explanations for ensembles is *exponential* with respect to their size. This is in stark contrast to the ability to provide explanations over (interpretable) base models, where computing explanations is polynomial (usually - linear) with respect to their size. This fundamentally shows a strict difference between ensembles and their base models and provides lower bounds over the capability of generating many types of explanations for ensembles.
>
> For example, this can help a practitioner compare the interpretability of a decision tree with, say, $X$ nodes - where explaining its decisions scales nearly linearly with the number of nodes - with an ensemble of, say, $Y$ decision trees, each having $Z$ nodes, where the runtime grows approximately exponentially with respect to the size of the ensemble. The exponential increase in complexity for ensembles means that the difficulty of interpreting them escalates rapidly, which must be factored in when striving for interpretability.
>
>
>
> **(2) When a practitioner tries to interpret an ensemble with very small base models**:
>
> Since ensembles are exponentially hard to interpret, a practitioner might consider different ways to simplify the ensemble to enhance its interpretability. Our first parameterized result demonstrates that even when the ensemble comprises very small base models (e.g., a tree ensemble consisting of decision trees with just 2-3 nodes), the overall ensemble remains exponentially hard to interpret. This finding indicates that, in practice, reducing the size of the base models does not improve interpretability; the ensemble as a whole continues to exhibit exponential complexity for generating explanations, relative to its size. This is yet another negative finding regarding the interpretability of these models, offering evidence for the challenges or potential infeasibility of explaining decisions in this context.
>
>
>
> **(3) When a practitioner tries to interpret an ensemble with a small number of trees**:
>
> However, if a practitioner seeks to simplify an ensemble by focusing on ensembles with a small number of base models, this approach can indeed enhance the interpretability of the ensemble. We demonstrate that this is particularly true for ensembles composed of decision trees (e.g., XGBoost, Random Forest, etc.). For example, if the practitioner considers an ensemble with a very small number of decision trees, even if the individual trees are arbitrarily large, the interpretability of the ensemble remains *polynomial* with respect to its size - similar to standalone decision trees. Therefore, this scenario provides a positive interpretability outcome.
>
>
>
> **(4) When a practitioner tries to interpret an ensemble with only 2 linear models**
>
> Unlike the previous result, if the ensemble already contains a fixed number of linear base models (commonly just 2 base models), the overall ensemble immediately becomes exponentially difficult to interpret relative to its size. This emphasizes to practitioners that incorporating linear models into an ensemble leads to rapid loss of interpretability, even with as few as 2 linear models that are incorporated in their ensemble.
>
> **Can you provide algorithms for which tractable results are fulfilled?**
>
> Yes, while most of our findings highlight *negative* complexity results, emphasizing fundamental lower bounds (i.e., the lack of polynomial-time algorithms) and demonstrating the challenges of generating explanations in various contexts, we also uncover some positive results. Specifically, we show that diverse types of explanations can be efficiently produced in polynomial time for ensembles when limiting the number of decision trees (for models like XGBoost, random forests, etc.), utilizing polynomial-time algorithms. Although these algorithms are currently described in text within the appendix, we agree that presenting them as pseudo-code would enhance the clarity and accessibility of our work. We will make this adjustment in the final version.

---

> > ### Comment · Reviewer_dQgK · 2024-11-19
> >
> > Thanks for your clarifications. I will consider adjusting my score after reading the other reviews -- as I said, my expertise in computational complexity is rather limited.

---

### Official Review · Reviewer_h6P8 · 2024-11-01

**Soundness:** 3
**Presentation:** 2
**Contribution:** 1
**Rating:** 5
**Confidence:** 4

**Summary:**

The paper provides an analysis of the computational complexity of ensemble models’ interpretability. The authors investigate whether explaining ensemble models is inherently more computationally demanding than explaining individual models. The authors find that explaining ensembles made of interpretable base models, e.g., decision trees, is computationally more expensive than the base models. However, there is no gap in the computational complexity between explaining expressive, uninterpretable models, e.g., neural networks, and their ensembles.

The paper also studies the parameterized complexity of explaining ensembles, examining how specific factors, e.g., the size or the number of base models, affect interpretability. The results show that reducing the size of the base models in the ensemble does not make the ensemble interpretable. The effect of the number of base models on the interpretability depends on their type, i.e., linear models or decision trees. Ensembles with a small number of decision trees can be interpreted efficiently, while a small number of linear models in an ensemble makes it computationally intractable to interpret.

**Strengths:**

1- The manuscript provides original work based on a theoretical foundation.

2- The paper shows differences between linear model ensembles and decision tree ensembles in parameterized complexity results, which reveals that not all ensembles are equally hard to interpret.

**Weaknesses:**

1- The paper’s contribution is marginal and provides evidence for the already-known un-interpretability of ensemble models, as the authors mention in line 530: “Our work provides mathematical evidence for the folklore belief: “ensembles are not interpretable”.” However, the paper succeeds in showing that not all ensembles are equally hard to interpret.

2- The paper lacks motivation for the targeted problem and why the contribution can be significant. It can be helpful if the introduction is expanded to include examples of how the findings can impact research or practice in explainable AI.

3- The authors claim that the main focus of the paper was on understanding the complexity of ensemble models and their impact on model interpretability. However, the explanations of the compared models were not evaluated or compared using explainability-related metrics, e.g., fidelity, robustness, or using a user-based evaluation. Therefore, it can be helpful to clarify explicitly that the focus is on the theoretical computational complexity of explanations, not an evaluation of interpretability in general.

**Questions:**

Why can the findings of this work be significant to researchers in the domain of interpretability and explainable AI if the findings are already “folklore belief”?

---

> ### Author Response · Authors · 2024-11-19
>
> Thank you for your thoughtful comments. We appreciate the review, as it has brought attention to several important points in our work that we believe need further emphasis. We hope our response addresses the concerns raised effectively.
>
> In **Q1**, we will answer why the core issue of the (un)-interpretability of ensembles is inherently nuanced and multifaceted from a computational view, despite often being regarded as “folklore”. In **Q2**, we will highlight the *practical implications* of our work for explainable AI. In **Q3**, we will discuss how our approach differs from other forms of interpretability.
>
>
> **Q1: Why is mathematically analyzing the (un)-interpretability of ensembles interesting if it is already “folklore”?**
>
> While the assertion that "ensembles are not interpretable" is indeed a widely held belief, our analysis approaches this question through the lens of computational complexity. This perspective offers a far more *nuanced* understanding of the topic, uncovering many unexpected insights that go beyond the conventional folklore argument. Simply claiming that "ensembles are not interpretable" fails to address key aspects regarding the interpretability of ensembles that our computational complexity framework is capable of answering. Some of these questions are:
>
> 1. Although ensembles are often considered "less interpretable" than their base models, the precise mathematical nature of this gap remains unclear - are they *polynomially* harder to interpret, or are they *exponentially* harder? This is a critical and fundamental question with potentially far-reaching implications. Our results provide an in-depth and nuanced analysis of this issue.
>
> 2. How does the interpretability of ensembles vary across different types of explanations? We demonstrate that there can be a significant disparity between explanation types (we analyze 5 different common forms of explanations) - are some explanations harder to obtain than others?
>
> 3. How does the interpretability of ensembles vary across different model types, such as decision trees, linear models, and neural networks? Our findings once again highlight the intricate cases of this complexity analysis.
>
> 4. How do various attributes of an ensemble influence its interpretability? How does the size of the base models impact interpretability? How does the number of base models affect interpretability? Which is "more interpretable": an ensemble with a few large models or one with many smaller models? Does the answer depend on other factors, such as the type of base models or the explanation method used?
>
> 5. The former question also has practical significance and touches on a practitioner's ability to *simplify* an ensemble to enhance its interpretability. Specifically, if an ensemble (initially uninterpretable) is modified by significantly *reducing the size* of each base model, does it suddenly become interpretable? Additionally, does *reducing the number* of base models make the entire ensemble more interpretable? Could the answer to this also depend on the types of base models used and the forms of explanations applied?
>
> Our work addresses these questions, offering a significantly deeper and more detailed mathematical understanding of ensemble interpretability, extending far beyond the initial folklore notion that ensembles are inherently (un)-interpretable. We agree with the reviewer that these aspects, along with the overall contributions of our work, should be more prominently highlighted, and we will incorporate these revisions into the final draft.

---

> > ### Author Response · Authors · 2024-11-19
> >
> > **Q2: What are the implications these results have for researchers in explainable AI?**
> >
> > We thank the reviewer for emphasizing this point. We will address it in our final version. Some of the key implications include:
> >
> > **1. For many ensemble configurations and popular explanation forms (SHAP, sufficient explanations, contrastive explanations, etc.), finding explanations for the ensemble has an exponential lower bound relative to its size (under standard complexity assumptions).**
> >
> > For many widely used explanation methods and model types, we establish a fundamentally negative complexity result. This underscores the challenges of explaining the decisions of ensemble models, which have complexity exponential in their size, compared to more interpretable models like linear models and decision trees, where explanations exhibit polynomial - and often linear - complexity relative to their size. This establishes key limitations and lower bounds for generating explanations in explainable AI.
> >
> > **2. Even when simplifying an ensemble by reducing the sizes of base models to a constant (regardless of the base-model type), obtaining popular explanations like SHAP, sufficient explanations, or contrastive explanations remains exponentially hard with respect to their size.**
> >
> > Now, let us consider an ensemble consisting of an arbitrary number of models, each limited to a size of 3. This reflects an effort by a practitioner to simplify the ensemble by reducing the size of each individual model significantly. However, our findings demonstrate that for a wide range of models and explanation forms, the ensemble as a whole remains exponentially hard to interpret, even under these constraints. This constitutes a negative result which again limits the applicability of producing explanations for ensembles in this setting.
> >
> > **3. If we simplify a tree ensemble (XGBoost, random forest, etc.) by reducing the number of trees within it, obtaining many types of popular explanations (SHAP, sufficient explanations, contrastive explanations, etc.) becomes tractable concerning their size.**
> >
> > However, let us now explore a different approach to "simplify" the ensemble. Consider an ensemble of decision trees (such as XGBoost, Random Forest, etc.) constrained to a relatively small, fixed number $k$ of trees while allowing each tree to be arbitrarily large. In this case, the practitioner attempts to simplify the model by reducing the number of trees. We demonstrate that generating explanations in this setting is computationally tractable, offering *positive* complexity results. This finding is shown by giving  *practical algorithms* that can effectively provide explanations in such scenarios (such as XGboost or random forest with a reduced number of arbitrarily large decision trees). As recommended by reviewer dQgK, we will convert these poly-time algorithms, currently described in text form in the appendix, into pseudo-code to enhance their accessibility.
> >
> > **4. If we have only 2 linear models in an ensemble, then many popular explanation forms already become exponentially hard to obtain**
> >
> > In contrast to the previous point, we show that any ensemble consisting of just two linear models (and only five for one specific form of explanation) already becomes exponentially difficult to interpret (under standard complexity assumptions). This highlights *negative* complexity results concerning the (un)-interpretability of ensembles containing even a constant number of linear models, offering valuable insights for practitioners into the infeasibility of providing explanations in such cases.
> >
> > **Q3: Highlighting that the results are related to complexity aspects, and not metrics (e.g., infidelity) or human evaluations**
> >
> > We agree with the reviewer that our results indeed primarily address the mathematical and computational aspects of generating explanations for ensembles across a diverse range of settings. They do not however touch on other concepts that relate to interpretability such as assessing explanation quality via different metrics or conducting human evaluations. The definition of interpretability is inherently elusive and typically revolves around the extent to which *humans* can comprehend the decisions made by ML models. In contrast, our work centers on recent efforts to develop a more formal and mathematically grounded perspective on interpretability [e.g., 1-4]. We appreciate the reviewer bringing up this point and will make sure to clarify this distinction in the final version.
> >
> > [1] Model Interpretability through the Lens of Computational Complexity (Barcelo et al., Neurips 2020)
> >
> >
> > [2] Foundations of Symbolic Languages for Model Interpretability (Arenas et al., Neurips 2021)
> >
> >
> > [3] Local vs. Global Interpretability: A Computational Complexity Perspective (Bassan et al., ICML 2024)
> >
> >
> > [4] A Theory of Interpretable Approximations (Bressan et al., COLT 2024)

---

> ### Comment · Reviewer_h6P8 · 2024-11-23
>
> Thank you for providing detailed clarifications and responses to the questions raised. After reviewing the authors' responses, I found that some of my concerns were adequately addressed, and several of the arguments presented were convincing. Therefore, I raised the paper's rating. However, I still believe that the paper's contribution offers limited practical implications for researchers in the field of interpretability and explainable AI.

---

> > ### Author Response · Authors · 2024-11-24
> >
> > We thank the reviewer for raising their score and are happy to address any remaining questions or provide further clarification if needed.

---

### Official Review · Reviewer_42Bg · 2024-11-04

**Soundness:** 3
**Presentation:** 1
**Contribution:** 4
**Rating:** 6
**Confidence:** 2

**Summary:**

The paper theoretically studies the class of ensemble models from an interpretability perspective. Specifically, the paper discusses the computational complexity associated with different kinds of "interpretability queries" (i.e. SHAP, counterfactuals) and different kind of ensemble models (deep ensembles, tree ensembles, etc.). The paper finds evidence that ensembles are in fact less interpretable than base models. This is substantiated by an extensive list of theoretical results.

**Strengths:**

- The subject matter is an important research area within machine learning, making this research highly relevant. The focus aligns well with ongoing discussions and research questions (TreeSHAP + Tree vs. TreeSHAP + Forest). The paper adds a good contribution to explainability and theoretical foundations in AI.
- A clear strengths of this paper are the theoretical contributions, of which there are many. While I have problems with the presentation of the results, they are very interesting and like the authors say "give theoretical merit to the folklore saying that ensembles are less interpretable than base models". I do very much like the paper because of this. I particularly like the additional results for the decision tree classes (FBDDs) and the non-SHAP related explanation queries.
- All in all, while the paper is very technical, the writing is strong and precise.

**Weaknesses:**

- **The paper is too technical**, which could lead to the ICLR audience might missing the core contributions. It contains many acronyms that are not generally known, making it difficult for readers to follow along. Furthermore, one theoretical result follows another without putting the results well into context or grounding it. At the moment the paper contains 8 Theorems and 8 propositions making it 16 theoretical results on 6 pages. Lines 246-258 are just a proof sketch right where the central part of the paper could be. This may hinder the audience from fully appreciating the significance and implications of each result within the broader field. The appendix is quite long (44 pages give or take) and contains a lot of details missing in the main text. This reinforces the impression that the contribution may not be well suited to a conference paper and would maybe better fit a journal like JMLR.
- While the paper is sound and provides a plethora of proofs, I am **missing** some **empirical validation** or at least **illustrative examples**. The work stays very abstract. I acknowledge that computational complexity results do not necessarily need "experiments", however they do help a lot in grounding the theoretical results for practitioners or uncover edge cases.
- The paper does not contain a limitations section.

**Questions:**

- How can you make the paper more approachable provided an additional page?
- I highly suggest, you move technical proof sketches out of the main paper and put the results better into context and tell the reader why result X is meaningful for model class B. You may further streamline the paper by moving more minor results into the appendix all together.

---

> ### Author Response · Authors · 2024-11-19
>
> We thank the reviewer for the valuable comments. See our response below.
>
>
> **The main paper includes too many technical results. How can you make the paper more approachable?**
>
>
> We appreciate the reviewers' many suggestions on potential ways to restructure the paper. We agree that the structure of our current draft can be improved to better emphasize the main results and their implications while reducing the focus on technical details. We believe that we can address these adjustments effectively in the final version, especially with the additional page available. Specifically, we plan to:
>
>
> **1. Move proof sketches to the appendix:** Following the reviewer’s suggestion, we will instead position our proof sketches at the beginning of each proof in the appendix, and not in the main text. This will preserve the main paper's focus on fundamental ideas, corollaries, and implications, ensuring the overall flow remains unaffected. Essentially, this structure enables readers to choose their level of engagement: they can focus on the main text to understand the key corollaries, review the proof sketches provided at the beginning of each proof in the appendix, or engage fully with proofs for a deeper exploration of the paper's technical details.
>
>
> **2. Reduce the number of corollaries:** Some of the corollaries in our paper can be combined together to emphasize one larger point. This can reduce the total number of propositions and corollaries (which, as was highlighted by the reviewer, is quite high) and will leave more space to discuss ideas, examples, and implications.
>
>
> **3. Improve discussion on practical implications:** Based on the reviewers' feedback, we will refine our paper to place greater emphasis on discussing the practical implications of our findings. These include both the fundamental lower bounds - such as the lack of polynomial-time algorithms of interpreting ensembles (under standard complexity assumptions), which holds both generally, as well as under even highly simplified configurations like ensembles with constant base-model sizes or a constant number of linear models, as well as the more optimistic complexity results, which demonstrate the feasibility of poly-time computations for generating diverse explanations on ensembles of decision trees when the number of trees is reduced. To underscore these points, we will include specific examples that clearly illustrate our arguments. Based on suggestions from reviewer dQgK, we will also include pseudocode to present some of these results, enhancing their accessibility.
>
>
>
>
> **4. Adding illustrative examples:** Based on the reviewers' feedback and the additional page allowance, we will include illustrative examples in our work to enhance the applicability of some results. We will specifically achieve this using a running example. For instance, we will provide examples such as an ensemble of decision trees with $X$ models of size $Y$, among others, including ensembles of different types of models (such as containing decision trees, linear models, etc.). In this manner, we will highlight the differences in complexity across various types of explanations in diverse scenarios. This approach will enable us to specify the parameters of the ensembles and demonstrate how they influence complexity across various configurations.
>
>
>
>
> **The paper does not contain a limitations section**
>
> Although our work does not include a dedicated limitations section, we address various limitations of our framework throughout the main text. These include: (1) the restriction of our framework, like other works in this area, to specific explanation forms and model types, and (2) the potential to extend our approach to various additional settings and domains, many of which are preliminarily discussed in the appendix. In the final draft, we plan to incorporate an explicit limitations section, given the additional page allowance.

---

> > ### Comment · Reviewer_42Bg · 2024-11-22
> > **Acknowledgement of Rebuttal**
> >
> > Dear Authors,
> >
> > **Thank you** for responding to my review. I now have seen that ICLR'2025 **does not provide an additional page upon acceptance**, which makes your **suggested changes even more important to actually do**. Your suggestions are good and will improve the quality of the paper. However, as there is no updated manuscript, I cannot judge how well the suggestions can be incorporated into your manuscript. This is why my score remains unchanged: Your paper is a good technical contribution. This is why it think it is above the acceptance threshold. However, because of its too technical presentation and accessibility problems, I do not think it ranks higher than this.
> >
> > Sincerely,
> >
> > Reviewer 42Bg

---

> > > ### Author Response · Authors · 2024-11-23
> > >
> > > Dear Reviewer 42Bg,
> > >
> > > Thank you for your thoughtful suggestions on improving the accessibility of our paper and for your interest in seeing these changes in the revised manuscript. We recognize the importance of these adjustments, especially given the lack of an extra page allowance for ICLR papers this year. We have implemented several of the changes we committed to during this rebuttal period and uploaded a revised version of the manuscript. However, due to time constraints of the rebuttal period, we were only able to address *some* of the suggestions at this stage. We are fully committed to implementing additional improvements in the final version based on your suggestions and those of the other reviewers.
> > >
> > >
> > > Specifically, our current revisions include:
> > >
> > > 1. As per the reviewer's suggestion, we have relocated the proof sketches to the beginning of each proof in the appendix, while the main text now includes only references to these proofs.
> > >
> > > 2. We streamlined the presentation by consolidating several propositions and corollaries, as well as integrating some corollaries directly into the main text. This refinement reduced the total number of theoretical statements from 16 to 10.
> > >
> > > 3. As suggested by the reviewer, we have added a *concise figure on page 7* to highlight key high-level concepts illustrated by our parameterized complexity results. This addition aims to enhance the accessibility and understanding of our findings.
> > >
> > > 4. We have included a dedicated *main contributions* section in a prominent position within the introduction (page 2) to better emphasize the significance of our work.
> > >
> > > 5. As per the reviewer's suggestion, we have included a limitations and future work section on page 10.
> > >
> > > 6. We have reduced the use of less familiar abbreviations in the paper. Specifically, in our two main tables (Table 1 on page 5 and Table 2 on page 8), we now use the full names of the explanation forms, such as "minimum sufficient reason" instead of only "MSR," to enhance clarity. Additionally, we have replaced certain terms, like "FBDDs," with more accessible alternatives, such as "decision trees," within the main text. The rigorous definitions, along with categorizations of the models we use, are retained in the appendix for reference.
> > >
> > > We hope the updates made so far have resolved your concerns, and as noted earlier, we remain committed to implementing further improvements. These include adding a running example, providing pseudo-code for our algorithms, further reducing the use of less familiar abbreviations, and enhancing the focus on practical implications.
> > >
> > > Thank you once again for your valuable feedback.

---

> > > > ### Author Response · Authors · 2024-12-02
> > > >
> > > > Dear Reviewer 42Bg,
> > > >
> > > > Thank you once again for your detailed and thoughtful feedback, as well as for the many valuable suggestions on how to enhance our work.
> > > >
> > > > We have made efforts to improve the accessibility of several aspects of our paper in line with your suggestions. Furthermore, we intend to include further accessibility improvements in the final version, as mentioned in our response.
> > > >
> > > > As the rebuttal period comes to a close, we would greatly appreciate it if you could let us know if there are any additional questions or concerns we can address.
> > > >
> > > >
> > > > Best regards,
> > > >
> > > > The Authors

---

### Official Review · Reviewer_GUuE · 2024-11-04

**Soundness:** 4
**Presentation:** 4
**Contribution:** 4
**Rating:** 8
**Confidence:** 2

**Summary:**

The paper presents an expanded framework for analyzing the complexity of interpreting ensemble models in comparison to their constituent base models and to each other. The authors investigate three broad categories of base models—FBDDs (generalized trees), linear models, and multilayer perceptrons (MLPs)—alongside three families of interpretability metrics (referred to as 'explainability queries'): Sufficient Reasons, Contrastive Reasons, and SHAP values. This framework encompasses a wide range of interpretability approaches commonly used in machine learning. The authors provide an in-depth computational analysis of interpretability for these models, demonstrating that the computational costs differ significantly when comparing base models with ensembles, particularly in the cases of linear models and tree ensembles.

To extend this analysis, the authors apply parameterized complexity techniques to explore how model size and the number of base models affect the computational complexity. One key insight from this parametric perspective is that, while increasing the number of models in a linear model ensemble does not impact interpretability tractability, limiting the number of trees in a tree ensemble can render interpretation computationally feasible.

**Strengths:**

* The authors offer a well-rounded perspective that reflects a wide range of approaches used in the field. They introduce a comprehensive and formal framework for analysing interpretability complexity across different model types and interpretability metrics.
 This makes the work broadly applicable and useful for various ML subfields, especially formal analysis on interpretability methods.
* The use of parameterized complexity to analyze the effect of model size and the number of base models adds depth to the analysis. This aspect of the work allows for a nuanced view of interpretability that reveals under what conditions interpretability is tractable, which is particularly innovative and sheds more insight on formal complexity of interpretability in ensembles of different classes.
* The paper is well-structured, coherent, and logically organized. The authors thoughtfully delegate details to Appendices, where they provide comprehensive and easy to follow proofs. Furthermore, including proof sketches in the main text is an excellent choice, as it allows readers to understand the core ideas without extensive back-and-forth with the Appendices.

**Weaknesses:**

While there are no major weaknesses in the scope, there is always room for expansion. For instance, discussing the interpretability of regression models versus classifiers more extensively, exploring continuous domains and fundamental differences (which is actaully touched on in an Appendix), and extending metrics to observational SHAP and other interpretability metrics. However, the current scope is already comprehensive, and these suggestions could serve as directions for future work. To be fair, extending it in any above-mentioned directions could end up costing the clarity.

**Questions:**

Given the assumption of feature independence for Shap values, is the work still includes methods like interventional TreeSHAP, which are proved to be computing SHAP values under the same assumption [1], while leveraging the tree structures in the ensemble rather than computing the intractable original SHAP formula? Is such methods that are computing exact SHAP values not directly from the original SHAP formula something to be considered in your work?

[1] Laberge, Gabriel, and Yann Pequignot. "Understanding interventional treeshap: How and why it works." arXiv preprint arXiv:2209.15123 (2022).

---

> ### Author Response · Authors · 2024-11-19
>
> We appreciate the reviewer’s detailed and insightful comments. Please find our responses below.
>
>
> **Extension to diverse setting configurations**
>
>
> We agree that exploring additional settings where our results could apply presents compelling directions for future research. As the reviewer noted, we have already touched on some of these topics in the appendices. Importantly, the fundamental nature of sufficient and contrastive explanations does not change in terms of complexity when transitioning from classification to regression, as a subset $S$ can be defined to ensure the prediction remains or changes within a $\delta$ range [1, 2]. However, this is not necessarily the case for SHAP, which indeed warrants further investigation in future studies. We thank the reviewer for highlighting the matter and will make sure to emphasize it further in the final version.
>
>
>
>
> **Extension of complexity results for interventional SHAP**
>
>
> We agree with the reviewer’s comment and acknowledge that including a discussion on interventional SHAP will add valuable insights to the final draft. We note that interventional SHAP aligns with conditional expectation SHAP when the feature independence assumption holds [3]. Consequently, the results of our framework apply to this scenario as well. We note that while interventional SHAP was indeed shown to be computationally feasible for diverse tree-based models, this tractability diminishes when moving to classification tasks, unlike in regression settings (see [4,5]). However, our work indeed provides a parameterized extension of this finding, by demonstrating that reducing the number of decision trees that participate in the ensemble - whether in the classification or the regression setting  - enables polynomial-time computations of explanations for the ensemble. Since this result holds for SHAP under conditional expectations and assuming the feature independence assumption, it also directly applies to interventional SHAP, due to the intersection of the two definitions in this configuration. We recognize the importance of this observation and will address it in the final version of the paper. Thank you for highlighting this!
>
>
> [1] Verix: Towards Verified Explainability of Deep Neural Networks (Wu et al., Neurips 2023)
>
>
> [2] Distance-Restricted Explanations: Theoretical Underpinnings & Efficient Implementation (Izza et al., KR 2024)
>
>
> [3] The Many Shapley Values for Model Explanation (Sundararajan et al., ICML 2020)
>
>
>
>
> [4] On the Tractability of SHAP Explanations (Van den Broeck et al., JAIR 2022)
>
>
> [5] Updates on the Complexity of SHAP Scores (Huang et al.,  IJCAI 2024)

---

### Official Review · Reviewer_24uH · 2024-11-04

**Soundness:** 2
**Presentation:** 2
**Contribution:** 2
**Rating:** 3
**Confidence:** 3

**Summary:**

Authors develop a theoretical basis for evaluation of the interpretability of different ensembles (voting or weighted-voting) with different base models: linear, decision tree and neural network. Authors examine computational complexity of deriving different types of the explanations for ensembles and compare them to single base learners, and provide mathematical guarantees. Authors focus on 3 types of explainability: Sufficient Reason Feature Selection (SRFS), Contrastive Explanations (CF), Shapley Value Feature Attributions (SVFA). Authors analyze a parametrized complexity to show how different parameters such as base-model size, number of base learners affect ensemble interpretability.

**Strengths:**

- The paper provides computational complexity analysis for ensembles of different type of base models with mathematical guarantees (and proofs).
- Authors reviewed different explainability queries

**Weaknesses:**

- Besides mathematical proofs and guarantees, the contribution of the paper is very questionable. The conclusions authors made are well-known. For example, weighted voting of neural networks can be expressed as a bigger neural network with another linear layer, which is still a neural network, therefore, complexity gap should not be high (if any). Similarly, ensembles with a constant number of linear models can be shown (for specific ensemble type) as a neural network of two layers with non-linear activation (for classification), which is for interpretable.
- In my opinion, the most interesting part of the paper is hidden in the appendix part and needs restructuring.

**Questions:**

N/A

---

> ### Author Response · Authors · 2024-11-19
>
> Thank you for your comments. We value the review, as it has provided us with valuable guidance on improving the clarity of our paper and highlighted several important aspects that should indeed be made more explicit. We hope that our response clarifies the concerns that were raised.
>
> **A specific concern that was mentioned regarding two different results being considered trivial**
>
> The reviewer labels two specific results as trivial: (1) adding a linear layer to a neural network, which shows no complexity gap between neural networks and their ensembles. While this is indeed straightforward (as we explicitly say in the paper), it occupies only a significantly minor part of our proofs (only a few lines), while the bulk of the paper concerns numerous non-trivial cases and highly non-trivial proofs. (2) Reducing ensembles of linear models to neural networks, is claimed to lead to trivial conclusions about interpreting ensembles with a constant number of linear models. We will clarify why this implication does not hold. For our detailed response:
>
>
>
>
>
>
>
> (1) Regarding the first point, it is correct that the lack of a complexity gap between neural networks and ensembles is not surprising, and we did not present it as such; the proof is indeed brief. Your point about adding a linear layer aligns directly with the proof we used, which we explicitly describe in the paper. The reason we emphasize this point is to highlight the *contrast* between interpretable models, which show a strict complexity gap between individual models and ensembles, and more expressive models, where this gap vanishes, clarifying differences in interpretability behavior. That said, following the reviewers' feedback, we will emphasize this more clearly in the final version.
>
>
>
>
>
>
>
>
> (2) We respectfully disagree with the reviewer’s second argument but appreciate them raising it, as it allows us to clarify this point further in the final version. While an ensemble of a constant number of linear models can indeed be reduced to a neural network, this does *not* imply that such ensembles are computationally hard to interpret.
>
>
> Proving the computational hardness of interpreting an ensemble of a constant number of linear models by leveraging the hardness of interpreting neural networks would (hypothetically) require *a reduction in the reverse direction*. Specifically, one would need to be able to reduce any neural network to a constant-sized ensemble of linear models. However, this approach is impractical because neural networks are substantially more expressive, and reducing them to an ensemble of linear models would require exponential time and space. This makes such a reduction unachievable within the constraint of a constant number of linear models. This is contrary to the reviewer's suggested reduction. If the implication that was made was valid, we could make the same argument about decision trees and linear models, as they too can be reduced in polynomial time to neural networks. However, this is indeed not the case - providing explanations for these models can be done in polynomial time (and is not computationally hard) - because the *reverse reduction* does not hold.
>
>
>
>
>
>
> Contrary to claims of triviality, our results show that proving that interpreting ensembles of a constant k number of linear models is computationally hard requires highly technical reductions (see, e.g., Lemma 22). Notably, this holds even for a very small k, such as k=2 for most explainability queries, and k=5 for the MSR query, demonstrating that ensembles with as few as 2 linear models can be intractable to interpret. That said, following the reviewer's feedback, we recognize that these points should be clarified more effectively in the revised text, and we will make the necessary updates accordingly.

---

> > ### Author Response · Authors · 2024-11-19
> >
> > **The contributions of this work**:
> >
> >
> > We agree that our main contributions should be more clearly highlighted, and we briefly emphasize them here. Our work explores the computational complexity of generating various explanations for ensembles in multiple settings including: (1) an analysis of five popular explanation types (SHAP, sufficient explanations, contrastive explanations, etc.), (2) the coverage of three base-model families - decision trees, linear models, and neural networks, and (3) a detailed study of how ensemble attributes, like base model size and count, influence the complexity of explanation generation.
> >
> >
> > Some of our main results are:
> >
> >
> > (1) Proving that, for various types of explanations and models, generating explanations for ensembles is exponential (under standard complexity assumptions) in their size, unlike interpretable models such as decision trees and linear models, where explanations can be derived in polynomial time relative to their size. This result establishes fundamental lower bounds on explanation generation for ensembles and has substantial practical implications.
> >
> >
> > (2) Developing a comprehensive range of computational complexity results (e.g., NP-completeness, $\Sigma^P_2$-completeness, #P-completeness, etc.) for various explanation types across different ensemble models with distinct base-model types. This highlights the inherent complexity of the problem across diverse scenarios. These results offer valuable insights into the varying levels of tractability and intractability associated with computing different forms of explanations for different ensemble configurations.
> >
> >
> >
> >
> > (3) Proving that even a highly *simplified* version of an ensemble, composed of models with a *constant* size, remains computationally intractable to interpret for many types of explanations and base models. This establishes a critical lower bound (i.e., the lack of polynomial-time algorithms), demonstrating that reducing the size of all base models in an ensemble does not inherently make it more "interpretable" and sets a foundational limit for numerous explainable AI algorithms.
> >
> >
> >
> >
> > (4) We prove that in a different simplified version of an ensemble, consisting of a reduced number of base models (even when each base model is arbitrarily large), it can be interpreted in *polynomial time* relative to the ensemble's size if the base models are decision trees. This is particularly relevant for many popular ensemble models like XGBoost and Random Forest. Our results enable practical algorithms that can be employed to derive various forms of explanations in this context.
> >
> >
> >
> >
> > (5) Lastly, we demonstrate that even a significantly simplified version of an ensemble, containing a constant number of linear models (often as few as two for most explanation types), becomes exponentially hard (under standard complexity assumptions) to interpret as a whole. This, again, establishes fundamental lower bounds for this scenario with significant practical implications on explainable AI algorithms.
> >
> >
> >
> >
> > Following the reviewer’s feedback, we will clarify these points in the final version.
> >
> >
> > **The appendix includes the most important parts of the paper and the need for reconstruction**
> >
> >
> > While we agree with the reviewer that the appendices are lengthy due to several technical proofs, we note that we follow common practice and provide full proofs in the appendix and sketches in the main text. As all technical elements are explicitly referenced, we believe the paper is self-contained. That said, we agree restructuring could enhance clarity. Based on this feedback and Reviewer dQgK’s input, we plan to streamline proof sketches and better emphasize our contributions and their implications in the main text.

---

> ### Author Response · Authors · 2024-11-19
>
> We thank the reviewer for raising their score to a 6. We hope we have thoroughly addressed all your concerns and would be happy to respond to any further questions.

---

> > ### Comment · Reviewer_24uH · 2024-11-25
> >
> > I have thoroughly reviewed the responses, other reviews, and the paper itself. While the authors have clearly invested substantial effort, several issues remain concerning the paper’s contribution, relevance, and structure.
> > A key concern lies in the definition and assumptions around interpretability. The authors acknowledge that interpretability is a long-studied and not well-defined concept. While they propose their own definition, it is neither widely accepted nor practically relevant. The paper assumes there is a common agreement around interpretability criteria that does not exist. For example, the authors claim that Shapley values are "commonly recognized" as an interpretability metric, yet these values are widely criticized in the field.
> > The structure of the paper is another significant issue. At 60 pages, including the appendix, it is overly dense and challenging to go through. Frequent references to material in the appendix make the paper even harder to follow. For a conference submission, the work would benefit from being split into several smaller, focused papers with clearer organization and presentation.
> > Some of the conclusions drawn in the paper are trivial or widely known. For instance, the discussion in lines 78–92 simply reiterates the well-established fact that ensembles are not interpretable. Insights derived from the authors’ formulation add little practical value for the machine learning community, limiting the paper’s broader impact.
> > There are also technical inaccuracies that require attention. For example, the authors discuss decision trees without specifying that they are axis-aligned, which is an important distinction since oblique decision trees (with linear splits) also exist. Additionally, the treatment of MCR (also known as counterfactual explanations) in Table 1 as a novel contribution is misleading, as its NP-completeness is already well known.
> > A broader concern is the disconnect between the theoretical results presented and the practical needs in machine learning. The authors focus on binary inputs, yet most practical ML systems handle continuous data naturally. While it is possible to discretize continuous inputs, such an approach is rarely necessary in practice and limits the relevance of the proposed methods.
> > Overall, while the paper presents interesting theoretical work, its current form lacks the clarity, practical relevance, and structural organization necessary for a strong conference submission. Significant revisions are needed to address these issues effectively.

---

> > > ### Author Response · Authors · 2024-11-27
> > >
> > > Dear Reviewer 24uH,
> > >
> > > We appreciate your thorough evaluation of our paper and the insightful points raised. We are confident that we can address all of these new concerns through our response and by clarifying a few points in the final version. Thank you for the thoughtful and constructive feedback.
> > >
> > > **The input setting**
> > >
> > > Perhaps the most major concern that was raised by the reviewer regards the notion that the proofs in this paper are presented in the binary setting, and hence may not be applicable to other settings. As explicitly mentioned in the main paper, we adopt this focus on binary inputs, which aligns with common conventions on the topic [1-3], with the purpose of simplifying the presentation of the proofs - this is purely a technical choice. Importantly, all our results can be generalized to any discrete setting. Additionally, nearly all of them can be extended to continuous settings.
> > >
> > > More specifically, the above statement includes all complexity results for all explanation forms except for Shapley values (though almost all results apply there as well), i.e., sufficient explanations, contrastive explanations, etc., and all model types. For SHAP, all results established for decision tree ensembles also apply to continuous settings. For non-tree models (specifically in the context of SHAP), hardness results extend to other models as well, though some membership proofs for high-complexity classes cannot be directly carried over. However, since these significantly "intractable" classes (e.g., #P) are already well-known for their difficulty, proving *membership* in these classes is less critical - the hardness results are more significant, and they still hold. Importantly, the *tractable* results we establish for decision tree ensembles (such as XGBoost and random forests), i.e., the polynomial time algorithms, also apply to continuous settings.
> > >
> > > To conclude, only a very small and, we believe, insignificant portion of our results is not applicable to the continuous setting.
> > >
> > > We understand the importance of emphasizing this point in the main paper (currently it is only mentioned in the paper and elaborated on in the appendix), and will adjust the manuscript accordingly. We thank the reviewer for raising this point.
> > >
> > > **A computational complexity view of interpretability:**
> > >
> > > We agree with the reviewer that the term "interpretability" is inherently elusive - a point we explicitly acknowledge and discuss in our paper, as the reviewer has also noted. The term “interpretability” typically refers to the ability of *humans* to comprehend the decisions made by ML models. However, as the reviewer observed, we adopt a more mathematically grounded perspective on interpretability, analyzing it through the lens of computational complexity. The reviewer expressed concerns that the computational complexity view of interpretability might have been newly coined in our paper. However, we emphasize that this concept has been previously explored in the literature (e.g., [1-3]), and our work builds on an existing line of research rather than introducing it for the first time.
> > >
> > > However, we want to stress that we are not claiming that studying interpretability through a computational lens is the "definitive" approach to understanding interpretability, which, as noted earlier, remains an inherently elusive concept. Instead, we adopt this perspective to investigate how analyzing the computational complexity of generating different types of explanations for ensembles can contribute to a more rigorous grasp of the interpretability of these models in various contexts. This perspective does not necessarily align with other definitions of interpretability, such as the *human* capacity to understand these models.
> > >
> > > That said, we agree that this point could benefit from additional clarification. In the final version, we will ensure this aspect is thoroughly addressed and clearly emphasize that references to the "interpretability" of an ML model are framed from a computational perspective. Thank you for highlighting this important detail.

---

> > > > ### Author Response · Authors · 2024-11-27
> > > >
> > > > **The first point in the main contributions section is mentioned as trivial**
> > > >
> > > > The reviewer suggests that the first part of our main contributions, addressing intractability results for ensembles, is trivial.
> > > >
> > > > Firstly, we want to emphasize that the core of this work lies in the *parameterized complexity* proofs, which constitute the bulk of our contributions. These proofs provide novel insights, particularly into the impact of various problem parameters on the (computational) interpretability of ensembles in different contexts - findings that we regard as quite significant. Respectfully, we find that the reviewer’s comment overlooks this significant part of our paper. We will adjust the final manuscript to highlight these contributions more explicitly.
> > > >
> > > > Secondly, while it is true that the idea of ensembles being "uninterpretable" is a commonly held assumption, this does not imply that the precise complexity behavior of these models is trivial, even outside the parameterized framework. For example, how does interpretability complexity vary with different types of base models? How does it change with different explanation techniques? Addressing these questions required deriving results across a diverse range of complexity classes - including NP, coNP, $\Sigma^P_2$, #P, pseudo-polynomial time, and others. All of these are integrated into our work, demonstrating a wide range of complex behaviors that are quite non-trivial. This is in addition to our parameterized complexity results, which represent the central contribution of this work, and expands our spectrum of findings even further. We will highlight this point in the final version of the manuscript.
> > > >
> > > >
> > > >
> > > >
> > > > **The validity of Shapley value explanations:**
> > > >
> > > > The reviewer rightly observes that some papers have addressed specific challenges in using Shapley values as an explanation tool (e.g., [4], along many others). However, it is important to emphasize that Shapley values remain one of the most widely adopted feature attribution techniques. Furthermore, the issue of validity is not exclusive to Shapley values but applies broadly to nearly all post-hoc explanation methods. It is widely recognized that no single explanation method “perfectly" captures a model’s internal workings. Different methods, such as sufficient explanations, contrastive explanations, Shapley values, or other forms of additive attributions, each offer distinct advantages relative to one another.
> > > >
> > > >
> > > > This indeed highlights a potential limitation in studying interpretability through a computational lens, as it necessitates focusing on *various* forms of explanations, for which the complexity can behave differently. However, to assess the broader implications of obtaining explanations, it is common for such frameworks to examine *multiple* explanation types rather than a single form (e.g., [1-3]). In our work, we analyze five distinct types of explanations, encompassing various interpretability approaches such as sufficient explanations, contrastive explanations, and additive feature attributions. While we acknowledge that additional explanation forms could be proposed - an explicit limitation of our study which is mentioned - we believe that our comprehensive analysis across diverse explanation types offers valuable insights into the computational aspects of generating explanations for ensemble models.
> > > >
> > > >
> > > > **The length of the appendix**
> > > >
> > > > We acknowledge that the appendices of our paper are long due to the inclusion of several technical proofs. However, it is a well-established norm for theoretical papers at ICLR to feature detailed appendices, with the main text often containing only references or outlines of these proofs. We also highlight that our paper is self-contained, as it consolidates many proofs of claims to present cohesive and unified concepts. For instance, we provide a series of proofs demonstrating that interpreting ensembles of linear models becomes computationally intractable with just a constant number of linear models in the ensemble. These results are presented for various explanation forms (e.g., para-NP, para-$\Sigma^P_2$-hardness), unified under Proposition 4 in the paper.  Similarly, we show that certain explanation forms can be computed in polynomial time by reducing the number of trees in a decision tree ensemble, which is detailed under Proposition 5. Thus, while there are numerous proofs, they collectively contribute to a coherent narrative.

---

> > > > > ### Author Response · Authors · 2024-11-27
> > > > >
> > > > > **Axis aligned vs. oblique decision trees**
> > > > >
> > > > > The reviewer points out that we refer to general decision trees without mentioning that they are axis-aligned. This is a standard assumption in many ML works, especially given models such as random forests and XGBoost which are popular ensemble models which typically incorporate axis-aligned decision trees and not oblique ones. We respectfully disagree with the reviewer as categorizing this as a technical inaccuracy of the paper. Moreover, we explicitly define the exact formal structure of our decision trees within the paper. However, in light of the reviewer's comment, we will indeed make sure to emphasize that our ensembles are axis-aligned rather than oblique. Thank you for bringing this to our attention.
> > > > >
> > > > >
> > > > > **Practical implications**
> > > > >
> > > > > Our work has significant practical implications for understanding the tractability and intractability of obtaining explanations for ensemble models. Our findings cover a wide range of popular explanation forms, ensemble types, and base-model configurations. They highlight when explanations for ensembles are computationally feasible and when they are not, offering valuable insights for practitioners. In particular, our results are directly applicable to widely used ensemble methods such as XGBoost and random forests, as well as various popular explanation techniques. Therefore, we respectfully disagree with the claim that our findings lack practical relevance. That said, we acknowledge the importance of emphasizing these implications more clearly, and we will work on enhancing this aspect in the final version of our paper.
> > > > >
> > > > > **The MCR query**
> > > > >
> > > > > The reviewer has raised concerns about the novelty of our results related to this query. First, we would like to clarify that the MCR query is not identical to counterfactuals but instead focuses on contrastive explanations (although the two terms are indeed related). Specifically, the MCR query identifies subsets of *features* that alter a prediction, while counterfactuals focus on feature *assignments* that cause a prediction to change. Specifically, the MCR query identifies a minimal subset of features that may cause a prediction to change. The MCR query is a well-established concept in analyzing the computational complexity of explanation methods (e.g., [1-3]). To our knowledge, there are no prior results concerning this query for the ensembles studied in this work. If the reviewer knows of a relevant reference for an NP-Hardness proof of this problem, we would be happy to incorporate it into our final version. Additionally, it is worth noting that the (non-parameterized) MCR query represents only a minor aspect of our work (requiring just a few lines of proof), while the primary contribution of our paper lies in the non-trivial proofs addressing parameterized complexity configurations.
> > > > >
> > > > >
> > > > >
> > > > > [1] Model Interpretability through the Lens of Computational Complexity (Barcelo et al., Neurips 2020)
> > > > >
> > > > >
> > > > > [2] Local vs. Global Interpretability: A Computational Complexity Perspective (Bassan et al., ICML 2024)
> > > > >
> > > > >
> > > > > [3] Foundations of Languages for Interpretability and Bias Detection (Arenas et al., Neurips 2021)
> > > > >
> > > > > [4]  The many Shapley values for model explanation (Sundararajan et al., ICML 2020)

---

> > > > > > ### Author Response · Authors · 2024-12-02
> > > > > >
> > > > > > Dear Reviewer 24uH,
> > > > > >
> > > > > > Thank you once again for your thorough and insightful feedback, which has been invaluable in highlighting areas of our paper that could benefit from further clarification.
> > > > > >
> > > > > > We believe we have addressed all new concerns that were raised and can incorporate some clarifications into the final version to resolve any remaining points.
> > > > > >
> > > > > > As the rebuttal period nears its conclusion, we would appreciate knowing if you have any additional questions or concerns that we could address.
> > > > > >
> > > > > > Best regards,
> > > > > >
> > > > > > The Authors

---

### Comment · Area_Chair_EhqB · 2024-11-13
**authors - reviewers discussion open until November 26 at 11:59pm AoE**

Dear authors & reviewers,

The reviews for the paper should be now visible to both authors and reviewers. The discussion is open until November 26 at 11:59pm AoE.

Your AC

---

> ### Comment · Area_Chair_EhqB · 2024-11-25
>
> Dear reviewers,
>
> The authors have provided individual responses to your reviews. Can you acknowledge you have read them, and comment on them as necessary? The discussion will come to a close very soon now:
> - Nov 26: Last day for reviewers to ask questions to authors.
> - Nov 27: Last day for authors to respond to reviewers.
>
> Your AC

---

### Author Response · Authors · 2024-11-19
**Summary of the rebuttal phase**

We thank the reviewers once again for their insightful feedback and for recognizing the significance of our work.

Due to the many discussions, we will provide a short overview on the author-reviewer discussions -

1. 3 out of 5 reviewers (GUuE, 42Bg, and dQgK) provided positive feedback on the paper, with scores of 8, 6, and 6, respectively. Reviewer dQgK (score: 6) expressed openness to increasing their score. Reviewer 42Bg (score: 6) indicated that addressing certain accessibility concerns in a revised manuscript could lead to a higher score. We incorporated most of these suggested changes into the manuscript. However, the reviewer has not yet responded to the updated submission.

2. Reviewer 24uH initially assigned a score of "3", citing two concerns that we addressed in our rebuttal. As a result, the reviewer raised their score to a "6". However, on the last day of the (original) rebuttal period, the reviewer raised a new set of concerns and reverted the score back to "3". *We believe we have effectively addressed these issues.* For example, a key concern about the applicability of our results to continuous domains was resolved when we clarified that our findings are indeed relevant to such domains. The reviewer has not yet responded to these points following our rebuttal.



3. Finally, reviewer h6P8 initially questioned the significance of studying ensemble (un)-interpretability. We responded with a detailed explanation of its importance, addressing overlooked aspects of the folklore claim that our computational complexity framework can address, as well as important practical implications relevant to explainable AI. This led the reviewer to raise their score.




Overall, while all reviewers acknowledged the significance of the theoretical aspects of our work, a primary pertaining concern is our paper’s accessibility. During the rebuttal period, we made significant revisions to address this concern and are committed to incorporating further adjustments, as outlined in individual threads, in the final version.


We sincerely appreciate the reviewers' detailed and valuable feedback, which has helped us improve our work!

---

### Meta-Review · Area_Chair_EhqB · 2024-12-19

**Metareview:**

The paper provides an analysis of the interpretability of ensemble models from a computational point of view. In general, the reviewers found the paper hard to understand, more like a summary of a longer journal paper. There were several important reasons mentioned by one or more reviewers not to accept the paper: a narrow focus in the way the problem is set up, that may not be of general interest; little novelty in the main results; and little impact in what is new, at least concerning practical impact in interpretability and explainable AI.

**Additional Comments On Reviewer Discussion:**

N/A

---

### Decision · Program_Chairs · 2025-01-22

Reject